# Enabling Differentially Private Federated Learning for Speech Recognition: Benchmarks, Adaptive Optimizers and Gradient Clipping

**Martin Pelikan**[*], **Sheikh Shams Azam**[*], **Vitaly Feldman**, **Jan "Honza" Silovsky**,
**Kunal Talwar**, **Christopher G. Brinton**[†], **Tatiana Likhomanenko**[*]
Apple, Purdue University[†]
{mpelikan,s_azam,vitalyf,jsilovsky}@apple.com
cgb@purdue.edu,{ktalwar,antares}@apple.com

## Abstract

While federated learning (FL) and differential privacy (DP) have been extensively studied, their application to automatic speech recognition (ASR) remains largely unexplored due to the challenges in training large transformer models. Specifically, large models further exacerbate issues in FL as they are particularly susceptible to gradient heterogeneity across layers, unlike the relatively uniform gradient behavior observed in shallow models. As a result, prior works struggle to converge with standard optimization techniques, even in the absence of DP mechanisms. To the best of our knowledge, no existing work establishes a competitive, practical recipe for FL with DP in the context of ASR. To address this gap, we establish **the first benchmark for FL with DP** in end-to-end ASR. Our approach centers on per-layer clipping and layer-wise gradient normalization: theoretical analysis reveals that these techniques together mitigate clipping bias and gradient heterogeneity across layers in deeper models. Consistent with these theoretical insights, our empirical results show that FL with DP is viable under strong privacy guarantees, provided a population of at least several million users. Specifically, we achieve user-level $(7.2, 10^{-9})$-DP (resp. $(4.5, 10^{-9})$-DP) with only a 1.3% (resp. 4.6%) absolute drop in word error rate when extrapolating to high (resp. low) population scales for FL with DP in ASR. Although our experiments focus on ASR, the underlying principles we uncover — particularly those concerning gradient heterogeneity and layer-wise gradient normalization — offer broader guidance for designing scalable, privacy-preserving FL algorithms for large models across domains. Code of all experiments and benchmarks is available at https://github.com/apple/ml-pfl4asr.

## 1 Introduction

Federated learning (FL) allows training models in a distributed manner without storing data centrally on a server [1]. While FL eliminates privacy risks associated with data aggregation, it remains vulnerable to inference attacks [2, 3, 4, 5, 6]. Stronger user-level privacy guarantees can be achieved by combining FL with differential privacy (DP) [7, 8] and secure aggregation [9, 10]. FL introduces several challenges in training including: heterogeneous data distribution [11, 12], sensitivity to cohort size [13], and slower convergence rate due to local training [14]. A *practical* FL with DP with limited privacy budget also limits extensive hyper-parameter tuning as it incurs additional privacy overhead apart from communication and computation cost [15, 16], thus necessitating robust training strategies.

---

[*]Equal contribution.

39th Conference on Neural Information Processing Systems (NeurIPS 2025).

Consequently, training end-to-end (E2E) automatic speech recognition (ASR) models using FL is also challenging [17, 18, 19, 20, 21], primarily due to the inherently heterogeneous data [22, 20] across clients but also exacerbated by the depth of the models [23, 24]. Additionally, training large transformer-based models [25, 26, 27, 28] that underlie most E2E ASR models require optimization techniques such as learning rate warm-up and decay, gradient clipping, adaptive optimizers, careful initialization, etc. [29, 30, 31]. Moreover, FL alone provides limited privacy even in the context of ASR [32, 21]. This work is, to our knowledge, the first to demonstrate a practical training recipe to enable FL with DP for ASR, along with a strong benchmark and supporting convergence guarantees.

Most prior works on both FL and DP rely on small-scale models, primarily due to (i) communication complexity [33] and (ii) the difficulty in training large-scale models with DP [34, 35, 36]. We argue that: (i) practical model sizes are steadily increasing – including for ASR [37] – and (ii) the optimization of larger over-parametrized models is often easier [38][2]. To address this gap in understanding large-scale models in the context of FL and DP and to mitigate the optimization challenges associated with training smaller models, we focus exclusively on a large vanilla transformer model for ASR in this work. Our **key contributions** can be summarized as follows:

 (i) We *empirically study the performance of FL with DP on E2E ASR* using a large (250M parameters) vanilla encoder-based transformer model trained with the Connectionist Temporal Classification (CTC) loss [40]. Based on the results, we successfully establish the first practical and competitive benchmark and baselines for FL with DP in ASR with realistic $(\varepsilon, \delta)$-DP guarantees.

 (ii) We systematically analyze the impact of several key FL factors – including *data heterogeneity, optimization hyperparamters, and seed models initialization* (pre-trained with or without domain shift) – on convergence and performance of ASR trained under FL and FL with DP.

(iii) We revisit *per-layer clipping* – deemed ineffective by prior works – and demonstrate that combining it with *layer-wise adaptive gradient normalization* is the key to achieving strong model performance under FL with DP. Furthermore, we provide a rigorous *theoretical analysis of the algorithm's convergence properties*, offering insights into observed empirical behavior.

We show that FL can be used to train competitive models for several datasets, covering English, German, French languages: FL models are at worst $\sim 0.3\%$-$1.4\%$ absolute word error rate (WER) behind the corresponding central models with a limited number of central steps. Competitive models are obtained even with heterogeneous data, especially when the training starts from a seed model. The seed model can even come from another domain and perform relatively poorly on the target dataset. We also show that FL with user-level DP, which is more preferable to example-level DP, and large models is viable for E2E ASR and promising even for low-resource languages. With per-layer clipping, our models achieve $(7.2, 10^{-9})$-**DP** (resp. $(4.5, 10^{-9})$-**DP**) with $1.3\%$ (resp. $4.6\%$) degradation in absolute WER for extrapolations to high (resp. low) population scale for **FL with DP in ASR**.

## 2 Federated Learning with Differential Privacy: Background and Notation

**Federated Learning (FL)**   In this paper, we focus on synchronous cross-device FL where only a small fraction $q$ of users (clients) participate in each step of central (global) aggregation (step), where $K$ is the total number of users (population): every user is sampled i.i.d. with probability $q$ from all users, and $S = qK$, termed *cohort size*, is the expected number of users participating in every central step. Users do not maintain a state across central steps. Each user $k$ has its own local data $\boldsymbol{x} \sim \mathcal{D}_k$, where $\boldsymbol{x} \in \mathbb{R}^N$ and $\mathcal{D}_k$ is $k$-client's data distribution ($\boldsymbol{x}$ is paired audio and the corresponding ground-truth transcription for ASR task). The objective of FL is to minimize the total loss function $\mathscr{L}(\boldsymbol{\theta})$ given the ASR parameters $\boldsymbol{\theta} \in \mathbb{R}^D$ and all user data: $\min_{\boldsymbol{\theta} \in \mathbb{R}^D} \{ \mathscr{L}(\boldsymbol{\theta}) \triangleq \sum_{k=1}^K \omega_k \mathscr{L}_k(\boldsymbol{\theta}) \}$, where $w_k > 0$, $\sum_{k=1}^K \omega_k = 1$, $\mathscr{L}_k(\boldsymbol{\theta}) = \mathbb{E}_{\boldsymbol{x} \sim \mathcal{D}_k} [\ell(\boldsymbol{x}, \boldsymbol{\theta})]$ and $\ell(\boldsymbol{x}, \boldsymbol{\theta})$ is a loss function for a sample $\boldsymbol{x} \in \mathbb{R}^N$. In practice, we optimize $\mathscr{L}(\boldsymbol{\theta})$ by sampling a set of users $\mathcal{K}^t$ at a central step $t$ who receive a copy of latest global model $\boldsymbol{\theta}^{(t)}$. Each client $k$ then performs optimization over the local copy of the global model $\boldsymbol{\theta}_k^{(t,0)} = \boldsymbol{\theta}^{(t)}$ using their own data $\boldsymbol{x} \sim \mathcal{D}_k$ via the update step $\boldsymbol{\theta}_k^{(t,t_{\mathrm{loc}}+1)} = \boldsymbol{\theta}_k^{(t,t_{\mathrm{loc}})} - \eta_{\mathrm{loc}} \mathbf{g}_k^{(t,t_{\mathrm{loc}})}$ at step $t_{\mathrm{loc}}$, where $\mathbf{g}_k(\boldsymbol{\theta}) = \mathbf{g}_k(\mathscr{B}_k, \boldsymbol{\theta})$ (e.g. obtained by SGD) is an estimator of the $\nabla \mathscr{L}_k(\boldsymbol{\theta})$, and $\mathscr{B}_k = \{\boldsymbol{x}_i\}_{i=1}^B, \boldsymbol{x}_i \sim \mathcal{D}_k$. The clients periodically upload their model updates

---
[2]Distillation from a large to a small model remains the dominant method for training compact models [39].

**Algorithm 1:** Federated learning with differential privacy (marked as red)

---

**Inputs:** Initial model $\boldsymbol{\theta}^0$ (either randomly initialized or pre-trained on server data), weights $\omega_k \in (0,1)$ such that $\sum_{k=1}^{K} \omega_k = 1$, central steps $T$, central optimizer $\mathrm{opt}$, clients sampling rate $q = S/K$, local steps $T_{\mathrm{loc}}$, local optimizer $\mathrm{opt}_{\mathrm{loc}}$, clipping function $\mathrm{clip}(\boldsymbol{v}, C) = \boldsymbol{v} \cdot \left( \frac{C}{\max(C, \|\boldsymbol{v}\|)} \right)$, local clipping bound $C_{\mathrm{loc}}$, DP clipping bound $C$ and DP noise $\sigma_{\mathrm{DP}}$.

**Result:** ASR model $\boldsymbol{\theta}^T$

1   Initialize central optimizer $\mathrm{opt}$
2   **for** $t = 1, 2, \ldots, T$ **do**
3     Sample every client i.i.d. with probability $q$ to form a subset $\mathscr{K}^t$ of clients from all clients $\mathscr{K}$ ($|\mathscr{K}| = K$)
4     // For practical implementation we fix the size of the cohort $\mathscr{K}^t$ to $S$ throughout the training.
5     **for** $i = 1, 2, \ldots, |\mathscr{K}^t|$, $k_i \in \mathscr{K}^t$ **in parallel do**
6       Initialize local model $\boldsymbol{\theta}_{k_i}^{(t,0)} \leftarrow \boldsymbol{\theta}^{(t-1)}$ and local optimizer $\mathrm{opt}_{\mathrm{loc}}$
7       **for** $t_{\mathrm{loc}} = 1, 2, \ldots, T_{\mathrm{loc}}$ **do**
8         // We also use local epochs instead of steps: then this loop has different number of steps per client.
9         Sample train mini-batch $\mathscr{B}_{k_i}^{(t_{\mathrm{loc}})} \in \mathcal{D}_{\mathscr{K}_{k_i}^t}$ and compute gradient estimate $\mathbf{g}_{k_i}^{(t,t_{\mathrm{loc}})}(\mathscr{B}_{k_i}^{(t_{\mathrm{loc}})}; \boldsymbol{\theta}_{k_i}^{(t,t_{\mathrm{loc}}-1)})$
10         Clip gradients $\mathbf{g}_{k_i}^{(t,t_{\mathrm{loc}})} \leftarrow \mathrm{clip}(\mathbf{g}_{k_i}^{(t,t_{\mathrm{loc}})}, C_{\mathrm{loc}})$ and update a local model $\boldsymbol{\theta}_{k_i}^{(t,t_{\mathrm{loc}})} \leftarrow \mathrm{opt}_{\mathrm{loc}}(\mathbf{g}_{k_i}^{(t,t_{\mathrm{loc}})})$
11       Compute client's delta $\boldsymbol{\Delta}_{k_i}^{(t)} = \boldsymbol{\theta}_{k_i}^{(t,0)} - \boldsymbol{\theta}_{k_i}^{(t,T_{\mathrm{loc}})} = \boldsymbol{\theta}^{(t-1)} - \boldsymbol{\theta}_{k_i}^{(t,T_{\mathrm{loc}})}$
12       Clip client's delta $\boldsymbol{\Delta}_{k_i}^{(t)} \leftarrow \mathrm{clip}(\boldsymbol{\Delta}_{k_i}^{(t)}, C)$
13       Add Gaussian noise to client's delta $\boldsymbol{\Delta}_{k_i}^{(t)} \leftarrow \boldsymbol{\Delta}_{k_i}^{(t)} + \mathcal{N}\left(0, \boldsymbol{I} C^2 \sigma_{\mathrm{DP}}^2 \frac{q}{\sum_{k=1}^{K} \omega_k^2}\right)$
14     Compute central model's pseudo-gradient $\mathbf{g}^{(t)} = \boldsymbol{\Delta}^{(t)} = \frac{1}{q} \sum_{i=1}^{|\mathcal{K}^t|} \omega_{k_i} \boldsymbol{\Delta}_{k_i}^{(t)}$
15     Update the central model $\boldsymbol{\theta}^{(t)} \leftarrow \mathrm{opt}(\mathbf{g}^{(t)})$

---

$\boldsymbol{\Delta}_k^{(t)}$ to the server after $T_{\mathrm{loc}}$ local steps given by $\boldsymbol{\Delta}_k^{(t)} = \boldsymbol{\theta}_k^{(t,0)} - \boldsymbol{\theta}_k^{(t,T_{\mathrm{loc}})} = \eta_{\mathrm{loc}} \mathbf{G}_k^{(t)}$ where $\mathbf{G}_k^{(t)} = \sum_{t_{\mathrm{loc}}=0}^{T_{\mathrm{loc}}-1} \mathbf{g}_k^{(t,t_{\mathrm{loc}})}$. The server then aggregates the updates $\boldsymbol{\Delta}^{(t)} = 1/q \sum_{k_i \in \mathcal{K}^t} \omega_{k_i} \boldsymbol{\Delta}_{k_i}^{(t)}$ and performs the central model step either through conventional federated averaging [41], or through an adaptive optimizer [42]. The updated central model is broadcasted to another sampled set of users and the process is repeated either for a fixed number of central steps $T$ or until convergence.

**FL with Differential Privacy (DP)**   Since no prior work exists that can efficiently train private FL for ASR, we establish the first competitive baselines for private FL in ASR in the rest of the paper. We start by referring to DP [43, 44, 7], which provides a mathematical formalism of guarantees on the amount of information learnt by machine learning models from the user private data:

**Definition 1.** *Differential privacy: A randomized mechanism $\mathcal{M} : \mathcal{D} \to \mathcal{R}$ with a domain $\mathcal{D}$ (e.g., possible training datasets) and range $\mathcal{R}$ (e.g., all possible trained models) satisfies $(\varepsilon, \delta)$-differential privacy if for any two adjacent datasets $D, D' \in \mathcal{D}$ and for any subset of outputs $R \subseteq \mathcal{R}$ it holds that $Pr[\mathcal{M}(D) \in R] \le e^{\varepsilon} Pr[\mathcal{M}(D') \in R] + \delta$.*

One key DP component is **adjacent datasets** [7]. In some applications, prior works consider the example-level privacy [45, 8]. For FL where each user has multiple data points, user-level [46] is preferable to example-level privacy [45, 8]. We thus use the following adjacency relation:

**Definition 2.** *User-adjacent datasets: Let $D$ and $D'$ be two datasets of training examples, where each example is associated with a user. Then, $D$ and $D'$ are adjacent if $D'$ can be formed by adding or removing all of the examples associated with a single user from $D$.*

To incorporate user-level DP into FL, the client updates $\boldsymbol{\Delta}_k^{(t)}$ are: (i) clipped such that their $l_2$ norm is bounded, i.e., $\|\boldsymbol{\Delta}_k^{(t)}\|_2 \le C$ at every central training step $t$ and then (ii) perturbed via Gaussian mechanism, such that client updates under FL with DP are given by $\boldsymbol{\Delta}_k^{(t)} + \mathcal{N}\left(0, \boldsymbol{I} C^2 \sigma_{\mathrm{DP}}^2 \frac{q}{\sum_{i=1}^{K} \omega_i^2}\right)$, where $\boldsymbol{\Delta}_k^{(t)} = \eta_{\mathrm{loc}} \alpha_k^{(t)} \mathbf{G}_k^{(t)}$ and $\alpha_k^{(t)} = \frac{C}{\max\left(C, \|\eta_{\mathrm{loc}} \mathbf{G}_k^{(t)}\|\right)}$. We use the moments accountant [8] to achieve tight privacy bounds and restate the main theorem of [46] in our parametrization of noise added to every user's model update before averaging, assuming $\omega_k = 1/K$ for simplicity:

**Theorem 1.** *For the DP-mechanism in Algorithm 1, the moments accountant of the sampled Gaussian mechanism correctly computes privacy loss with the noise scale of $z = \sigma_{\mathrm{DP}}/\mathbb{S}$ and central steps $T$, where $\mathbb{S} = 1/(qK)$ and noise $\sigma_{\mathrm{DP}}$, probability of user selection $q$, and total number of users in the population $K$ are given in Algorithm 1.*

Although this work uses the moments accountant and uniform sampling, alternative approaches such as DP-FTRL [47] or device-level sampling [10] can also be applied. These alternatives are expected to yield similar results, potentially at the cost of a small constant overhead in the required population sizes. Since we use large transformer ASR models, user-level DP significantly reduces the utility of training ASR models even in the absence of FL because the noise overpowers the gradients [48, 34]. Our initial experiments confirmed this problem, which we mitigate via per-layer clipping. *The FL with DP and corresponding terminology are summarized in Algorithm 1.*

## 3    Theoretical Analysis: Adaptive Optimizers and Per-Layer Clipping

**LAMB Optimizer.**    We utilize the layer-wise adaptive optimizer LAMB [49] for updating the global model using pseudo-gradient $\Delta^{(t)}$ (see Appendix E.4 for its definition). Originally proposed for the large batch training, LAMB scales learning rate for each layer using the ratio of weight norms to the gradient norms (termed *trust ratio*), which makes it particularly effective in handling the gradient scale disparities in deep networks. We posit LAMB is helpful in large model training using FL since inter-layer gradient heterogeneity is further exacerbated by "divergence accumulation" [23, 24] wherein *deeper layers demonstrate higher divergences in contrast to the shallow.*

**Per-Layer Clipping.**    Per-layer clipping was proposed by [46]. However, the authors did not report a significant improvement in their setting of LSTM models for language. On the contrary, our work shows that for FL with DP and large transformer models, per-layer clipping mitigates the imbalance of gradients across different layers in the attention blocks. Formally, we change the global clipping of clients' deltas from Algorithm 1, Step 12, to per-layer clipping $\text{clip}_{layer}(\mathbf{g}, C)$ defined as follows:

**Definition 3.** *Per-layer clipping: Let the model gradient be $\mathbf{g} = (\mathbf{g}_1, \mathbf{g}_2, ..., \mathbf{g}_H)$, where $\mathbf{g}_h$ is the h-th layer gradient with total $H$ layers in the model. Then per-layer clipping with clipping parameter $C = \sqrt{\sum_{h=1}^{H} C_h^2}$ is given as $\text{clip}_{layer}(\mathbf{g}, C) = (\tilde{\mathbf{g}}_1, \tilde{\mathbf{g}}_2, ..., \tilde{\mathbf{g}}_H)$ where $\tilde{\mathbf{g}}_h = \text{clip}(\mathbf{g}_h, C_h)$.*

In our experiments we use either $C_h = \frac{C}{\sqrt{H}}$ ("uniform" variant) or $C_h = C\sqrt{\frac{d_h}{\sum_{i=1}^{H} d_i}}$ ("dim" variant based on a layer dimension) where $d_h$ is the dimension of the $h$-th layer and $h = 1, 2, \ldots, H$, so that after per-layer clipping we still guarantee $\|\Delta_k^{(t)}\|_2 \leq C$ necessary for Theorem 1 to hold.

**Assumptions.**    Given a global model comprising of $H$ layers, the model parameters are defined as $\boldsymbol{\theta} = (\boldsymbol{\theta}_1, \cdots, \boldsymbol{\theta}_h, \cdots \boldsymbol{\theta}_H)$. It is presumed that the loss function for each sample $\boldsymbol{x}$ is bounded below: $\min_{\boldsymbol{\theta} \in \mathbb{R}^D} \ell(\boldsymbol{x}, \boldsymbol{\theta}) > -\infty$, where $\boldsymbol{x} \sim \mathcal{D}_k, \forall k$. Let $\| \cdot \|$ denote the $l_2$-norm. Our analysis uses the following standard assumptions [12, 42, 50, 51, 52, 53, 54]:

1. *Smoothness of Loss Function Gradient:* $\nabla \ell(\boldsymbol{x}, \boldsymbol{\theta})$ is layer-wise $L_h$-smooth for $\forall h$ [49]:

$$\|\nabla_h \ell(\boldsymbol{x}, \boldsymbol{\theta}) - \nabla_h \ell(\boldsymbol{x}, \boldsymbol{\theta}')\| \leq L_h \|\boldsymbol{\theta} - \boldsymbol{\theta}'\|, \ \ \forall \boldsymbol{\theta}, \boldsymbol{\theta}' \in \mathbb{R}^D, \ \boldsymbol{x} \in \mathbb{R}^N, \forall k, \qquad \textbf{(A1)}$$

   where $\nabla_h$ denotes gradient with respect to parameters $\boldsymbol{\theta}_h$ of layer $h$.

2. *Local Gradient Property:* Given user $k$, $\mathscr{B}_k = \{\boldsymbol{x}_i\}_{i=1}^{B}, \boldsymbol{x}_i \sim \mathcal{D}_k$ and local gradient $\nabla \ell(\boldsymbol{x}, \boldsymbol{\theta})$, its unbiased estimator $\mathbf{g}_k(\boldsymbol{\theta}) = \mathbf{g}_k(\mathscr{B}_k, \boldsymbol{\theta})$ has a bounded variance $\forall k$ [12, 50, 54]:

$$\mathbb{E}_{\mathscr{B}_k} \left[ \|\mathbf{g}_k(\boldsymbol{\theta}) - \nabla \mathscr{L}_k(\boldsymbol{\theta})\|^2 \right] \leq \sigma_{\text{loc}}^2, \ \sigma_{\text{loc}}^2 \geq 0, \ \ \forall \boldsymbol{\theta} \in \mathbb{R}^D. \qquad \textbf{(A2)}$$

3. *Global Pseudo-Gradient Property:* The variance of global (pseudo-) gradient is bounded [52, 42]:

$$\sum_{k=1}^{K} \omega_k \|\nabla \mathscr{L}_k(\boldsymbol{\theta}) - \nabla \mathscr{L}(\boldsymbol{\theta})\|^2 \leq \sigma_{\text{glob}}^2, \ \ \sigma_{\text{glob}} \geq 0, \ \ \forall \boldsymbol{\theta} \in \mathbb{R}^D. \qquad \textbf{(A3)}$$

**Corollary 1.** *Assume A1.1, A2.1, A2.2, and A3, $\eta_{\text{glob}} L < 1$ and $\kappa = \left[ 1 - 8(1 - \eta_{\text{loc}} T_{\text{loc}})^2 \right] > 0$. If the trust ratio in LAMB optimizer is controlled in the Algorithm 1 (global optimizer is LAMB and local optimizer is SGD) and $\eta_{\text{glob}} = \Theta\left(\frac{1}{L\sqrt{T}}\right)$ and $\eta_{\text{loc}} = \Theta\left(\frac{1}{L\sqrt{T_{\text{loc}}T}}\right)$, then Algorithm 1 converges*

*to a stationary point of the global loss function with the convergence bound characterized as:*

$$\frac{\kappa}{T}\sum_{t=0}^{T-1}\mathbb{E}_{t_{\text{loc}}}\left[\left\|\nabla\mathscr{L}\left(\boldsymbol{\theta}^{(t)}\right)\right\|^2\right] \le \underbrace{\mathcal{O}\left(\frac{1}{\sqrt{T}}\right)}_{optimization} + \underbrace{\mathcal{O}\left(\frac{T_{\text{loc}}\sigma_{\text{glob}}^2}{T}\right)}_{global\ update\ noise} + \underbrace{\mathcal{O}\left(\frac{T_{\text{loc}}\sigma_{\text{loc}}^2}{T}\right)}_{local\ update\ noise}$$

$$+ \underbrace{\mathcal{O}\left(C^2\sigma_{\text{DP}}^2\sum_{h=1}^{H}R_h^2 d_h\right)}_{differential\ privacy\ noise} + \underbrace{\mathcal{O}\left(\frac{T_{\text{loc}}}{T}\sum_{h=1}^{H}\frac{M_h^2}{C_h^2}\right)}_{clipping\ bias} + \underbrace{\mathcal{O}\left(\frac{T_{\text{loc}}}{T}\sum_{h=1}^{H}\frac{R_h^2 M_h^2}{C_h^2}\left[\Psi_{\text{h}}^{\text{intra}} + \Psi_{\text{h}}^{\text{inter}}\right]\right)}_{intra\text{-}\ and\ inter\text{-}client\ update\ variance},$$

(1)

*where* $\Psi_{\text{h}}^{\text{intra}} = \mathbb{E}_{t,k}\left[\text{Var}_{t_{\text{loc}}}\left(\left\|\mathbf{G}_{h,k}^{(t)}\right\|\right)\right]$ *and* $\Psi_{\text{h}}^{\text{inter}} = \mathbb{E}_{t}\left[\text{Var}_{k}\left(\mathbb{E}_{t_{\text{loc}}}\left[\left\|\mathbf{G}_{h,k}^{(t)}\right\|\right]\right)\right]$.

Refer Appendix E for the proof of Theorem 2 and its Corollary 1 with derived asymptotic bound.

***Interpreting the Bounds.*** Corollary 1 highlights the key contributors to the convergence behavior: (i) the optimization process, (ii) global and local update noises, (iii) DP noise, and (iv) clipping. Specifically, it emphasizes the complex coupling among per-layer clipping ($C_h$), layer-wise scaling $R_h$, and intra-client ($\Psi_{\text{h}}^{\text{intra}}$) and inter-client ($\Psi_{\text{h}}^{\text{inter}}$) update variance. Although the analysis presents several of these terms separately, they are often interdependent and may interact in non-trivial ways. The remainder of this section summarizes the key takeaways from the bound in Corollary 1.

***Recovering Prior Bounds.*** As a validation, we recover bounds similar to several prior works. For example, by setting $\sigma_{\text{DP}}^2 = 0$ and letting $C_h \to \infty$, we obtain a bound similar to adaptive optimizers [42] and vanilla FL [12, 16] – modulo constant factors. Similarly, the bound for [55] can be recovered by choosing a constant clipping value $C$ for all layers and adding an appropriate DP noise. These reductions demonstrate that our result generalizes several known convergence guarantees as special cases. See Appendix E.7 for details on how specific prior bounds are recovered.

***Impact of Gradient Heterogeneity across Batches and Clients.*** The terms $\Psi_{\text{h}}^{\text{intra}}$ and $\Psi_{\text{h}}^{\text{inter}}$ in the convergence bound quantify the impact of data-heterogeneity within and across clients, respectively. Within-client heterogeneity $\Psi_{\text{h}}^{\text{intra}}$ can be reduced by shuffling data locally on each client. However, this becomes challenging when client data is limited. In such cases, data augmentation can serve as a practical alternative, reducing batch-level variance and improving performance [56]. Similarly, inter-client heterogeneity $\Psi_{\text{h}}^{\text{inter}}$ can be tackled by incorporating (i) server-side adaptive optimizers that intrinsically reduce gradient heterogeneity across clients [56], (ii) anchored optimization methods such as SCAFFOLD [57], FedProx [11], and (iii) adaptive client weighting [12].

***Trade-offs Between Clipping Constant and DP noise.*** While an inverse relationship with the clipping $C_h$ suggests that increasing $C_h$ would improve convergence [58], the proportional relationship $\sigma_{\text{DP}}^2 \propto C_h^2$ complicates the dynamics; while increasing $C_h$ reduces clipping bias, it also requires proportionally more DP noise for the same privacy guarantees. Additionally, the convergence bound indicates a linear decay of clipping bias with $T$, whereas DP noise increases linearly with it. Thus, over long training horizons, the impact of clipping becomes negligible relative to that of DP noise. However, in practical settings with limited central steps $T$, clipping bias can remain significant – particularly when gradient norm $M_h$ and intra-client ($\Psi_{\text{h}}^{\text{intra}}$) and inter-client ($\Psi_{\text{h}}^{\text{inter}}$) update variances are large. Unlike [55], we capture this coupling explicitly that underscores the importance of tuning both $C_h$ and DP noise jointly to optimize privacy-utility trade-off. Consistent with our theoretical bound, Table 1 shows a negligible impact of clipping on centralized model training whereas DP noise significantly degrades performance in FL with DP. While local clipping, used for transformer training stability [59], reduces model's sensitivity to global clipping, the model is still affected by DP noise.

***Benefits of Per-Layer Intervention.*** Our convergence bound is decomposed over several layer-wise dynamics including gradient norm $M_h$, trust ratio $R_h$, clipping constant $C_h$, and variance terms $\Psi_{\text{h}}^{\text{intra}}$ and $\Psi_{\text{h}}^{\text{inter}}$. This per-layer decomposition gives a tighter bound when: (i) heterogeneous gradient distribution is observed across layers and transformer blocks as seen in Figure 5 and Figures 17-19 and (ii) "divergence accumulation" in deep networks in FL training [60] further amplifies the mismatch across layers. Based on these observations, we only redistribute the total clipping budget

$C$ across the model via per-layer clipping $C_h$ given by $C_h = C/\sqrt{H}$ or $C_h = Cd_h/(\sum_{i=1}^{H} d_i)$, thus ensuring that overall DP noise remains unchanged. Consequently, the redistribution of clipping budget can be viewed as altering the signal to noise ratio (SNR) at the layer level relative to DP noise. In tandem, the per-layer trust ratio $R_h$ further modulates both noise scale and clipping bias. Empirically, under similar settings LAMB extracts better performance in FL with DP when compared to Adam. Advantages of LAMB was also reported by [56] showing that it improves FL when used as a local optimizer. We instead use SGD locally owing to the memory overhead of LAMB that can be prohibitive on resource-constrained devices. Together, these layer-wise treatments should empirically result in an improved convergence compared to global clipping for cases with greater gradient heterogeneity or stronger DP noise. This is in fact evident from the following observations:

(i) Per-layer clipping has a more significant impact on FL with DP compared to centralized training. This improvement is more pronounced for higher DP noise levels (see Tables 1 & 18).

(ii) Experiments on CV-en show both a higher improvement compared to CV-fr & CV-de (see Table 1 vs. Table 18) and a higher gradient diversity across layers (see Fig. 17 vs Fig. 18 & 20).

## 4 Empirical Analysis

**Data** We use **LibriSpeech (LS)** data [61]: *train-clean-100* (*LS-100*), *train-clean-360* (*LS-360*) and *train-other-500* (*LS-500*) as training data. *LS-960* is the union of *LS-100*, *LS-360* and *LS-500*. *LS-860* is the union of *LS-360* and *LS-500*. We use standard validation (*dev-clean* and *dev-other*) and test (*test-clean* and *test-other*) sets. We also use **Common Voice (CV), v13.0 (English, German and French)** data [62]: the train, validation and test sets are provided in the dataset. In addition, we split the training data using a specific percentage of users to train a seed model only and the rest of users for FL training: e.g., we create *CV-en-train-10(-5)* by selecting all the data for a randomly chosen 10% (5%) of the users from *CV-en-train* and we denote the remaining data by *CV-en-train-90(-95)*. Statistics on speakers are given in Figure 1: it shows that CV data are much more heterogeneous than LS as highlighted by [20]. CV data thus enable a more realistic scenario for testing FL and FL with DP. The most realistic scenario for FL uses a small central dataset to train a seed model (e.g. *LS-100*), and a larger dataset from a different distribution for FL (e.g. *CV-en-train*).[3]

**Central Training** We use standard feature extraction for audio [25, 27] by computing log-mel filterbanks with 80 coefficients with a 25ms sliding window and 10ms stride length, later normalized to zero mean and unit variance for each input sequence. We employ a vanilla encoder-based transformer model trained with the CTC loss [40].

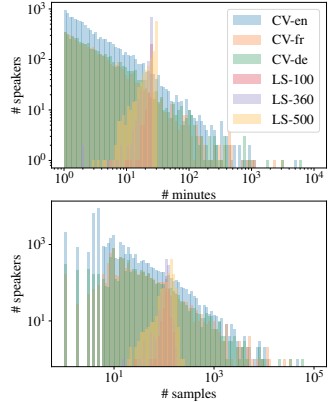

We start our experimentation with the state-of-the-art model on *LS-100* from [63] with 255M parameters. We use SpecAugment [64] and clip all gradients during training to have a norm of at most 1 (see Appendix F and G.6 for a discussion). We found it difficult (see Appendix G.3) to switch to FL from central training when post-LayerNorm was used (similar issues were reported by [31]). Following [31] we thus do central training with pre-LayerNorm (also used in FL), LARS [65], and relatively high (0.5) learning rate (LR) without any warmup and with stepwise decay to simplify the recipe and have stable training while maintaining the performance.

**Federated Training** We simulate FL by considering every speaker and its data as a separate user. In most experiments, SGD [66] with constant LR is used as the local optimizer and LAMB [65] is used as the central optimizer. We found this combination most robust (see Appendix G.4). The central LR is constant with further exponential decay unless noted otherwise, gradient clipping is set to 1 for each client. Unless noted otherwise, we restrict the number of central steps to 2k. Although most

Figure 1: Train distribution in LS and CV: per speaker #minutes (top) and #samples (bottom).

simulations would further improve after 2k steps, the per-step latency and DP noise addition typically

---

[3]Datasets such as LibriSpeech, CommonVoice, VCTK, TED-LIUM offer speaker metadata necessary for creating heterogeneous FL clients. Datasets like People's Speech, GigaSpeech, SPGISpeech lack speaker metadata. We choose LibriSpeech and CommonVoice as they offer the greatest speaker diversity.

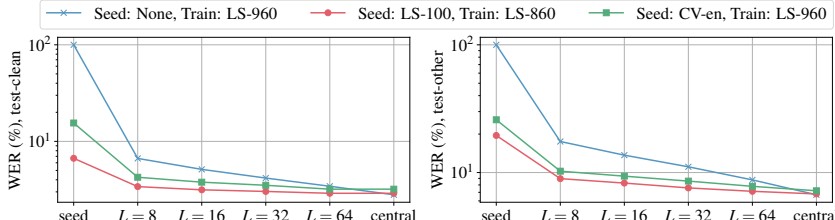

Figure 2: Impact of the cohort size $S$ and seed models on FL models trained on LS. We use exponential decay for central LR starting at $t = 1,000$, decay rate 0.6, and transition steps 500 (w/o seed model) or 250 (w/ seed model) with $T = 2k$ total central steps and 10 local epochs. Local (central) LR is 0.4 (0.006) (w/o seed model) or 0.2 (0.003) (w/ seed model). See details in Appendix G.2, Table 3.

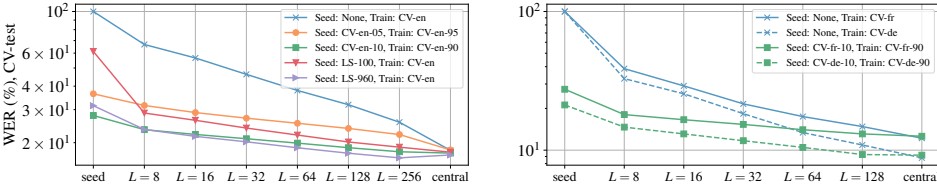

Figure 3: Impact of the cohort size $S$ and seed models on FL models trained on CV: English (left) and French/German (right). We use exponential decay for central LR starting at $t = 1,000$ (w/o seed model) or 750 (w/ seed model), decay rate 0.6, and transition steps 500 (w/o seed model) or 750 (w/ seed model) with $T = 2k$ total central steps and 10 local epochs. Local (central) LR is 0.4 (0.006) (w/o seed model) or 0.2 (0.002) (w/ seed model). See details in Appendix G.2, Tables 4 and 10.

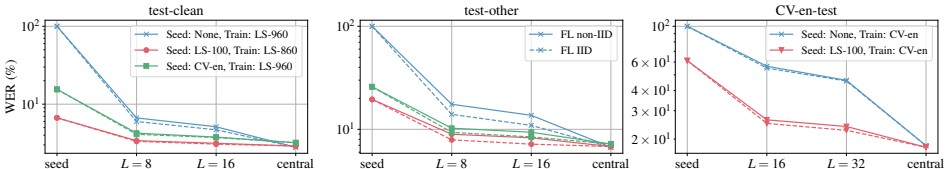

Figure 4: Impact of randomizing the distribution of data across users for LS (left, middle) and CV (right) measured by WER. Parameter settings are described in Figure 2 for LS and Figure 3 for CV. While the original training data are non-IID (solid), IID (dashed) versions of *LS-960*, *LS-860* and *CV-en-train* are created by choosing a user id uniformly and randomly from the set of user ids for each data point in the corresponding dataset. Detailed numbers are in Appendix G.2, Tables 5 and 6.

limit the number of iterations in practical private FL systems to this range [67, 56]. To keep simple and robust training recipes, we do not do extensive hyper-parameters search. After finding the best configuration on one training setup we apply the same hyper-parameters to the rest of experiments.

## 4.1 Impact of Seed Models and Cohort Size

In Figures 2 and 3 we show that initializing FL with seed models instead of randomly significantly decrease word error rate (WER) for both LS and CV (all languages), even with domain shift for the seed model training (e.g, using LS seed model for CV and vice-versa). Using seed model initialization for FL, we can almost close the gap between central and FL trainings within 2k central steps and moderate cohort sizes: $\geq 64$ ($\geq 128$) for LS (CV). Larger cohorts consistently improve the outcomes within 2k central steps – increasing the cohort size directly increases the amount of seen data. Even without seed models, FL is competitive with central models given a large enough cohort size.

Increasing the amount of data for seed model training improves the trained FL models regardless of whether the data come from the same domain or not (e.g. compare *CV-en-train-05* seed vs. *CV-en-train-10* seed or *LS-100* seed vs. *LS-960* seed on *CV-en-train* in Figure 3). In fact, the use of seed models trained on considerably more data from another domain can outperform the use of seed models trained on less data from the same domain: the results on *CV-en-train* with a *LS-960* seed model are better than the results with a *CV-en-train-10* seed model on *CV-en-train-90* (see

Table 1: Results for FL with DP and a model pre-trained on *LS-100* ($\sim$100h) used as central data and afterwards fine-tuned with FL on *CV-en-train* ($\sim$1.6k hours) used as clients data. We report added noise $\mathcal{N}(0, IC^2\sigma_{\mathrm{DP}}^2 qK)$ per client ($\omega_k = \frac{1}{K}$) and CV dev and test WERs (%) for two clipping variants with clipping $C$: global and per-layer "uniform" ("dim"). The total number of users is $K$, the cohort size is $S = qK$, and the number of central steps is $T$. We set $\delta = 10^{-9}$ following [46] and report $\varepsilon$ for which $(\varepsilon, \delta)$-DP holds for given $S$ and $K$ using the moments accountant of [8]. For scaling $S$ and $K$ where it is practically intractable to run model training (marked "-"), we extrapolate $(\varepsilon, \delta)$-DP following [46] and, assuming the training dynamic remains unchanged, thus similar WER could be obtained. Central training gives 14.7%/17.8% WER on dev/test. Extended results are given in Appendix H and in Table 17. $\varepsilon$ should be below 10 to be practically useful (marked with blue).

| $z$ | $\sigma_{\mathrm{DP}}$ $(\cdot 10^{-6})$ | $C$ | $S$ | $K$ | $q = S/K$ | $T$ | $\varepsilon$ | Renyi order | global clipping dev WER | test WER | per-layer clipping: uniform (dim) dev WER | test WER |
|---|---|---|---|---|---|---|---|---|---|---|---|---|
| - | - | - | 0 | 34,753 | 0 | 0 | 0 | - | 54.7 | 61.2 | 54.7 | 61.2 |
| 0.03072 | 30.0 | 0.01 | 1,024 | 34,753 | 0.0295 | 2,006 | $1.1{\cdot}10^6$ | 1.1 | - | - | 25.2 (24.2) | 29.3 (28.2) |
| 0.3072 | 30.0 | 0.01 | 10,240 | 347,530 | 0.0295 | 2,006 | $3.7{\cdot}10^2$ | 1.1 | - | - | - | - |
| 1.536 | 30.0 | 0.01 | 51,200 | 1,737,650 | 0.0295 | 2,006 | $6.5{\cdot}10^0$ | 7.0 | - | - | - | - |
| 0.02048 | 20.0 | 0.01 | 1,024 | 34,753 | 0.0295 | 2,006 | $2.6{\cdot}10^6$ | 1.1 | - | - | 23.7 (22.6) | 27.6 (26.5) |
| 1.024 | 20.0 | 0.01 | 51,200 | 1,737,650 | 0.0295 | 2,006 | $1.3{\cdot}10^0$ | 4.0 | - | - | - | - |
| 2.048 | 20.0 | 0.01 | 102,400 | 3,475,300 | 0.0295 | 2,006 | $4.5{\cdot}10^0$ | 9.0 | - | - | - | - |
| 0.01024 | 10.0 | 0.01 | 1,024 | 34,753 | 0.0295 | 2,006 | $1.1{\cdot}10^7$ | 1.1 | 30.7 | 35.2 | **21.3 (20.1)** | **25.0 (23.7)** |
| 0.512 | 10.0 | 0.01 | 51,200 | 1,737,650 | 0.0295 | 2,006 | $7.2{\cdot}10^1$ | 1.5 | - | - | - | - |
| 1.024 | 10.0 | 0.01 | 102,400 | 3,475,300 | 0.0295 | 2,006 | $1.3{\cdot}10^1$ | 4.0 | - | - | - | - |
| 2.048 | 10.0 | 0.01 | 204,800 | 6,950,600 | 0.0295 | 2,006 | $4.5{\cdot}10^0$ | 9.0 | - | - | - | - |
| 0.003072 | 3.0 | 0.01 | 1,024 | 34,753 | 0.0295 | 2,006 | $1.2{\cdot}10^8$ | 1.1 | 27.0 | 31.1 | **17.9 (17.1)** | **21.2 (20.4)** |
| 0.3072 | 3.0 | 0.01 | 102,400 | 3,475,300 | 0.0295 | 2,006 | $3.7{\cdot}10^2$ | 1.1 | - | - | - | - |
| 0.6144 | 3.0 | 0.01 | 204,800 | 6,950,600 | 0.0295 | 2,006 | $4.2{\cdot}10^1$ | 2.0 | - | - | - | - |
| 0.6144 | 3.0 | 0.01 | 204,800 | 69,506,000 | 0.00295 | 2,034 | $7.2{\cdot}10^0$ | 3.0 | - | - | - | - |
| 0.6144 | 3.0 | 0.01 | 204,800 | 695,060,000 | 0.000295 | 3,390 | $3.7{\cdot}10^0$ | 6.0 | - | - | - | - |
| 0.001024 | 1.0 | 0.01 | 1,024 | 34,753 | 0.0295 | 2,006 | $1.1{\cdot}10^9$ | 1.1 | 22.9 | 26.7 | 16.2 (16.0) | 19.5 (19.3) |
| 0.2048 | 1.0 | 0.01 | 204,800 | 6,950,600 | 0.0295 | 2,006 | $1.1{\cdot}10^3$ | 1.1 | - | - | - | - |
| 0.2048 | 1.0 | 0.01 | 204,800 | 69,506,000 | 0.00295 | 2,034 | $2.7{\cdot}10^2$ | 1.1 | - | - | - | - |
| 0.2048 | 1.0 | 0.01 | 204,800 | 695,060,000 | 0.000295 | 3,390 | $9.4{\cdot}10^1$ | 1.3 | - | - | - | - |
| - | 0 | 0.01 | 1,024 | 34,753 | 0.0295 | 2,000 | inf | - | 15.7 | 18.9 | 15.9 | 19.1 |
| - | 0 | 1.0 | 1,024 | 34,753 | 0.0295 | 2,000 | inf | - | 15.7 | 18.9 | 15.7 | 18.9 |

more ablations in Appendix G.9, Table 15). The gap between FL models with different seed models decreases as the cohort size increases – the latter directly increases seen data in FL training.

To demonstrate robustness of found hyper-parameters and observed results in Figure 3 (left), we applied the **exact same training configuration** to train FL models on CV French and German data. *We confirm in Figure 3 (right) that the training configuration found on English data is robust: similar trends and results hold for French and German.*

## 4.2 Impact of Data Heterogeneity

Prior works argued that data heterogeneity poses a challenge for FL [11, 12]. Figure 4 shows that distributing data uniformly and randomly across users indeed improves performance for all settings. Since for LS, every client's data are of similar duration and we use dynamic batching, this is unlikely to be due to the differences in the amount of data between clients. The impact of using i.i.d. data decreases with increasing cohort size. Figure 4 suggests that algorithms such as FedProx [11], ProxSkip [68], and SCAFFOLD [57] could further improve FL performance. We evaluated FedProx, which marginally improved FL performance in some cases (see Appendix G.7, Table 13).

## 4.3 Federated Learning with Differential Privacy

For FL with DP we consider a setting close to the real-world scenario: *LS-100* is used as central data to train a seed model (without DP); *CV-en-train* is considered as clients' data on which the seed model is trained afterwards using FL. In this setting (i) the clients' data are $\sim$16 times bigger than the server data and (ii) there is a domain shift in clients' data.

As discussed in Section 2, DP is challenging for larger models due to their size. To make the model training more resistant to noise, we need to increase the cohort size, e.g. in recent work [69] used 150k cohort size for FL with DP. We take exactly the same setup as in Figure 3 with the data *CV-en-train* and the seed model trained on *LS-100*. First we scale the FL training to the cohort size of 1024; to mitigate the resulting increase in the computational cost of the training, we switch from

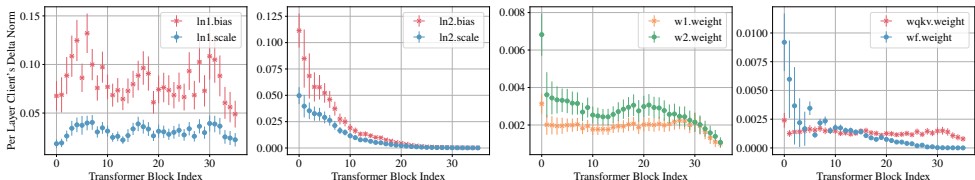

Figure 5: Client delta norms computed per layer in the model. We average statistics across all clients and central steps, and plot the mean and standard deviation. The model is trained with $\sigma_{\mathrm{DP}} = 3 \cdot 10^{-6}$ and global clients' deltas clipping $C = 10^{-2}$ (Algorithm 1). Transformer block consists of attention parameters (wqkv and wf) with LayerNorm (ln1), and MLP (w1 and w2) with LayerNorm (ln2).

10 local epochs to 10 local steps (see Appendix H.2, all other hyper-parameters stay the same). As we discuss in Appendix H.2, we expect that more local compute that would be feasible in a real deployment, should lead to better results than what we get in our experiments. Increasing the cohort size further closes the gap with the central baseline. Second, we use and vary the clipping $C$ applied to clients' deltas without adding DP noise yet. Although the average norm of clients' deltas is 0.7 (see Appendix H, Figure 7), they can be clipped with $C$ as low as $C = 10^{-8}$ without any impact on model's quality. This is consistent with Corollary 1: the interaction of trust ratio $R_h$ with $C_h$ re-normalizes the gradients. Further we set $C = 10^{-2}$ to prevent numerical precision errors. Finally, we add different levels of noise $\sigma_{\mathrm{DP}}$ to every client's delta before averaging the deltas across clients.

In Table 1, we estimate $(\varepsilon, \delta)$-DP by the moments accountant of [8] for every level of noise, number of clients $K$, clients sampling $q$, clients' deltas clipping $C$, and number of central training steps $T$, where $\omega_k = \frac{1}{K}$. Using FL with DP, we can improve over the poor performing *LS-100* seed model due to limited server data and their domain shift: WER is reduced from 61.2% to 31.1% with $\sigma_{\mathrm{DP}} = 3 \cdot 10^{-6}$ and $(7.2, 10^{-9})$-DP assuming the training effectiveness (WER) remains the same if, following [46], we extrapolate to ~70M clients with the cohort size of ~200k[4]. Lowering the DP noise $\sigma_{\mathrm{DP}}$ decreases model's WER, but DP guarantees become impractical even if we scale $K$ and $S$.

In Figure 5, we analyse the clients' deltas by computing model's per-layer deltas norm. We highlight that the norms are imbalanced across different transformer layers and also across different types of parameters: (i) first transformer layers have a larger deltas norm magnitude; and (ii) delta norms for attention parameters are an order of magnitude lower than those for LayerNorms. This observed imbalance motivates the application of per-layer intervention, as formally discussed in Section 3.

*To avoid $\sigma_{\mathrm{DP}}$ dominating the attention layers and slowing down the convergence, following Theorem 2,* we apply per-layer clipping (Definition 3) which significantly improves model convergence (see Figure 12 in Appendix): with the same $\sigma_{\mathrm{DP}} = 3 \cdot 10^{-6}$ we are able to closely match the model trained without DP noise ($\sigma_{\mathrm{DP}} = 0$) with only a small WER degradation (from 19.1% to 21.2% WER) while guaranteeing $(7.2, 10^{-9})$-DP assuming the training effectiveness remains the same if, following [46], we extrapolate to ~70M clients with the cohort size of ~200k. Moreover, we can now increase DP noise up to $\sigma_{\mathrm{DP}} = 10^{-5}$ getting 23.7% WER with $(4.5, 10^{-9})$-DP by following [46] and extrapolating only to ~7M clients with the cohort size of ~200k (see Table 1). The latter is a realistic scenario even for mid/low resource languages. We can further reduce WER by ~1% for the same $(\varepsilon, \delta)$-DP guarantee if we apply per-layer clipping based on the layer dimension (see Table 1).

## 5 Related Works

**FL for ASR** was first studied by [70] using attention-based Seq2Seq LSTM models. The paper showed that FL in ASR suffers from data heterogeneity, a known problem in FL [71, 33]. They proposed gradient weighting to speed up convergence and improve performance. Building on this, [22] used hybrid LSTM models and introduced client adaptive normalization to mitigate data heterogeneity. Similarly, [17] used RNN from [72] and added noise to local gradients to address data heterogeneity. However, these FL-trained ASR models significantly underperformed their centralized counterparts.

**End-to-End ASR models in FL** [19] used a ~120M parameters conformer [27] model together with federated dropout to train only a subset of parameters on each client. This reduced com-

---

[4][69, 67] showed it is realistic to (i) have millions of users to participate in FL and (ii) use a large cohort size of 150k in FL deployments.

munication and improved FL performance relative to central training. However, the setup used 10k-100k central steps and homogeneous data distribution, which is impractical in real-world scenarios. [20] used Seq2Seq model with a CNN encoder and RNN decoder trained with joint CTC-attention objective. They noted that training E2E ASR model from scratch in a realistic FL setup is *"nearly impossible"*, and proposed an additional training step on held-out server data, after model aggregation. They also emphasized switching from LS data to CV due to its more realistic data distribution. Recently, [73] trained a $\sim$130M parameter model using weighted client aggregation and word frequency histograms, initialized from a centrally pretrained model. [56] showed FL training with similarly sized conformer models using adaptive optimizers from scratch. We borrow several real-world settings from prior works: (i) limiting to *2k central steps* [56], (ii) training large transformer models from scratch [56], and (iii) using both CV [62] and LS [61] datasets for experiments [19, 56] to evaluate robustness across datasets and languages. Unlike prior work, we also study: (i) FL with DP for ASR and (ii) impact of domain mismatch between the data used for central pretraining and FL.

**Data Leakage in FL for ASR.** [21] improves ASR performance using large ($\sim$300M parameters) pre-trained self-supervised model (transformer) to initialize FL and observe speaker information leakage via model updates. Audio can further leak sensitive attributes such as gender and health conditions [74]. Given that FL alone does not guarantee user privacy [2, 4] and several recent works [32, 21] have explored privacy attacks targeting FL in ASR, it is very important to enable FL training with DP. To this end, our work addresses this critical gap by enabling FL with DP for ASR.

**Adaptive Clipping and Convergence Bounds** Adaptive clipping was first proposed in [8], but the authors reported no observable impact on convergence. Recently, [75] proposed adaptive clipping using privately estimated quartile statistics, incurring a negligible privacy budget. They noted a dependence on non-private data and fixed learning rate (LR), which can be prohibitive in practice. [76] later provided a comprehensive convergence analysis in a central setup, showing that LR depends on the clipping constant. [55] is one of the few works providing convergence bound under clipping using FedAvg [1]. However, it cannot be trivially extended to per-layer clipping or adaptive optimizers. Additionally, [77] is a contemporary work that proposes adaptive layer-wise clipping for DP-SGD by distributing clipping budget over the layers proportional to the layer-wise gradient statistics gathered on a public dataset. While this method can uncover more fine-grained gradient distribution over layers, it introduces a reliance on representative public dataset. In contrast, our work adopts a different perspective: rather than conditioning clipping on public dataset, we redistribute the clipping budget structurally (uniform or dimension-aware) and rely on the LAMB optimizer to dynamically regulate inter-layer heterogeneity. Thus, while [77] depends on *static, public-data informed* sensitivity distribution, our analysis and experiments highlight the importance of *dynamic, optimizer-driven* adaptivity. To the best of our knowledge, we present the first explicit convergence bound for FL with DP that incorporates per-layer clipping, LAMB optimizer, and DP noise – highlighting the interdependence among trust ratio in LAMB, per-layer clipping constant and DP noise in FL.

**Divergence Accumulation** Recently, [23, 24] showed that deeper models in FL suffer from *"divergence accumulation"* – accumulation of dissimilarities among client models during back-propagation.

# 6 Conclusion

ASR provides a valuable and realistic benchmark for (private) federated learning (FL), offering large datasets that are naturally partitioned by speakers and exhibit heterogeneity typical in real-world settings. With the exception of language modeling, benchmarks commonly used in works studying FL with DP lack these characteristics, limiting their practicality. In this work, we focused on real-world constraints such as the task of adapting a model trained centrally on LibriSpeech to Common Voice data via FL, a benchmark for both FL and FL with DP that captures core FL challenges: domain shift, user-level heterogeneity, and privacy constraints at scale. We demonstrate that with a *practical* number of central aggregations, it is possible to train large transformer models that perform competitively in the federated settings – both from scratch or when starting from an out-of-domain seed model. We highlight that enabling FL with DP for ASR is non-trivial and requires solutions that manage the interaction between privacy, clipping, and model size. To this end, we revived per-layer clipping and used layer-wise adaptive optimization, thus achieving user-level $(7.2, 10^{-9})$-DP (resp. $(4.5, 10^{-9})$-DP) with only a 1.3% (resp. 4.6%) absolute drop in the WER, when extrapolating to high (resp. low) population scale. These results establish a practical and scalable foundation for privacy-preserving FL training with DP for large models beyond ASR.

## Acknowledgments

We thank Samy Bengio, David Grangier, Filip Granqvist, Navdeep Jaitly and Vojta Jina for essential general discussion on the paper throughout all stages; Pierre Ablin and Dan Busbridge for discussion on scaling laws; Audra McMillan and Congzheng Song for discussion on differential privacy; Shuangfei Zhai for discussion on transformer stability and behavior of gradient norms; Ronan Collobert, Navdeep Jaitly, Audra McMillan and Barry Theobald for the helpful feedback on the initial drafts of the work; Dan Busbridge for detailed feedback and helpful suggestion to improve the paper; Satyen Kale for checking asymptotic of theoretical bounds and helpful feedback on prior theoretical work; Hassan Babaie, Cindy Liu, Rajat Phull, and the wider Apple infrastructure team for assistance with developing scalable, fault tolerant code. Names are in alphabetical order by last name within the group.

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

# Appendices

## A   Ethics Statement

For all experiments we use publicly available data for research: LibriSpeech (CC BY 4.0) and Common Voice v13.0 (CC BY-SA 3.0). In the paper, we aim to understand the behavior of large transformer models in federated learning (FL) with differential privacy. This is a step towards developing private FL in the context of speech recognition to provide strong guarantees of user privacy.

## B   Reproducibility Statement

For all experiments we use publicly available datasets for research: LibriSpeech (CC BY 4.0) and Common Voice v13.0 (CC BY-SA 3.0). Data processing is described in the main body of the paper. We describe all configurations, training details, ablations, and our procedure of selecting hyper-parameters throughout the paper and in Appendix. We also provide important discussions on different aspects of the empirical results as well as detailed plots of various characteristics tracked during training in the Appendix. The code is open sourced and available at `https://github.com/apple/ml-pfl4asr`.

## C   Societal Impact

This work explores research in the intersection of privacy, optimization, federated learning, and speech recognition. Given the widespread adoption of ASR models deployed in production environments ranging from virtual assistants to accessibility applications, enabling privacy-preserving training of ASR models using differential privacy has the potential to benefit the end users, particularly in sensitive domains such as healthcare and biometrics. This work contributes towards the responsible development of ASR models by overcoming a long-standing obstacle to applying DP to deep architectures. However, the deployment of FL with DP does not eliminate all privacy risks. Real-world deployments must ensure additional measures including secure aggregation and careful consideration of population-scale that influence the strength of the privacy introduced by DP in this work.

## D   Discussion

### D.1   Need for Private Federated Learning

In Section 1 we discussed that FL on its own does not guarantee user privacy. For example, [2] showed that the gradients sent to the server can be used to reconstruct the original training images and text. [3] showed that a model can memorize specific pieces of data that can be reconstructed using only the model itself. In the context of ASR, [32] developed two attacks that aim to infer speaker identity from the model updates without access to the actual users' audio data. [74] showed that audio data reveal information about the content but they can also be used to derive other pieces of sensitive information including biometric identity, physical traits, geographical origin, emotions, level of intoxication, age, gender and health.

These and many other works emphasize the necessity of developing private FL with strong guarantees on the user privacy. In this paper, we focus on providing first insights for private FL with DP for ASR.

### D.2   Why Do We Study Larger Models for FL and DP?

As discussed in Section 1, we focus on the model size of 250M parameters. Prior works in FL with DP primarily focused on studying models of up to 30M parameters, justifying the use of smaller models by communication and training costs associated with the model size and the difficulty of training reasonable models with DP because the impact of noise scales with the model size. However, [78, 79]

showed that it is possible to (centrally) fine-tune large language models with hundreds of millions of parameters with DP and DP impact does not prevent efficient training if gradients are low rank.

Our main reason to focus our study on larger models for both FL and DP is the observation that larger models are simpler to train in practice. It is a hard and open problem to efficiently train small models that perform the same or better than models obtained for example by distillation of large models into smaller models [39]. To disentangle the ability to train small models efficiently from the problem of matching central training with FL and FL with DP, we study larger models. Our results give a hint that the gap that existed between FL and central models could be related to the absence of proper training recipes for smaller models.

One could argue that current model sizes are huge in the era of large language models, and different techniques, like LoRA [80], could be used to reduce training time on clients as well as communication costs. This was done for example by [48] who used partial and low-rank model updates to train large language models with private FL. However, we believe that first we need to train competitive baseline models from scratch or from out-of-domain seed models, and understand their behaviour and limits.

### D.3 Clipping and Adaptive Optimizers

[55] investigated how clipping fights data heterogeneity in FL. As discussed in Section 2, clipping is also an essential part of DP. To be able to train transformer models, we must use clipping too, and thus the recipes used for transformers are aligned with FL with DP. In Appendix H Figure 7, we show that gradient clipping during local training leads to bounded norms of user deltas where the latter is necessary for DP. Without applying gradient clipping, the gradient norms would be huge already at the beginning of the training and even with LARS, pre-LayerNorm and central training we would not be able to train a reasonable model. Thus, it is extremely hard to disentangle any empirical results for transformers to understand how clipping helps the training for FL with DP.

[42] and [56] showed that adaptive optimizers alleviate the issue of data heterogeneity for FL. At the same time it is hard to train transformer models without adaptive optimizers [81, 55]. This is yet another example of alignment between FL and central training of transformer models; a technique that helps alleviate data heterogeneity in FL is a must when training large transformer models even centrally.

### D.4 Fusion of ASR Model with a Language Model

To further improve WERs, ASR models can be combined with language models during inference. This can be done in various ways, e.g. using beam-search decoding for CTC models [25, 82], or using shallow fusion [83], cold fusion [84], deep fusion [85], and simple fusion [86] for Seq2Seq or transducer-based models. In this paper, we leave the study on how a language model integration affects the final model performance as a future work. In the latter case, language models can also be trained using FL with DP [46, 48, 67].

### D.5 Conformer vs Transformer

Purposefully, we do not use the conformer architecture [27] in the paper. In prior work by [28], it was shown that, e.g., for CTC models both conformer and transformer architectures give similar results while conformer has fewer parameters. We focus on larger models to understand their behaviour. Moreover, vanilla transformers are still de facto a standard in other domains, while conformers were adopted only in speech recognition. Therefore, focusing on vanilla transformer models will broaden the impact of our findings for speech recognition on the FL and DP communities at large.

### D.6 Seed Models

[20] trained seed models to initialize FL using a small fraction of speakers (117 speakers, or 2.8%, for French and 99 speakers, or 13.2%, for Italian) and used the rest of the data for FL training. Recent work [87] showed that model quality depends on the number of speakers and the diversity of the training data: it is better to have more speakers with shorter total audio duration than to have fewer speakers with longer total audio duration.

Based on the recommendation of [87] to have at least 1k speakers in the training data, we randomly sampled $5\%$ (English) or $10\%$ (all languages) of speakers for the in-domain seed model training. This provided more than 1k users for training CV seed models for English. While for French the seed model is trained from only 685 users and for German the seed model is trained from only 712 users, we note that French and German languages are easier to train. Furthermore, for FL models training on CV (English) we use a seed model trained on *LS-100* that has only 251 speakers; however, *LS-100* has over 100 hours of audio, which is approximately $6.3\%$ of the total audio in CV.

Preliminary experiments showed that the seed model training on a subset of $5\%$ speakers with the shortest total audio does not converge: even for English the subset contains less than 2 hours of audio, which is known to be hard for any E2E ASR model training. In contrast, if we take a subset of $5\%$ speakers with the longest total audio as in [20], a seed model is very well trained as then the dataset has more than $64\%$ of total audio in the CV dataset for English language and training on the rest of the data brings little benefit. Thus, we found the subsets with minimum-duration or maximum-duration users to not be practical scenarios.

For LS, validation (test) set has 5h of audio with mean of $\sim$8min and standard deviation of 0.1min for the total duration per speaker. For CV, validation (test) set has $\sim$30h with mean of $\sim$15s and standard deviation of 1.5s for the total duration per speaker. Thus validation and test datasets have homogeneous distribution which weights speakers (users) equally for evaluation. For both LS and CV we use original validation and test sets, without any modification. Thus, the disjoint set of speakers in different splits and the disjoint set of speakers in a seed model and FL training ensure that speakers (clients) are not accounted twice in the privacy budget.

### D.7 Limitations

Our theoretical results are derived under some assumptions listed in Section 3. Empirical results are limited to i) LibriSpeech and CommonVoice (en, de, fr) read speech data; ii) monolingual models; iii) CTC-based models of size 100M-500M parameters; iv) absence of external language models; v) audio data assumed to be labeled. Future work would include theoretical and empirical analysis to overcome these limitations.

## E Theoretical Analysis

### E.1 Assumptions

Given a global model comprising of $H$ layers, the model parameters are defined as $\boldsymbol{\theta} = (\boldsymbol{\theta}_1, \cdots, \boldsymbol{\theta}_h, \cdots \boldsymbol{\theta}_H)$. It is presumed that the loss function for each sample $\boldsymbol{x}$ is bounded below: $\min_{\boldsymbol{\theta} \in \mathbb{R}^D} \ell(\boldsymbol{x}, \boldsymbol{\theta}) > -\infty$, where $\boldsymbol{x} \sim \mathcal{D}_k, \forall k$. Let $\|\cdot\|$ denote the $l_2$-norm. Our analysis uses the following standard assumptions [12, 42, 50, 51, 52, 53, 54]:

1. *Smoothness of Gradient of Loss Function:* Gradient of loss function is layer-wise $L_h$-smooth for $\forall h$ [49]:
$$\|\nabla_h \ell(\boldsymbol{x}, \boldsymbol{\theta}) - \nabla_h \ell(\boldsymbol{x}, \boldsymbol{\theta}')\| \leq L_h \|\boldsymbol{\theta} - \boldsymbol{\theta}'\|, \ \ \forall \boldsymbol{\theta}, \boldsymbol{\theta}' \in \mathbb{R}^D, \ \boldsymbol{x} \in \mathbb{R}^N, \forall k, \qquad \textbf{(A1.1)}$$
   where $\nabla_h$ denotes gradient with respect to parameters $\boldsymbol{\theta}_h$ of layer $h$. Consequently, the loss function is also $L$-smooth, where $L = \|(L_1, \cdots, L_H)\|_2$:
$$\|\nabla \ell(\boldsymbol{x}, \boldsymbol{\theta}) - \nabla \ell(\boldsymbol{x}, \boldsymbol{\theta}')\| \leq L \|\boldsymbol{\theta} - \boldsymbol{\theta}'\|, \ \ \forall \boldsymbol{\theta}, \boldsymbol{\theta}' \in \mathbb{R}^D, \boldsymbol{x} \in \mathbb{R}^N, \forall k. \qquad \textbf{(A1.2)}$$

2. *Local Gradient Characteristics:* Given user $k$, $\mathscr{B}_k = \{\boldsymbol{x}_i\}_{i=1}^B, \boldsymbol{x}_i \sim \mathcal{D}_k$ and local gradient $\nabla \ell(\boldsymbol{x}, \boldsymbol{\theta})$ for $\boldsymbol{x} \sim \mathcal{D}_k$, its estimator $\mathbf{g}_k(\boldsymbol{\theta}) = \mathbf{g}_k(\mathscr{B}_k, \boldsymbol{\theta})$ (e.g. obtained by SGD) is an unbiased estimator and have a bounded variance [12, 50, 54], thus:
$$\mathbb{E}_{\mathscr{B}_k}\left[\mathbf{g}_k(\boldsymbol{\theta})\right] = \nabla \mathscr{L}_k(\boldsymbol{\theta}), \text{ and} \qquad \textbf{(A2.1)}$$
$$\mathbb{E}_{\mathscr{B}_k}\left[\|\mathbf{g}_k(\boldsymbol{\theta}) - \nabla \mathscr{L}_k(\boldsymbol{\theta})\|^2\right] \leq \sigma_{\text{loc}}^2, \ \sigma_{\text{loc}}^2 \geq 0, \ \ \forall \boldsymbol{\theta} \in \mathbb{R}^D, \ \ \forall k. \qquad \textbf{(A2.2)}$$

3. *Global Pseudo-Gradient Characteristics:* The variance of global (pseudo-) gradient is assumed to be bounded [52, 42] such that:
$$\sum_{k=1}^K \omega_k \|\nabla \mathscr{L}_k(\boldsymbol{\theta}) - \nabla \mathscr{L}(\boldsymbol{\theta})\|^2 \leq \sigma_{\text{glob}}^2, \ \sigma_{\text{glob}} \geq 0, \ \ \forall \boldsymbol{\theta} \in \mathbb{R}^D. \qquad \textbf{(A3)}$$

To give a probabilistic interpretation of this assumption we can estimate the global loss gradient $\nabla\mathscr{L}(\boldsymbol{\theta})$ by sampling one user $u \sim \text{Categorical}(\omega_1,\ldots,\omega_K)$ and using the following unbiased estimator $\nabla\hat{\mathscr{L}}(u,\boldsymbol{\theta}) = \nabla\mathscr{L}_u(\boldsymbol{\theta})$. Then,

$$\mathbb{E}_{u\sim\text{Categorical}(\omega_1,\ldots,\omega_K)}\left[\nabla\hat{\mathscr{L}}(u,\boldsymbol{\theta})\right] = \sum_{k=1}^{K}\mathbb{P}[u=k]\nabla\mathscr{L}_k(\boldsymbol{\theta}) = \sum_{k=1}^{K}\omega_k\nabla\mathscr{L}_k(\boldsymbol{\theta}) = \nabla\mathscr{L}(\boldsymbol{\theta}).$$

From the latter we get the variance of the estimator and extend it to the left-hand side of Equation A3. Thus, we can interpret this assumption as the variance of global (pseudo-) gradient.

## E.2 DP Assumptions

To incorporate user-level DP into FL, we consider that every client is sampled i.i.d. with probability $q$ ($qK = S$) and then the client updates $\boldsymbol{\Delta}_k^{(t)}$ are: (i) clipped such that their $l_2$ norm is bounded, i.e., $\|\boldsymbol{\Delta}_k^{(t)}\|_2 \leq C$ at every central training step $t$ and then (ii) perturbed via Gaussian mechanism, such that final client updates under FL with DP are given by $\boldsymbol{\Delta}_k^{(t)} + \mathcal{N}\left(0, \boldsymbol{I}C^2\sigma_{\text{DP}}^2\frac{q}{\sum_{i=1}^{K}\omega_i^2}\right)$, where $\boldsymbol{\Delta}_k^{(t)} = \eta_{\text{loc}}\alpha_k^{(t)}\mathbf{G}_k^{(t)}$ and $\alpha_k^{(t)} = \frac{C}{\max\left(C,\|\eta_{\text{loc}}\mathbf{G}_k^{(t)}\|\right)}$. For $\sum_{k=1}^{K}\omega_k = 1$, where $\omega_k \in (0,1)$, we can extend Theorem 1 to the weighted loss case by defining sensitivity $\mathbb{S} = \max_{k=1}^{K}\omega_k/q$ per Lemma 1 from [46]. Having $\omega_k = 1/K$, we get exactly sensitivity definition $\mathbb{S} = 1/(qK)$ from Theorem 1.

## E.3 Helpful Lemmas

**Lemma 1.** *For any positive variables $C, X, Y \in \mathbb{R}^+$, we have*

$$\frac{1}{\max(C,X)} - \frac{1}{\max(C,Y)} \leq \frac{|X-Y|}{C^2} \tag{2}$$

*Proof.* We can prove it by analyzing three independent cases:

(i) if $C \geq X$ and $C \geq Y$ we trivially have

$$\frac{1}{\max(C,X)} - \frac{1}{\max(C,Y)} = \frac{1}{C} - \frac{1}{C} = 0 \leq \frac{|X-Y|}{C^2},$$

(ii) if $C < X$ and $C < Y$ we have

$$\frac{1}{\max(C,X)} - \frac{1}{\max(C,Y)} = \frac{1}{X} - \frac{1}{Y} \leq \frac{|X-Y|}{XY} \leq \frac{|X-Y|}{C^2}, \text{ and}$$

(iii) if $Y < C < X$ (equivalently the case $Y > C > X$) we have

$$\frac{1}{\max(C,X)} - \frac{1}{\max(C,Y)} = \frac{1}{X} - \frac{1}{C} \leq \frac{|X-C|}{XC} \leq \frac{|X-Y|}{C^2}.$$

Thus, we can conclude $\forall\, C, X, Y \in \mathbb{R}^+$

$$\frac{1}{\max(C,X)} - \frac{1}{\max(C,Y)} \leq \frac{|X-Y|}{C^2}.$$

∎

**Lemma 2.** *For $X \in \mathbb{R}$ and a constant $C > 0$,*

$$(X-C)_+ \leq \frac{X^2}{2C} \tag{3}$$

*where $(X-C)_+ = \max(0, X-C)$.*

*Proof.* For $X \leq C$, $(X - C)_+ \leq 0$, and inequality holds trivially. For $X > C$, we can use the algebraic identity:

$$X^2 \geq 2C(X - C) \tag{4}$$

which can be rewritten as $(X - C)_+ \leq \frac{X^2}{2C}$. ∎

**Lemma 3.** *For a random vector* $\mathbf{G} \in \mathbb{R}^d$ *with bounded norm* $\|\mathbf{G}\| \leq U$ *and a clipping constant* $C > 0$, *define the clipped vector as* $\mathbf{G}_C \in \mathbb{R}^d$ *such that* $\mathbf{G}_C = \mathbf{G} \cdot \frac{C}{\max(C, \mathbb{E}[\|\mathbf{G}\|])}$. *Then the squared distance between* $\mathbf{G}$ *and* $\mathbf{G}_C$ *is upper bounded by*

$$\|\mathbf{G} - \mathbf{G}_C\|^2 \leq \frac{U^4}{4C^2}. \tag{5}$$

*Proof.* For $\mathbb{E}[\|\mathbf{G}\|] \leq C$, we have

$$\|\mathbf{G} - \mathbf{G}_C\|^2 = \left\| \mathbf{G} - \mathbf{G} \cdot \frac{C}{\max(C, \mathbb{E}[\|\mathbf{G}\|])} \right\|^2 = \left\| \mathbf{G} - \mathbf{G} \cdot \frac{C}{C} \right\|^2 = 0. \tag{6}$$

For $\mathbb{E}[\|\mathbf{G}\|] > C$, we can use the algebraic identity:

$$\|\mathbf{G} - \mathbf{G}_C\|^2 = \left\| \mathbf{G} - \mathbf{G} \cdot \frac{C}{\max(C, \mathbb{E}[\|\mathbf{G}\|])} \right\|^2 = \left\| \mathbf{G} - \mathbf{G} \cdot \frac{C}{\mathbb{E}[\|\mathbf{G}\|]} \right\|^2$$

$$= \left( 1 - \frac{C}{\mathbb{E}[\|\mathbf{G}\|]} \right)^2 \cdot \|\mathbf{G}\|^2 = \left( \frac{\mathbb{E}[\|\mathbf{G}\|] - C}{\mathbb{E}[\|\mathbf{G}\|]} \right)^2 \cdot \|\mathbf{G}\|^2$$

$$\overset{\text{Lemma 2}}{\leq} \frac{(\mathbb{E}[\|\mathbf{G}\|])^4}{4C^2} \frac{\|\mathbf{G}\|^2}{(\mathbb{E}[\|\mathbf{G}\|])^2} \leq \frac{U^4}{4C^2}. \tag{7}$$

Thus, the trivial case in Equation 6 together with the inequality in Equation 7 results in the final bound. ∎

### E.4 LAMB

The per-layer update rule of LAMB is given by:

$$\boldsymbol{\theta}_h^{(t+1)} \leftarrow \boldsymbol{\theta}_h^{(t)} - \eta_{\text{glob}} \frac{\phi\left(\|\boldsymbol{\theta}_h^{(t)}\|\right)}{\|\mathbf{u}_h^{(t)} + \lambda\boldsymbol{\theta}_h^{(t)}\|} \left( \mathbf{u}_h^{(t)} + \lambda\boldsymbol{\theta}_h^{(t)} \right) \text{ where } \lambda \geq 0, \left[\mathbf{u}_h^{(t)}\right]_i = \frac{\left[\mathbf{m}_h^{(t)}\right]_i}{\left[\sqrt{\mathbf{v}_h^{(t)}} + \xi\right]_i}, \tag{8}$$

$$\mathbf{m}_h^{(t)} = \beta_1 \mathbf{m}_h^{(t-1)} + (1 - \beta_1)\boldsymbol{\Delta}_h^{(t)}, \mathbf{v}_h^{(t)} = \beta_2 \mathbf{v}_h^{(t-1)} + (1 - \beta_2)\left[\boldsymbol{\Delta}_h^{(t)}\right]^2, \ 0 \leq \beta_1, \beta_2 \leq 1. \tag{9}$$

$\phi : \mathbb{R} \to \mathbb{R}$ is a scaling function which is often defined as an identity in standard LAMB applications [49, 88]. While $\xi$ is a constant generally employed for numerical stability, [56] show that $\xi = 0.01$ leads to best results in FL, likely because it counteracts spurious pseudo-gradients early in the training. Let's define the trust ratio of LAMB:

$$r_h^{(t)} \triangleq \frac{\phi\left(\|\boldsymbol{\theta}_h^{(t)}\|\right)}{\|\mathbf{u}_h^{(t)}\|} \in \mathbb{R} \qquad \text{and} \qquad \left[\mathbf{p}_h^{(t)}\right]_i \triangleq \frac{r_h^{(t)}}{\left[\sqrt{\mathbf{v}_h^{(t)}} + \xi\right]_i}. \tag{10}$$

### E.5 Adaptive Optimizers and Per-Layer Clipping: The Main Proof

**Theorem 2.** *Assume* **A1.1**, **A2.1**, **A2.2**, *and* **A3**, $\eta_{\text{glob}} L < 1$ *and* $\kappa = \left[ 1 - 8(1 - \eta_{\text{loc}} T_{\text{loc}})^2 \right] > 0$. *If the trust ratio from Eq. 10 in* LAMB *optimizer is controlled in the Algorithm 1 (global optimizer is* LAMB *and local optimizer is SGD) such that* $r_h^{(t)} \leq R_h$ *and* $\left\| \mathbf{1} - \mathbf{p}_h^{(t)} \right\|_\infty \leq P_h$, $\beta_1 = 0$ *and* $\lambda = 0$

*in* LAMB *optimizer, and clients are i.i.d. sampled with probability* $q = 1$ *(no sampling), then after* $T$ *steps of aggregation the performance of FL with DP, per-layer clipping and layer-wise gradient normalization is characterized by the following upper bound:*

$$
\frac{\kappa}{T} \sum_{t=0}^{T-1} \mathbb{E}_{t_{\text{loc}}} \left[ \left\| \nabla \mathscr{L} \left( \boldsymbol{\theta}^{(t)} \right) \right\|^2 \right] \leq \frac{2 \left[ \mathscr{L} \left( \boldsymbol{\theta}^{(0)} \right) - \mathscr{L} \left( \boldsymbol{\theta}^{\star} \right) \right]}{\eta_{\text{glob}} T} + 16 H \eta_{\text{loc}}^2 T_{\text{loc}}^2 \sigma_{\text{glob}}^2 + 32 H \eta_{\text{loc}}^2 T_{\text{loc}}^2 \sigma_{\text{loc}}^2
$$

$$
+ \frac{C^2 \sigma_{\text{DP}}^2}{\xi^2} \sum_{h=1}^{H} R_h^2 d_h + 2 \eta_{\text{loc}}^2 T_{\text{loc}}^2 \sum_{h=1}^{H} M_h^2 \left[ 16 L^2 \eta_{\text{loc}}^2 T_{\text{loc}}^2 + \frac{1}{C_h^2} + 4 P_h^2 \right]
$$

$$
+ 4 \frac{\eta_{\text{loc}}^2 T_{\text{loc}}^2}{\xi^2 T} \sum_{h=1}^{H} \frac{R_h^2 M_h^2}{C_h^2} \sum_{t=0}^{T-1} \left[ \mathbb{E}_k \left[ \mathsf{Var}_{t_{\text{loc}}} \left( \left\| \mathbf{G}_{h,k}^{(t)} \right\| \right) \right] + \mathsf{Var}_k \left( \mathbb{E}_{t_{\text{loc}}} \left[ \left\| \mathbf{G}_{h,k}^{(t)} \right\| \right] \right) \right].
$$

*where* $k \sim \text{Categorical}(\omega_1, \ldots, \omega_K)$ *and* $\mathbb{E}_{t_{\text{loc}}} [\cdot]$ *denotes the expectation over sampled mini-batch* $\mathscr{B}_k^{(t_{\text{loc}})}$ *every local step* $t_{\text{loc}} = 1, \ldots, T_{\text{loc}}$ *from the client data:* $\boldsymbol{x}_k^{(t,t_{\text{loc}})} \sim \mathcal{D}_k, \boldsymbol{x}_k^{(t,t_{\text{loc}})} \in \mathscr{B}_k^{(t_{\text{loc}})}, |\mathscr{B}_k^{(t_{\text{loc}})}| = B_k.$

*Proof.* We assume $\beta_1 = 0$ and regularization $\lambda = 0$. Then the update rule for LAMB as the global optimizer at the FL server given by Equation 8 can be rewritten:

$$
\mathbf{v}_h^{(t)} = \beta_2 \mathbf{v}_h^{(t-1)} + (1 - \beta_2) \left[ \boldsymbol{\Delta}_h^{(t)} \right]^2, \ 0 \leq \beta_2 \leq 1, \tag{11}
$$

$$
\left[ \mathbf{u}_h^{(t)} \right]_i = \frac{\left[ \boldsymbol{\Delta}_h^{(t)} \right]_i}{\left[ \sqrt{\mathbf{v}_h^{(t)}} + \xi \right]_i}, \tag{12}
$$

$$
\boldsymbol{\theta}_h^{(t+1)} \leftarrow \boldsymbol{\theta}_h^{(t)} - \eta_{\text{glob}} \frac{\phi \left( \| \boldsymbol{\theta}_h^{(t)} \| \right)}{\| \mathbf{u}_h^{(t)} \|} \mathbf{u}_h^{(t)}, \forall h. \tag{13}
$$

Given definition of the trust ratio in Equation 10, the update rule can be expressed as:

$$
\boldsymbol{\theta}_h^{(t+1)} \leftarrow \boldsymbol{\theta}_h^{(t)} - \eta_{\text{glob}} \mathbf{p}_h^{(t)} \odot \boldsymbol{\Delta}_h^{(t)}. \tag{14}
$$

The aggregated clients updates, or pseudo-gradient, $\boldsymbol{\Delta}_h^{(t)}$ are given by (as $q = 1$):

$$
\boldsymbol{\Delta}_h^{(t)} = \sum_{k=1}^{K} \omega_k \left( \boldsymbol{\Delta}_{h,k}^{(t)} + \mathbf{z}_{h,k}^{(t)} \right), \tag{15}
$$

where $\boldsymbol{\Delta}_{h,k}^{(t)}$ is the accumulated client update (see Algorithm 1) and $\mathbf{z}_{h,k}^{(t)} \sim \mathcal{N} \left( 0, \mathbf{I}_h C^2 \sigma_{\text{DP}}^2 \frac{q}{\sum_{i=1}^{K} \omega_i^2} \right)$ is the random independent DP-noise added to client updates. For each client we perform $T_{\text{loc}}$ steps of SGD optimization by i) sampling a mini-batch $\mathscr{B}_k^{(t_{\text{loc}})}$ every local step $t_{\text{loc}} = 1, \ldots, T_{\text{loc}}$ from the client data: $\boldsymbol{x}_k^{(t,t_{\text{loc}})} \sim \mathcal{D}_k, \boldsymbol{x}_k^{(t,t_{\text{loc}})} \in \mathscr{B}_k^{(t_{\text{loc}})}, |\mathscr{B}_k^{(t_{\text{loc}})}| = B_k$; ii) performing a gradient step with a local step-size (learning rate) $\eta_{\text{loc}} > 0$ having $\boldsymbol{\theta}^{(t,0)} = \boldsymbol{\theta}^{(t)}$:

$$
\mathbf{g}_{h,k}^{(t,t_{\text{loc}})} \left( \boldsymbol{\theta}_{h,k}^{(t,t_{\text{loc}})} \right) = \frac{1}{B_k} \sum_{\boldsymbol{x}_k^{(t,t_{\text{loc}})} \in \mathscr{B}_k^{(t_{\text{loc}})}} \nabla \ell_h(\boldsymbol{x}_k^{(t,t_{\text{loc}})}, \boldsymbol{\theta}_{h,k}^{(t,t_{\text{loc}})}), \tag{16}
$$

$$
\boldsymbol{\theta}_{h,k}^{(t,t_{\text{loc}})} = \boldsymbol{\theta}_{h,k}^{(t,t_{\text{loc}}-1)} - \eta_{\text{loc}} \mathbf{g}_{h,k}^{(t,t_{\text{loc}}-1)} \left( \boldsymbol{\theta}_{h,k}^{(t,t_{\text{loc}}-1)} \right), \tag{17}
$$

where $\mathbf{g}_{h,k}^{(t,t_{\text{loc}})}$ are unbiased estimators of clients' gradients. Then for a given per-layer clipping constant $C_h > 0$, the client updates and the corresponding clipping multipliers are defined as:

$$
\mathbf{G}_{h,k}^{(t)} = \boldsymbol{\theta}^{(t,0)} - \boldsymbol{\theta}^{(t,T_{\text{loc}})} = \eta_{\text{loc}} \sum_{t_{\text{loc}}=0}^{T_{\text{loc}}-1} \mathbf{g}_{h,k}^{(t,t_{\text{loc}})} \tag{18}
$$

$$\mathbf{\Delta}_{h,k}^{(t)} = \alpha_{h,k}^{(t)} \, \mathbf{G}_{h,k}^{(t)} \qquad \text{with } \alpha_{h,k}^{(t)} = \frac{C_h}{\max\left(C_h, \left\|\mathbf{G}_{h,k}^{(t)}\right\|\right)}. \tag{19}$$

With triangle inequality we can upper bound the norm of a random variable $\mathbf{G}_{h,k}^{(t)}$ given theorem assumption that $\nabla \ell_h(\boldsymbol{x}, \boldsymbol{\theta})$ is $L$-Lipschitz smooth and thus $\|\nabla \ell_h(\boldsymbol{x}, \boldsymbol{\theta})\| \leq M_h$ (e.g. $M_h = \|\nabla \ell_h(\boldsymbol{x}_0, \boldsymbol{\theta}_0)\| + L \max_{\boldsymbol{\theta} \in \Theta} \|\boldsymbol{\theta}\|$, where $\Theta$ is a compact):

$$\|\mathbf{G}_{h,k}^{(t)}\| \leq \eta_{\mathrm{loc}} \sum_{t_{\mathrm{loc}}=0}^{T_{\mathrm{loc}}-1} \|\mathbf{g}_{h,k}^{(t,t_{\mathrm{loc}})}\| \leq \eta_{\mathrm{loc}} T_{\mathrm{loc}} M_h. \tag{20}$$

We next define the auxiliary terms in the context of clipping:

$$\widetilde{\mathbf{\Delta}}_{h,k}^{(t)} = \widetilde{\alpha}_{h,k}^{(t)} \mathbf{G}_{h,k}^{(t)} \qquad \text{with } \widetilde{\alpha}_{h,k}^{(t)} = \frac{C_h}{\max\left(C_h, \mathbb{E}_{t_{\mathrm{loc}}}\left[\left\|\mathbf{G}_{h,k}^{(t)}\right\|\right]\right)}, \tag{21}$$

$$\overline{\mathbf{\Delta}}_{h,k}^{(t)} = \overline{\alpha}_{h}^{(t)} \mathbf{G}_{h,k}^{(t)} \qquad \text{with } \overline{\alpha}_{h}^{(t)} = \frac{C_h}{\max\left(C_h, \mathbb{E}_{t_{\mathrm{loc}},k}\left[\left\|\mathbf{G}_{h,k}^{(t)}\right\|\right]\right)}, \tag{22}$$

Since gradient of loss function $\ell(\boldsymbol{x}, \theta)$ is $L-$Lipschitz smooth, we get the following for any two points $\boldsymbol{\theta}^{(t+1)}$ and $\boldsymbol{\theta}^{(t)}$:

$$\ell(\boldsymbol{x}, \boldsymbol{\theta}^{(t+1)}) \leq \ell(\boldsymbol{x}, \boldsymbol{\theta}^{(t)}) + \left\langle \nabla \ell(\boldsymbol{x}, \boldsymbol{\theta}^{(t)}), \boldsymbol{\theta}^{(t+1)} - \boldsymbol{\theta}^{(t)} \right\rangle + \frac{L}{2} \left\|\boldsymbol{\theta}^{(t+1)} - \boldsymbol{\theta}^{(t)}\right\|^2. \tag{23}$$

By taking expectation over the client $k$ data $\boldsymbol{x} \sim \mathcal{D}_k$, for every client we can write down:

$$\mathscr{L}_k(\boldsymbol{x}, \boldsymbol{\theta}^{(t+1)}) \leq \mathscr{L}_k(\boldsymbol{x}, \boldsymbol{\theta}^{(t)}) + \left\langle \nabla \mathscr{L}_k(\boldsymbol{x}, \boldsymbol{\theta}^{(t)}), \boldsymbol{\theta}^{(t+1)} - \boldsymbol{\theta}^{(t)} \right\rangle + \frac{L}{2} \left\|\boldsymbol{\theta}^{(t+1)} - \boldsymbol{\theta}^{(t)}\right\|^2. \tag{24}$$

By multiplying with $\omega_k$, summing all inequalities across clients, and using the update rule from Equation 14, we can get:

$$\mathscr{L}\left(\boldsymbol{\theta}^{(t+1)}\right) \leq \mathscr{L}\left(\boldsymbol{\theta}^{(t)}\right) + \left\langle \nabla \mathscr{L}\left(\boldsymbol{\theta}^{(t)}\right), \boldsymbol{\theta}^{(t+1)} - \boldsymbol{\theta}^{(t)} \right\rangle + \frac{L}{2} \left\|\boldsymbol{\theta}^{(t+1)} - \boldsymbol{\theta}^{(t)}\right\|^2 \tag{25}$$

$$= \mathscr{L}\left(\boldsymbol{\theta}^{(t)}\right) - \eta_{\mathrm{glob}} \left\langle \nabla \mathscr{L}\left(\boldsymbol{\theta}^{(t)}\right), \mathbf{p}^{(t)} \odot \mathbf{\Delta}^{(t)} \right\rangle + \frac{\eta_{\mathrm{glob}}^2 L}{2} \left\|\mathbf{p}^{(t)} \odot \mathbf{\Delta}^{(t)}\right\|^2. \tag{26}$$

**Bounding loss** $\mathbb{E}_{t_{\mathrm{loc}}}\left[\mathscr{L}\left(\boldsymbol{\theta}^{(t+1)}\right)\right]$ **with** $\mathbf{Z}_h$ **term**

Now, let's take the expectation over the mini-batches $\mathscr{B}_k^{(t_{\mathrm{loc}})}$ sampling in the local SGD optimization for both sides of inequality having random variables $\mathbf{p}_h^{(t)}$ and $\boldsymbol{\Delta}_h^{(t)}$ (for short notation we use $\mathbb{E}_{t_{\mathrm{loc}}}\left[\cdot\right]$):

$$
\mathbb{E}_{t_{\mathrm{loc}}}\left[\mathscr{L}\left(\boldsymbol{\theta}^{(t+1)}\right)\right] \le \mathscr{L}\left(\boldsymbol{\theta}^{(t)}\right) - \eta_{\mathrm{glob}} \sum_{h=1}^{H} \mathbb{E}_{t_{\mathrm{loc}}}\left[\left\langle \nabla\mathscr{L}_h\left(\boldsymbol{\theta}_h^{(t)}\right), \mathbf{p}_h^{(t)} \odot \boldsymbol{\Delta}_h^{(t)}\right\rangle\right]
$$

$$
+ \frac{\eta_{\mathrm{glob}}^2 L}{2} \sum_{h=1}^{H} \mathbb{E}_{t_{\mathrm{loc}}}\left[\left\|\mathbf{p}_h^{(t)} \odot \boldsymbol{\Delta}_h^{(t)}\right\|^2\right]
$$

$$
\overset{(i)}{=} \mathscr{L}\left(\boldsymbol{\theta}^{(t)}\right) - \frac{\eta_{\mathrm{glob}}}{2} \sum_{h=1}^{H} \mathbb{E}_{t_{\mathrm{loc}}}\left[\left\|\nabla\mathscr{L}_h\left(\boldsymbol{\theta}_h^{(t)}\right)\right\|^2\right] - \frac{\eta_{\mathrm{glob}}}{2} \sum_{h=1}^{H} \mathbb{E}_{t_{\mathrm{loc}}}\left[\left\|\mathbf{p}_h^{(t)} \odot \boldsymbol{\Delta}_h^{(t)}\right\|^2\right]
$$

$$
+ \frac{\eta_{\mathrm{glob}}}{2} \sum_{h=1}^{H} \underbrace{\mathbb{E}_{t_{\mathrm{loc}}}\left[\left\|\nabla\mathscr{L}_h\left(\boldsymbol{\theta}_h^{(t)}\right) - \mathbf{p}_h^{(t)} \odot \boldsymbol{\Delta}_h^{(t)}\right\|^2\right]}_{\mathbf{Z}_h}
$$

$$
+ \frac{\eta_{\mathrm{glob}}^2 L}{2} \sum_{h=1}^{H} \mathbb{E}_{t_{\mathrm{loc}}}\left[\left\|\mathbf{p}_h^{(t)} \odot \boldsymbol{\Delta}_h^{(t)}\right\|^2\right]
$$

$$
\le \mathscr{L}\left(\boldsymbol{\theta}^{(t)}\right) - \frac{\eta_{\mathrm{glob}}}{2}\left\|\nabla\mathscr{L}\left(\boldsymbol{\theta}^{(t)}\right)\right\|^2 - \frac{\eta_{\mathrm{glob}}\left(1 - \eta_{\mathrm{glob}} L\right)}{2} \sum_{h=1}^{H} \mathbb{E}_{t_{\mathrm{loc}}}\left[\left\|\mathbf{p}_h^{(t)} \odot \boldsymbol{\Delta}_h^{(t)}\right\|^2\right] + \frac{\eta_{\mathrm{glob}}}{2} \sum_{h=1}^{H} \mathbf{Z}_h
$$

$$
\overset{(ii)}{\le} \mathscr{L}\left(\boldsymbol{\theta}^{(t)}\right) - \frac{\eta_{\mathrm{glob}}}{2}\left\|\nabla\mathscr{L}\left(\boldsymbol{\theta}^{(t)}\right)\right\|^2 + \frac{\eta_{\mathrm{glob}}}{2} \sum_{h=1}^{H} \underbrace{\mathbb{E}_{t_{\mathrm{loc}}}\left[\left\|\nabla\mathscr{L}_h\left(\boldsymbol{\theta}_h^{(t)}\right) - \mathbf{p}_h^{(t)} \odot \boldsymbol{\Delta}_h^{(t)}\right\|^2\right]}_{\mathbf{Z}_h}.
$$

$$(27)$$

where $(i)$ uses $-2\left\langle a, b\right\rangle = -\|a\|^2 - \|b\|^2 + \|a-b\|^2$ and $(ii)$ uses the condition $\eta_{\mathrm{glob}} L < 1$. We can next bound $\mathbf{Z}_h$ using Equation 15, the auxiliary terms $\widetilde{\boldsymbol{\Delta}}_{h,k}^{(t)}$ and $\overline{\boldsymbol{\Delta}}_{h,k}^{(t)}$ defined in Equations 21-22,

$$
\mathbf{Z}_h = \mathbb{E}_{t_{\mathrm{loc}}}\left[\left\|\nabla\mathscr{L}_h\left(\boldsymbol{\theta}_h^{(t)}\right)\right.\right.
$$

$$
- \sum_{k=1}^{K} \omega_k \mathbf{G}_{h,k}^{(t)} + \sum_{k=1}^{K} \omega_k \mathbf{G}_{h,k}^{(t)}
$$

$$
- \sum_{k=1}^{K} \omega_k \mathbf{p}_h^{(t)} \odot \boldsymbol{\Delta}_{h,k}^{(t)} - \sum_{k=1}^{K} \omega_k \mathbf{p}_h^{(t)} \odot \mathbf{z}_{h,k}^{(t)}
$$

$$
+ \sum_{k=1}^{K} \omega_k \mathbf{p}_h^{(t)} \odot \widetilde{\boldsymbol{\Delta}}_{h,k}^{(t)} - \sum_{k=1}^{K} \omega_k \mathbf{p}_h^{(t)} \odot \widetilde{\boldsymbol{\Delta}}_{h,k}^{(t)}
$$

$$
\left.\left.+ \sum_{k=1}^{K} \omega_k \mathbf{p}_h^{(t)} \odot \overline{\boldsymbol{\Delta}}_{h,k}^{(t)} - \sum_{k=1}^{K} \omega_k \mathbf{p}_h^{(t)} \odot \overline{\boldsymbol{\Delta}}_{h,k}^{(t)}\right\|^2\right].
$$

$$(28)$$

As a reminder, Jensen's inequality for some $\boldsymbol{y}_i \in \mathbb{R}^D$ gives us:

$$
\left\|\sum_{k=1}^{K} \omega_i \boldsymbol{y}_k\right\|^2 \le \sum_{i=1}^{K} \omega_i \|\boldsymbol{y}_k\|^2 \text{ where } \sum_{k=1}^{K} \omega_k = 1, \ 0 \le \omega_k \le 1.
$$

$$(29)$$

**Helpful inequalities**

Using the triangle inequality first and then applying Hölder's inequality, we get for $\boldsymbol{y}_i \in \mathbb{R}^D$

$$\left\|\sum_{k=1}^{K} \boldsymbol{y}_k\right\|^2 \leq \left(\sum_{k=1}^{K} \|\boldsymbol{y}_k\|\right)^2 = \left(\sum_{k=1}^{K}(\|\boldsymbol{y}_k\| \cdot 1)\right)^2 \leq K \sum_{k=1}^{K} \|\boldsymbol{y}_k\|^2. \tag{30}$$

Also, if $\boldsymbol{y}_1$ and $\boldsymbol{y}_2$ are independent random variables and $\mathbb{E}\left[\boldsymbol{y}_1\right] = 0$ then:

$$\begin{aligned}
\mathbb{E}\left[\|\boldsymbol{y}_1 + \boldsymbol{y}_2\|^2\right] &= \mathbb{E}\left[\|\boldsymbol{y}_1\|^2 + \|\boldsymbol{y}_2\|^2 + 2 < \boldsymbol{y}_1, \boldsymbol{y}_2 >\right] \\
&= \mathbb{E}\left[\|\boldsymbol{y}_1\|^2\right] + \mathbb{E}\left[\|\boldsymbol{y}_2\|^2\right] + 2 < \mathbb{E}\left[\boldsymbol{y}_1\right], \mathbb{E}\left[\boldsymbol{y}_2\right] > \\
&= \mathbb{E}\left[\|\boldsymbol{y}_1\|^2\right] + \mathbb{E}\left[\|\boldsymbol{y}_2\|^2\right].
\end{aligned} \tag{31}$$

Let's estimate for any random variable $\mathbf{y}_h$ the following entity having that $\mathbf{y}_h$ and $\mathbf{p}_h^{(t)}$ are not independent variables:

$$\mathbb{E}_{t_{\text{loc}}}\left[\left\|\mathbf{p}_h^{(t)} \odot \mathbf{y}_h\right\|^2\right] = \sum_{i=1}^{d_h} \mathbb{E}_{t_{\text{loc}}}\left[\left[\mathbf{p}_h^{(t)}\right]_i^2 [\mathbf{y}_h]_i^2\right] \leq \frac{R_h^2}{\xi^2} \sum_{i=1}^{d_h} \mathbb{E}_{t_{\text{loc}}}[\mathbf{y}_h]_i^2 = \frac{R_h^2}{\xi^2} \mathbb{E}_{t_{\text{loc}}}\left[\|\mathbf{y}_h\|^2\right]. \tag{32}$$

**Bounding term with DP noise in $\mathbf{Z}_h$**

Having upper bound on the expectation $\mathbb{E}_{t_{\text{loc}}}\left[\mathbf{p}_h^{(t)}\right]_i^2 \leq \frac{R_h^2}{\xi^2}$ and random independent DP noise $\mathbf{z}_{h,k}^{(t)} \sim \mathcal{N}\left(0, \mathbf{I}_h C^2 \sigma_{\text{DP}}^2 \frac{1}{\sum_{i=1}^{K} \omega_i^2}\right)$ as $q = 1$ per theorem condition (thus $\mathbf{p}_h^{(t)}$ and $\mathbf{z}_{h,k}^{(t)}$ are independent variables), let's get the upper bound first for:

$$\begin{aligned}
\mathbb{E}_{t_{\text{loc}}}\left[\left\|\sum_{k=1}^{K} \omega_k \mathbf{p}_h^{(t)} \odot \mathbf{z}_{h,k}^{(t)}\right\|^2\right] &= \mathbb{E}_{t_{\text{loc}}}\left[\left\|\mathbf{p}_h^{(t)} \odot \sum_{k=1}^{K} \omega_k \mathbf{z}_{h,k}^{(t)}\right\|^2\right] = \sum_{i=1}^{d_h} \mathbb{E}_{t_{\text{loc}}}\left[\left[\mathbf{p}_h^{(t)}\right]_i^2 \left[\sum_{k=1}^{K} \omega_k \mathbf{z}_{h,k}^{(t)}\right]_i^2\right] \\
&= \sum_{i=1}^{d_h} \mathbb{E}_{t_{\text{loc}}}\left[\mathbf{p}_h^{(t)}\right]_i^2 \mathbb{E}_{t_{\text{loc}}}\left[\sum_{k=1}^{K} \omega_k \mathbf{z}_{h,k}^{(t)}\right]_i^2 \leq \frac{R_h^2}{\xi^2} d_h \mathbb{E}_{t_{\text{loc}}}\left[\sum_{k=1}^{K} \omega_k \mathbf{z}_{h,k}^{(t)}\right]_0^2 \\
&= \frac{R_h^2}{\xi^2} d_h C^2 \sigma_{\text{DP}}^2 \frac{1}{\sum_{k=1}^{K} \omega_k^2} \sum_{k=1}^{K} \omega_k^2 = \frac{R_h^2}{\xi^2} d_h C^2 \sigma_{\text{DP}}^2. \tag{33}
\end{aligned}$$

**Bounding $\mathbf{Z}_h$ with $\mathbf{Y}_1, \mathbf{Y}_2, \mathbf{Y}_3, \mathbf{Y}_4$ terms**

Given Equations 29, 33, 30 and 31 (we use the fact that DP noise $\mathbf{z}_{h,k}^{(t)}$ is zero-mean independent variable), we can bound $\mathbf{Z}_h$ in the following way:

$$\begin{aligned}
\mathbf{Z}_h \leq 4\,\mathbb{E}_{t_{\text{loc}}}\underbrace{\left[\sum_{k=1}^{K} \omega_k \left\|\nabla \mathscr{L}_h\left(\boldsymbol{\theta}_h^{(t)}\right) - \mathbf{G}_{h,k}^{(t)}\right\|^2\right]}_{\mathbf{Y}_1} &+ 4\sum_{k=1}^{K} \omega_k \mathbb{E}_{t_{\text{loc}}}\underbrace{\left[\left\|\mathbf{G}_{h,k}^{(t)} - \mathbf{p}_h^{(t)} \odot \overline{\boldsymbol{\Delta}}_{h,k}^{(t)}\right\|^2\right]}_{\mathbf{Y}_2} \\
+ 4\sum_{k=1}^{K} \omega_k \mathbb{E}_{t_{\text{loc}}}\underbrace{\left[\left\|\mathbf{p}_h^{(t)} \odot \left(\boldsymbol{\Delta}_{h,k}^{(t)} - \widetilde{\boldsymbol{\Delta}}_{h,k}^{(t)}\right)\right\|^2\right]}_{\mathbf{Y}_3} &+ 4\sum_{k=1}^{K} \omega_k \mathbb{E}_{t_{\text{loc}}}\underbrace{\left[\left\|\mathbf{p}_h^{(t)} \odot \left(\widetilde{\boldsymbol{\Delta}}_{h,k}^{(t)} - \overline{\boldsymbol{\Delta}}_{h,k}^{(t)}\right)\right\|^2\right]}_{\mathbf{Y}_4} \\
+ \frac{R_h^2}{\xi^2} d_h C^2 \sigma_{\text{DP}}^2. &\tag{34}
\end{aligned}$$

**Bounding $\mathbf{Y}_1$ term**

Defining $\mathbf{H}_{h,k}^{(t)} = \frac{1}{T_{\text{loc}}} \sum_{t_{\text{loc}}=0}^{T_{\text{loc}}-1} \mathbf{g}_{h,k}^{(t,t_{\text{loc}})}$ and $\mathbf{G}_{h,k}^{(t)} = \eta_{\text{loc}} T_{\text{loc}} \mathbf{H}_{h,k}^{(t)}$:

$$
\mathbf{Y}_1 = \mathbb{E}_{t_{\text{loc}}} \left[ \sum_{k=1}^{K} \omega_k \left\| \nabla \mathscr{L}_h \left( \boldsymbol{\theta}_h^{(t)} \right) - \mathbf{G}_{h,k}^{(t)} \right\|^2 \right] \overset{\text{Eq. 29}}{\le} \sum_{k=1}^{K} \omega_k \mathbb{E}_{t_{\text{loc}}} \left[ \left\| \nabla \mathscr{L}_h \left( \boldsymbol{\theta}_h^{(t)} \right) - \eta_{\text{loc}} T_{\text{loc}} \mathbf{H}_{h,k}^{(t)} \right\|^2 \right]
$$

$$
= \sum_{k=1}^{K} \omega_k \mathbb{E}_{t_{\text{loc}}} \left[ \left\| \nabla \mathscr{L}_h \left( \boldsymbol{\theta}_h^{(t)} \right) - \eta_{\text{loc}} T_{\text{loc}} \nabla \mathscr{L}_h \left( \boldsymbol{\theta}_h^{(t)} \right) + \eta_{\text{loc}} T_{\text{loc}} \nabla \mathscr{L}_h \left( \boldsymbol{\theta}_h^{(t)} \right) - \eta_{\text{loc}} T_{\text{loc}} \mathbf{H}_{h,k}^{(t)} \right\|^2 \right]
$$

$$
\overset{\text{Eq. 30}}{\le} 2 \sum_{k=1}^{K} \omega_k \mathbb{E}_{t_{\text{loc}}} \left[ \left\| \nabla \mathscr{L}_h \left( \boldsymbol{\theta}_h^{(t)} \right) - \eta_{\text{loc}} T_{\text{loc}} \nabla \mathscr{L}_h \left( \boldsymbol{\theta}_h^{(t)} \right) \right\|^2 \right]
$$

$$
+ 2 \sum_{k=1}^{K} \omega_k \mathbb{E}_{t_{\text{loc}}} \left[ \left\| \eta_{\text{loc}} T_{\text{loc}} \nabla \mathscr{L}_h \left( \boldsymbol{\theta}_h^{(t)} \right) - \eta_{\text{loc}} T_{\text{loc}} \mathbf{H}_{h,k}^{(t)} \right\|^2 \right]
$$

$$
= 2(1 - \eta_{\text{loc}} T_{\text{loc}})^2 \mathbb{E}_{t_{\text{loc}}} \left[ \left\| \nabla \mathscr{L}_h \left( \boldsymbol{\theta}_h^{(t)} \right) \right\|^2 \right] + 2 \eta_{\text{loc}}^2 T_{\text{loc}}^2 \underbrace{\sum_{k=1}^{K} \omega_k \mathbb{E}_{t_{\text{loc}}} \left[ \left\| \nabla \mathscr{L}_h \left( \boldsymbol{\theta}_h^{(t)} \right) - \mathbf{H}_{h,k}^{(t)} \right\|^2 \right]}_{\mathbf{X}},
$$

$$\tag{35}$$

where

$$
\mathbf{X} = \sum_{k=1}^{K} \omega_k \mathbb{E}_{t_{\text{loc}}} \left[ \left\| \nabla \mathscr{L}_h \left( \boldsymbol{\theta}_h^{(t)} \right) - \mathbf{H}_{h,k}^{(t)} \right\|^2 \right]
$$

$$
= \sum_{k=1}^{K} \omega_k \mathbb{E}_{t_{\text{loc}}} \left[ \left\| \nabla \mathscr{L}_h \left( \boldsymbol{\theta}_h^{(t)} \right) - \nabla \mathscr{L}_{h,k} \left( \boldsymbol{\theta}_h^{(t)} \right) + \nabla \mathscr{L}_{h,k} \left( \boldsymbol{\theta}_h^{(t)} \right) - \mathbf{H}_{h,k}^{(t)} \right\|^2 \right]
$$

$$
\overset{\text{Eq. 30}}{\le} 2 \sum_{k=1}^{K} \omega_k \mathbb{E}_{t_{\text{loc}}} \left[ \left\| \nabla \mathscr{L}_h \left( \boldsymbol{\theta}_h^{(t)} \right) - \nabla \mathscr{L}_{h,k} \left( \boldsymbol{\theta}_h^{(t)} \right) \right\|^2 \right] + 2 \sum_{k=1}^{K} \omega_k \mathbb{E}_{t_{\text{loc}}} \left[ \left\| \nabla \mathscr{L}_{h,k} \left( \boldsymbol{\theta}_h^{(t)} \right) - \mathbf{H}_{h,k}^{(t)} \right\|^2 \right]
$$

$$
\overset{\text{A3}}{\le} 2\sigma_{\text{glob}}^2 + 2 \sum_{k=1}^{K} \omega_k \mathbb{E}_{t_{\text{loc}}} \left[ \left\| \nabla \mathscr{L}_{h,k} \left( \boldsymbol{\theta}_h^{(t)} \right) - \frac{1}{T_{\text{loc}}} \sum_{t_{\text{loc}}=0}^{T_{\text{loc}}-1} \nabla \mathscr{L}_{h,k} \left( \boldsymbol{\theta}_{h,k}^{(t,t_{\text{loc}})} \right) + \frac{1}{T_{\text{loc}}} \sum_{t_{\text{loc}}=0}^{T_{\text{loc}}-1} \nabla \mathscr{L}_{h,k} \left( \boldsymbol{\theta}_{h,k}^{(t,t_{\text{loc}})} \right) - \mathbf{H}_{h,k}^{(t)} \right\|^2 \right]
$$

$$
\overset{\text{Eq. 30}}{\le} 2\sigma_{\text{glob}}^2 + 4 \sum_{k=1}^{K} \omega_k \mathbb{E}_{t_{\text{loc}}} \left[ \left\| \nabla \mathscr{L}_{h,k} \left( \boldsymbol{\theta}_h^{(t)} \right) - \frac{1}{T_{\text{loc}}} \sum_{t_{\text{loc}}=0}^{T_{\text{loc}}-1} \nabla \mathscr{L}_{h,k} \left( \boldsymbol{\theta}_{h,k}^{(t,t_{\text{loc}})} \right) \right\|^2 \right]
$$

$$
+ 4 \sum_{k=1}^{K} \omega_k \mathbb{E}_{t_{\text{loc}}} \left[ \left\| \frac{1}{T_{\text{loc}}} \sum_{t_{\text{loc}}=0}^{T_{\text{loc}}-1} \nabla \mathscr{L}_{h,k} \left( \boldsymbol{\theta}_{h,k}^{(t,t_{\text{loc}})} \right) - \frac{1}{T_{\text{loc}}} \sum_{t_{\text{loc}}=0}^{T_{\text{loc}}-1} \mathbf{g}_{h,k}^{(t,t_{\text{loc}})} \right\|^2 \right]
$$

$$
\overset{\text{Eq. 29}}{\le} 2\sigma_{\text{glob}}^2 + \frac{4}{T_{\text{loc}}} \sum_{k=1}^{K} \omega_k \sum_{t_{\text{loc}}=0}^{T_{\text{loc}}-1} \mathbb{E}_{t_{\text{loc}}} \left[ \left\| \nabla \mathscr{L}_{h,k} \left( \boldsymbol{\theta}_h^{(t)} \right) - \nabla \mathscr{L}_{h,k} \left( \boldsymbol{\theta}_{h,k}^{(t,t_{\text{loc}})} \right) \right\|^2 \right]
$$

$$
+ \frac{4}{T_{\text{loc}}} \sum_{k=1}^{K} \omega_k \sum_{t_{\text{loc}}=0}^{T_{\text{loc}}-1} \mathbb{E}_{t_{\text{loc}}} \left[ \left\| \nabla \mathscr{L}_{h,k} \left( \boldsymbol{\theta}_{h,k}^{(t,t_{\text{loc}})} \right) - \mathbf{g}_{h,k}^{(t,t_{\text{loc}})} \right\|^2 \right]
$$

$$
\overset{\text{A2.2}}{\le} 2\sigma_{\text{glob}}^2 + \frac{4}{T_{\text{loc}}} \sum_{k=1}^{K} \omega_k \sum_{t_{\text{loc}}=0}^{T_{\text{loc}}-1} \mathbb{E}_{t_{\text{loc}}} \left[ \left\| \nabla \mathscr{L}_{h,k} \left( \boldsymbol{\theta}_{h,k}^{(t,0)} \right) - \nabla \mathscr{L}_{h,k} \left( \boldsymbol{\theta}_{h,k}^{(t,t_{\text{loc}})} \right) \right\|^2 \right] + 4\sigma_{\text{loc}}^2
$$

$$\overset{\text{A1.1}}{\leq} 2\sigma_{\text{glob}}^2 + 4\sigma_{\text{loc}}^2 + \frac{4L^2}{T_{\text{loc}}} \sum_{k=1}^{K} \omega_k \sum_{t_{\text{loc}}=0}^{T_{\text{loc}}-1} \mathbb{E}_{t_{\text{loc}}} \left[ \left\| \boldsymbol{\theta}_{h,k}^{(t,0)} - \boldsymbol{\theta}_{h,k}^{(t,t_{\text{loc}})} \right\|^2 \right]$$

$$\overset{\text{Eq. 17}}{=} 2\sigma_{\text{glob}}^2 + 4\sigma_{\text{loc}}^2 + \frac{4L^2}{T_{\text{loc}}} \sum_{k=1}^{K} \omega_k \underbrace{\sum_{t_{\text{loc}}=0}^{T_{\text{loc}}-1} \mathbb{E}_{t_{\text{loc}}} \left[ \left\| \eta_{\text{loc}} \sum_{s=0}^{t_{\text{loc}}-1} \mathbf{g}_{h,k}^{(t,s)} \right\|^2 \right]}_{\mathbf{W}}, \tag{36}$$

where

$$\mathbf{W} = \sum_{t_{\text{loc}}=0}^{T_{\text{loc}}-1} \mathbb{E}_{t_{\text{loc}}} \left[ \left\| \eta_{\text{loc}} \sum_{s=0}^{t_{\text{loc}}-1} \mathbf{g}_{h,k}^{(t,s)} \right\|^2 \right] \overset{\text{Eq. 30}}{\leq} \eta_{\text{loc}}^2 \sum_{t_{\text{loc}}=0}^{T_{\text{loc}}-1} t_{\text{loc}} \sum_{s=0}^{t_{\text{loc}}-1} \mathbb{E}_{t_{\text{loc}}} \left[ \left\| \mathbf{g}_{h,k}^{(t,s)} \right\|^2 \right]$$

$$\overset{\|\nabla \ell_h(\boldsymbol{x},\boldsymbol{\theta})\| \leq M_h}{\leq} \eta_{\text{loc}}^2 M_h^2 \sum_{t_{\text{loc}}=0}^{T_{\text{loc}}-1} t_{\text{loc}}^2 \leq \eta_{\text{loc}}^2 M_h^2 T_{\text{loc}}^3. \tag{37}$$

Substituting it back in $\mathbf{X}$, we get:

$$\mathbf{X} \leq 2\sigma_{\text{glob}}^2 + 4\sigma_{\text{loc}}^2 + \frac{4L^2}{T_{\text{loc}}} \sum_{k=1}^{K} \omega_k \eta_{\text{loc}}^2 M_h^2 T_{\text{loc}}^3 = 2\sigma_{\text{glob}}^2 + 4\sigma_{\text{loc}}^2 + 4L^2 \eta_{\text{loc}}^2 T_{\text{loc}}^2 M_h^2, \tag{38}$$

which we can substitute in $\mathbf{Y}_1$, thus getting the bound:

$$\mathbf{Y}_1 \leq 2(1 - \eta_{\text{loc}} T_{\text{loc}})^2 \mathbb{E}_{t_{\text{loc}}} \left[ \left\| \nabla \mathscr{L}_h \left( \boldsymbol{\theta}_h^{(t)} \right) \right\|^2 \right] + 2\eta_{\text{loc}}^2 T_{\text{loc}}^2 \underbrace{\sum_{k=1}^{K} \omega_k \mathbb{E}_{t_{\text{loc}}} \left[ \left\| \nabla \mathscr{L}_h \left( \boldsymbol{\theta}_h^{(t)} \right) - \mathbf{H}_{h,k}^{(t)} \right\|^2 \right]}_{\mathbf{X}}$$

$$\leq 2(1 - \eta_{\text{loc}} T_{\text{loc}})^2 \mathbb{E}_{t_{\text{loc}}} \left[ \left\| \nabla \mathscr{L}_h \left( \boldsymbol{\theta}_h^{(t)} \right) \right\|^2 \right] + 2\eta_{\text{loc}}^2 T_{\text{loc}}^2 \left[ 2\sigma_{\text{glob}}^2 + 4\sigma_{\text{loc}}^2 + 4L^2 \eta_{\text{loc}}^2 T_{\text{loc}}^2 M_h^2 \right]$$

$$= 2(1 - \eta_{\text{loc}} T_{\text{loc}})^2 \mathbb{E}_{t_{\text{loc}}} \left[ \left\| \nabla \mathscr{L}_h \left( \boldsymbol{\theta}_h^{(t)} \right) \right\|^2 \right] + 4\eta_{\text{loc}}^2 T_{\text{loc}}^2 \sigma_{\text{glob}}^2 + 8\eta_{\text{loc}}^2 T_{\text{loc}}^2 \sigma_{\text{loc}}^2 + 8L^2 \eta_{\text{loc}}^4 T_{\text{loc}}^4 M_h^2. \tag{39}$$

**Bounding $\mathbf{Y}_2$ term**

We next bound $\mathbf{Y}_2$ using $\mathbf{G}_{h,k}^{(t)}$ defined in Equation 18 and its bound defined in Equation 20:

$$\mathbf{Y}_2 = \mathbb{E}_{t_{\text{loc}}} \left[ \left\| \mathbf{G}_{h,k}^{(t)} - \overline{\alpha}_h^{(t)} \mathbf{G}_{h,k}^{(t)} + \overline{\alpha}_h^{(t)} \mathbf{G}_{h,k}^{(t)} - \mathbf{p}_h^{(t)} \odot \overline{\boldsymbol{\Delta}}_{h,k}^{(t)} \right\|^2 \right]$$

$$\overset{\text{Eq. 30}}{\leq} 2\mathbb{E}_{t_{\text{loc}}} \left[ \left\| \mathbf{G}_{h,k}^{(t)} - \overline{\alpha}_h^{(t)} \mathbf{G}_{h,k}^{(t)} \right\|^2 \right]$$

$$+ 2\mathbb{E}_{t_{\text{loc}}} \left[ \left\| \overline{\alpha}_h^{(t)} \mathbf{G}_{h,k}^{(t)} - \mathbf{p}_h^{(t)} \odot \overline{\boldsymbol{\Delta}}_{h,k}^{(t)} \right\|^2 \right]$$

$$\overset{\text{Lemma 3 and Eq. 20}}{\leq} \frac{\eta_{\text{loc}}^2 T_{\text{loc}}^2 M_h^2}{2C_h^2} + 2\mathbb{E}_{t_{\text{loc}}} \left[ \left\| \overline{\alpha}_h^{(t)} \mathbf{G}_{h,k}^{(t)} - \mathbf{p}_h^{(t)} \odot \overline{\boldsymbol{\Delta}}_{h,k}^{(t)} \right\|^2 \right]. \tag{40}$$

Given the theorem's $\left\| \mathbf{1} - \mathbf{p}_h^{(t)} \right\|_\infty \leq P_h$ assumption[5], the latter term we can bound as:

$$\mathbb{E}_{t_{\mathrm{loc}}}\left[\left\|\overline{\alpha}_h^{(t)}\mathbf{G}_{h,k}^{(t)} - \mathbf{p}_h^{(t)}\odot\overline{\mathbf{\Delta}}_{h,k}^{(t)}\right\|^2\right] = \mathbb{E}_{t_{\mathrm{loc}}}\left[\left\|\left(\mathbf{1}-\mathbf{p}_h^{(t)}\right)\odot\overline{\alpha}_h^{(t)}\mathbf{G}_{h,k}^{(t)}\right\|^2\right]$$

$$\overset{\text{similar to Eq. 32}}{\leq} P_h^2\mathbb{E}_{t_{\mathrm{loc}}}\left[\left\|\overline{\alpha}_h^{(t)}\mathbf{G}_{h,k}^{(t)}\right\|^2\right] \overset{\text{Eq. 20}}{\leq} P_h^2\eta_{\mathrm{loc}}^2 T_{\mathrm{loc}}^2 M_h^2\left|\overline{\alpha}_h^{(t)}\right|^2 \leq P_h^2\eta_{\mathrm{loc}}^2 T_{\mathrm{loc}}^2 M_h^2. \quad (41)$$

Substituting the latest bound back into $\mathbf{Y}_2$, we finally can write:

$$\mathbf{Y}_2 \leq \frac{\eta_{\mathrm{loc}}^2 T_{\mathrm{loc}}^2 M_h^2}{2C_h^2} + 2P_h^2\eta_{\mathrm{loc}}^2 T_{\mathrm{loc}}^2 M_h^2. \quad (42)$$

**Bounding $\mathbf{Y}_3$ term**

We can next bound $\mathbf{Y}_3$ as follows:

$$\mathbf{Y}_3 = \mathbb{E}_{t_{\mathrm{loc}}}\left[\left\|\mathbf{p}_h^{(t)}\odot\left(\mathbf{\Delta}_{h,k}^{(t)} - \widetilde{\mathbf{\Delta}}_{h,k}^{(t)}\right)\right\|^2\right] \overset{\text{Eq. 32}}{\leq} \frac{R_h^2}{\xi^2}\mathbb{E}_{t_{\mathrm{loc}}}\left[\left\|\mathbf{\Delta}_{h,k}^{(t)} - \widetilde{\mathbf{\Delta}}_{h,k}^{(t)}\right\|^2\right]$$

$$= \frac{R_h^2}{\xi^2}\mathbb{E}_{t_{\mathrm{loc}}}\left[\left\|\alpha_{h,k}^{(t)}\mathbf{G}_{h,k}^{(t)} - \widetilde{\alpha}_{h,k}^{(t)}\mathbf{G}_{h,k}^{(t)}\right\|^2\right] = \frac{R_h^2}{\xi^2}\mathbb{E}_{t_{\mathrm{loc}}}\left[\left(\alpha_{h,k}^{(t)} - \widetilde{\alpha}_{h,k}^{(t)}\right)^2\left\|\mathbf{G}_{h,k}^{(t)}\right\|^2\right]$$

$$\overset{\text{Eq. 20}}{\leq} \frac{R_h^2}{\xi^2}\eta_{\mathrm{loc}}^2 T_{\mathrm{loc}}^2 M_h^2\mathbb{E}_{t_{\mathrm{loc}}}\left[\left(\alpha_{h,k}^{(t)} - \widetilde{\alpha}_{h,k}^{(t)}\right)^2\right]. \quad (43)$$

Using $\alpha_{h,k}^{(t)}$ and $\widetilde{\alpha}_{h,k}^{(t)}$ defined in Equations 19 and 21, we have the following:

$$\left(\alpha_{h,k}^{(t)} - \widetilde{\alpha}_{h,k}^{(t)}\right)^2 = \left(\frac{C_h}{\max\left(C_h,\left\|\mathbf{G}_{h,k}^{(t)}\right\|\right)} - \frac{C_h}{\max\left(C_h,\mathbb{E}_{t_{\mathrm{loc}}}\left[\left\|\mathbf{G}_{h,k}^{(t)}\right\|\right]\right)}\right)^2$$

$$\overset{\text{Lemma 1}}{\leq} \frac{\left(\left\|\mathbf{G}_{h,k}^{(t)}\right\| - \mathbb{E}_{t_{\mathrm{loc}}}\left[\left\|\mathbf{G}_{h,k}^{(t)}\right\|\right]\right)^2}{C_h^2}. \quad (44)$$

Consequently, $\mathbf{Y}_3$ can be bounded as

$$\mathbf{Y}_3 \leq \frac{R_h^2}{\xi^2}\eta_{\mathrm{loc}}^2 T_{\mathrm{loc}}^2 M_h^2\mathbb{E}_{t_{\mathrm{loc}}}\left[\left(\alpha_{h,k}^{(t)} - \widetilde{\alpha}_{h,k}^{(t)}\right)^2\right]$$

$$\leq \frac{R_h^2}{\xi^2}\eta_{\mathrm{loc}}^2 T_{\mathrm{loc}}^2 M_h^2\frac{\mathbb{E}_{t_{\mathrm{loc}}}\left[\left(\left\|\mathbf{G}_{h,k}^{(t)}\right\| - \mathbb{E}_{t_{\mathrm{loc}}}\left[\left\|\mathbf{G}_{h,k}^{(t)}\right\|\right]\right)^2\right]}{C_h^2}$$

$$= \frac{R_h^2}{\xi^2}\eta_{\mathrm{loc}}^2 T_{\mathrm{loc}}^2 M_h^2\frac{\mathsf{Var}_{t_{\mathrm{loc}}}\left(\left\|\mathbf{G}_{h,k}^{(t)}\right\|\right)}{C_h^2}. \quad (45)$$

**Bounding $\mathbf{Y}_4$ term**

We can finally bound $\mathbf{Y}_4$ as follows:

$$\mathbf{Y}_4 = \mathbb{E}_{t_{\mathrm{loc}}}\left[\left\|\mathbf{p}_h^{(t)}\odot\left(\widetilde{\mathbf{\Delta}}_{h,k}^{(t)} - \overline{\mathbf{\Delta}}_{h,k}^{(t)}\right)\right\|^2\right] \overset{\text{Eq. 32}}{\leq} \frac{R_h^2}{\xi^2}\mathbb{E}_{t_{\mathrm{loc}}}\left[\left\|\widetilde{\mathbf{\Delta}}_{h,k}^{(t)} - \overline{\mathbf{\Delta}}_{h,k}^{(t)}\right\|^2\right]$$

$$= \frac{R_h^2}{\xi^2}\mathbb{E}_{t_{\mathrm{loc}}}\left[\left\|\widetilde{\alpha}_{h,k}^{(t)}\mathbf{G}_{h,k}^{(t)} - \overline{\alpha}_{h,k}^{(t)}\mathbf{G}_{h,k}^{(t)}\right\|^2\right] = \frac{R_h^2}{\xi^2}\mathbb{E}_{t_{\mathrm{loc}}}\left[\left(\widetilde{\alpha}_{h,k}^{(t)} - \overline{\alpha}_{h,k}^{(t)}\right)^2\left\|\mathbf{G}_{h,k}^{(t)}\right\|^2\right]$$

$$\overset{\text{Eq. 20}}{\leq} \frac{R_h^2}{\xi^2}\eta_{\mathrm{loc}}^2 T_{\mathrm{loc}}^2 M_h^2\mathbb{E}_{t_{\mathrm{loc}}}\left[\left(\widetilde{\alpha}_{h,k}^{(t)} - \overline{\alpha}_{h,k}^{(t)}\right)^2\right]. \quad (46)$$

---

[5]This assumption is reasonable given LAMB optimizer bounds its trust ratio.

Similarly, using $\widetilde{\alpha}_{h,k}^{(t)}$ and $\overline{\alpha}_{h,k}^{(t)}$ defined in Equations 21 and 22 we have the following:

$$
\left(\widetilde{\alpha}_{h,k}^{(t)} - \overline{\alpha}_{h,k}^{(t)}\right)^2 = \left(\frac{C_h}{\max\left(C_h, \mathbb{E}_{t_{\mathrm{loc}}}\left[\left\|\mathbf{G}_{h,k}^{(t)}\right\|\right]\right)} - \frac{C_h}{\max\left(C_h, \mathbb{E}_{t_{\mathrm{loc}},k}\left[\left\|\mathbf{G}_{h,k}^{(t)}\right\|\right]\right)}\right)^2
$$

$$
\overset{\text{Lemma } 1}{\leq} \frac{\left(\mathbb{E}_{t_{\mathrm{loc}}}\left[\left\|\mathbf{G}_{h,k}^{(t)}\right\|\right] - \mathbb{E}_{t_{\mathrm{loc}},k}\left[\left\|\mathbf{G}_{h,k}^{(t)}\right\|\right]\right)^2}{C_h^2}. \tag{47}
$$

We can thus bound $\mathbf{Y}_4$ as

$$
\mathbf{Y}_4 \leq \frac{R_h^2}{\xi^2}\eta_{\mathrm{loc}}^2 t_{\mathrm{loc}}^2 M_h^2 \mathbb{E}_{t_{\mathrm{loc}}}\left[\left(\widetilde{\alpha}_{h,k}^{(t)} - \overline{\alpha}_{h,k}^{(t)}\right)^2\right]
$$

$$
\leq \frac{R_h^2}{\xi^2}\eta_{\mathrm{loc}}^2 T_{\mathrm{loc}}^2 M_h^2 \frac{\left(\mathbb{E}_{t_{\mathrm{loc}}}\left[\left\|\mathbf{G}_{h,k}^{(t)}\right\|\right] - \mathbb{E}_{t_{\mathrm{loc}},k}\left[\left\|\mathbf{G}_{h,k}^{(t)}\right\|\right]\right)^2}{C_h^2}. \tag{48}
$$

**Final bound on $\mathbf{Z}_h$ term**

Substituting $\mathbf{Y}_1, \mathbf{Y}_2, \mathbf{Y}_3,$ and $\mathbf{Y}_4$ back in $\mathbf{Z}_h$ and having $k \sim \mathrm{Categorical}(\omega_1, \dots, \omega_K)$, we get:

$$
\mathbf{Z}_h \leq 8(1 - \eta_{\mathrm{loc}}T_{\mathrm{loc}})^2 \mathbb{E}_{t_{\mathrm{loc}}}\left[\left\|\nabla\mathscr{L}_h\left(\boldsymbol{\theta}_h^{(t)}\right)\right\|^2\right] + 16\eta_{\mathrm{loc}}^2 T_{\mathrm{loc}}^2 \sigma_{\mathrm{glob}}^2 + 32\eta_{\mathrm{loc}}^2 T_{\mathrm{loc}}^2 \sigma_{\mathrm{loc}}^2 + 32L^2\eta_{\mathrm{loc}}^4 T_{\mathrm{loc}}^4 M_h^2
$$

$$
+ \frac{2\eta_{\mathrm{loc}}^2 T_{\mathrm{loc}}^2 M_h^2}{C_h^2} + 8P_h^2\eta_{\mathrm{loc}}^2 T_{\mathrm{loc}}^2 M_h^2
$$

$$
+ 4\frac{R_h^2}{\xi^2}\eta_{\mathrm{loc}}^2 T_{\mathrm{loc}}^2 M_h^2 \frac{\sum_{k=1}^K \omega_k \mathsf{Var}_{t_{\mathrm{loc}}}\left(\left\|\mathbf{G}_{h,k}^{(t)}\right\|\right)}{C_h^2}
$$

$$
+ 4\frac{R_h^2}{\xi^2}\eta_{\mathrm{loc}}^2 T_{\mathrm{loc}}^2 M_h^2 \frac{\sum_{k=1}^K \omega_k \left(\mathbb{E}_{t_{\mathrm{loc}}}\left[\left\|\mathbf{G}_{h,k}^{(t)}\right\|\right] - \mathbb{E}_{t_{\mathrm{loc}},k}\left[\left\|\mathbf{G}_{h,k}^{(t)}\right\|\right]\right)^2}{C_h^2} + \frac{R_h^2}{\xi^2}d_h C^2 \sigma_{\mathrm{DP}}^2
$$

$$
= 8(1 - \eta_{\mathrm{loc}}T_{\mathrm{loc}})^2 \mathbb{E}_{t_{\mathrm{loc}}}\left[\left\|\nabla\mathscr{L}_h\left(\boldsymbol{\theta}_h^{(t)}\right)\right\|^2\right] + 16\eta_{\mathrm{loc}}^2 T_{\mathrm{loc}}^2 \sigma_{\mathrm{glob}}^2 + 32\eta_{\mathrm{loc}}^2 T_{\mathrm{loc}}^2 \sigma_{\mathrm{loc}}^2
$$

$$
+ \eta_{\mathrm{loc}}^2 T_{\mathrm{loc}}^2 M_h^2 \left[32L^2\eta_{\mathrm{loc}}^2 T_{\mathrm{loc}}^2 + \frac{2}{C_h^2} + 8P_h^2\right]
$$

$$
+ 4\frac{R_h^2}{\xi^2}\frac{\eta_{\mathrm{loc}}^2 T_{\mathrm{loc}}^2 M_h^2}{C_h^2}\left[\mathbb{E}_k\left[\mathsf{Var}_{t_{\mathrm{loc}}}\left(\left\|\mathbf{G}_{h,k}^{(t)}\right\|\right)\right] + \mathsf{Var}_k\left(\mathbb{E}_{t_{\mathrm{loc}}}\left[\left\|\mathbf{G}_{h,k}^{(t)}\right\|\right]\right)\right] + \frac{R_h^2}{\xi^2}d_h C^2 \sigma_{\mathrm{DP}}^2. \tag{49}
$$

**Final bound on loss $\mathbb{E}_{t_{\mathrm{loc}}}\left[\mathscr{L}\left(\boldsymbol{\theta}^{(t+1)}\right)\right]$**

We can thus rewrite Equation 27 having $\kappa = \left[1 - 8(1 - \eta_{\mathrm{loc}}T_{\mathrm{loc}})^2\right]$ as

$$
\mathbb{E}_{t_{\mathrm{loc}}}\left[\mathscr{L}\left(\boldsymbol{\theta}^{(t+1)}\right)\right] \leq \mathscr{L}\left(\boldsymbol{\theta}^{(t)}\right) - \frac{\kappa\eta_{\mathrm{glob}}}{2}\mathbb{E}_{t_{\mathrm{loc}}}\left[\left\|\nabla\mathscr{L}\left(\boldsymbol{\theta}^{(t)}\right)\right\|^2\right]
$$

$$
+ 8H\eta_{\mathrm{glob}}\eta_{\mathrm{loc}}^2 T_{\mathrm{loc}}^2 \sigma_{\mathrm{glob}}^2 + 16H\eta_{\mathrm{glob}}\eta_{\mathrm{loc}}^2 T_{\mathrm{loc}}^2 \sigma_{\mathrm{loc}}^2
$$

$$
+ \eta_{\mathrm{glob}}\eta_{\mathrm{loc}}^2 T_{\mathrm{loc}}^2 \sum_{h=1}^H M_h^2 \left[16L^2\eta_{\mathrm{loc}}^2 T_{\mathrm{loc}}^2 + \frac{1}{C_h^2} + 4P_h^2\right]
$$

$$
+ 2\frac{\eta_{\mathrm{glob}}\eta_{\mathrm{loc}}^2 T_{\mathrm{loc}}^2}{\xi^2}\sum_{h=1}^H \frac{R_h^2 M_h^2}{C_h^2}\left[\mathbb{E}_k\left[\mathsf{Var}_{t_{\mathrm{loc}}}\left(\left\|\mathbf{G}_{h,k}^{(t)}\right\|\right)\right] + \mathsf{Var}_k\left(\mathbb{E}_{t_{\mathrm{loc}}}\left[\left\|\mathbf{G}_{h,k}^{(t)}\right\|\right]\right)\right]
$$

$$
+ \frac{\eta_{\mathrm{glob}}C^2\sigma_{\mathrm{DP}}^2}{2\xi^2}\sum_{h=1}^H R_h^2 d_h. \tag{50}
$$

Rearranging and taking an average over all aggregation steps $t = 0, \ldots, T-1$ and having $\boldsymbol{\theta}^\star$ such that $\mathscr{L}(\boldsymbol{\theta}^\star) \leq \mathbb{E}_{t_{\text{loc}}}[\mathscr{L}(\boldsymbol{\theta}^t)]$, we finally get:

$$
\frac{\kappa}{T} \sum_{t=0}^{T-1} \mathbb{E}_{t_{\text{loc}}} \left[ \left\| \nabla \mathscr{L}\left(\boldsymbol{\theta}^{(t)}\right) \right\|^2 \right] \leq \frac{2\left[\mathscr{L}\left(\boldsymbol{\theta}^{(0)}\right) - \mathscr{L}\left(\boldsymbol{\theta}^\star\right)\right]}{\eta_{\text{glob}} T} + 16 H \eta_{\text{loc}}^2 T_{\text{loc}}^2 \sigma_{\text{glob}}^2 + 32 H \eta_{\text{loc}}^2 T_{\text{loc}}^2 \sigma_{\text{loc}}^2
$$

$$
+ \frac{C^2 \sigma_{\text{DP}}^2}{\xi^2} \sum_{h=1}^{H} R_h^2 d_h + 2\eta_{\text{loc}}^2 T_{\text{loc}}^2 \sum_{h=1}^{H} M_h^2 \left[ 16 L^2 \eta_{\text{loc}}^2 T_{\text{loc}}^2 + \frac{1}{C_h^2} + 4 P_h^2 \right]
$$

$$
+ 4 \frac{\eta_{\text{loc}}^2 T_{\text{loc}}^2}{\xi^2 T} \sum_{h=1}^{H} \frac{R_h^2 M_h^2}{C_h^2} \sum_{t=0}^{T-1} \left[ \mathbb{E}_k \left[ \mathsf{Var}_{t_{\text{loc}}} \left( \left\| \mathbf{G}_{h,k}^{(t)} \right\| \right) \right] + \mathsf{Var}_k \left( \mathbb{E}_{t_{\text{loc}}} \left[ \left\| \mathbf{G}_{h,k}^{(t)} \right\| \right] \right) \right] \tag{51}
$$

Per Theorem 1 in [8] and Lemma 1 and Theorem 1 from [46] to guarantee $(\varepsilon, \delta)$-privacy $z^2 \geq const \frac{q^2 T \log 1/\delta}{\varepsilon^2}$, while $\sigma_{\text{DP}} = z \cdot \mathbb{S} = z \cdot \max_{i=1}^{K} \omega_i / q$. Then to get the final bound with $(\varepsilon, \delta)$-privacy guarantee, we must select $\sigma_{\text{DP}}^2 = const \frac{\max_{i=1}^{K}(\omega_i)^2 T \ln \frac{1}{\delta}}{\varepsilon^2}$. $\blacksquare$

**Remark:** For simplicity we assumed that $\beta_1 = 0$ and regularizer $\lambda = 0$ in the LAMB optimizer. However, the proof can be extended to the cases with $\beta_1 > 0$ and $\lambda > 0$.

### E.6  Finite-Time Convergence Rates

**Corollary 1.** *Assume **A1.1**, **A2.1**, **A2.2**, and **A3**, $\eta_{\text{glob}} L < 1$ and $\kappa = \left[1 - 8(1 - \eta_{\text{loc}} T_{\text{loc}})^2\right] > 0$. If the trust ratio from Eq. 10 in* LAMB *optimizer is controlled in the Algorithm 1 (global optimizer is* LAMB *and local optimizer is SGD) such that $r_h^{(t)} \leq R_h$ and $\left\| \mathbf{1} - \mathbf{p}_h^{(t)} \right\|_\infty \leq P_h$, $\beta_1 = 0$ and $\lambda = 0$ in the* LAMB *optimizer, clients are i.i.d. sampled with probability $q = 1$ (no sampling), and $\eta_{\text{glob}} = \Theta\left(\frac{1}{L\sqrt{T}}\right)$ and $\eta_{\text{loc}} = \Theta\left(\frac{1}{L\sqrt{T_{\text{loc}} T}}\right)$, then Algorithm 1 converges to a stationary point of the global loss function with the convergence bound characterized as:*

$$
\frac{\kappa}{T} \sum_{t=0}^{T-1} \mathbb{E}_{t_{\text{loc}}} \left[ \left\| \nabla \mathscr{L}\left(\boldsymbol{\theta}^{(t)}\right) \right\|^2 \right] \leq \underbrace{\mathcal{O}\left(\frac{1}{\sqrt{T}}\right)}_{\textit{optimization}} + \underbrace{\mathcal{O}\left(\frac{T_{\text{loc}} \sigma_{\text{glob}}^2}{T}\right)}_{\textit{global update noise}} + \underbrace{\mathcal{O}\left(\frac{T_{\text{loc}} \sigma_{\text{loc}}^2}{T}\right)}_{\textit{local update noise}}
$$

$$
+ \underbrace{\mathcal{O}\left(C^2 \sigma_{\text{DP}}^2 \sum_{h=1}^{H} R_h^2 d_h\right)}_{\textit{differential privacy noise}} + \underbrace{\mathcal{O}\left(\frac{T_{\text{loc}}}{T} \sum_{h=1}^{H} \frac{M_h^2}{C_h^2}\right)}_{\textit{clipping bias}} + \underbrace{\mathcal{O}\left(\frac{T_{\text{loc}}}{T} \sum_{h=1}^{H} \frac{R_h^2 M_h^2}{C_h^2} \left[\Psi_{\text{h}}^{\text{intra}} + \Psi_{\text{h}}^{\text{inter}}\right]\right)}_{\textit{intra and inter-client update variance}},
$$

$$\tag{52}$$

*where $\Psi_{\text{h}}^{\text{intra}} = \mathbb{E}_{t,k} \left[ \mathsf{Var}_{t_{\text{loc}}} \left( \left\| \mathbf{G}_{h,k}^{(t)} \right\| \right) \right]$ and $\Psi_{\text{h}}^{\text{inter}} = \mathbb{E}_t \left[ \mathsf{Var}_k \left( \mathbb{E}_{t_{\text{loc}}} \left[ \left\| \mathbf{G}_{h,k}^{(t)} \right\| \right] \right) \right]$, $k \sim$ Categorical$(\omega_1, \ldots, \omega_K)$ and $\mathbb{E}_{t_{\text{loc}}}[\cdot]$ denotes the expectation over sampled mini-batch $\mathscr{B}_k^{(t_{\text{loc}})}$ every local step $t_{\text{loc}} = 1, \ldots, T_{\text{loc}}$ from the client data: $\boldsymbol{x}_k^{(t,t_{\text{loc}})} \sim \mathcal{D}_k$, $\boldsymbol{x}_k^{(t,t_{\text{loc}})} \in \mathscr{B}_k^{(t_{\text{loc}})}$, $|\mathscr{B}_k^{(t_{\text{loc}})}| = B_k$.*

*Proof.* Using Theorem 2, we have

$$
\frac{\kappa}{T} \sum_{t=0}^{T-1} \mathbb{E}_{t_{\text{loc}}} \left[ \left\| \nabla \mathscr{L}\left(\boldsymbol{\theta}^{(t)}\right) \right\|^2 \right] \leq \frac{2\left[\mathscr{L}\left(\boldsymbol{\theta}^{(0)}\right) - \mathscr{L}\left(\boldsymbol{\theta}^\star\right)\right]}{\eta_{\text{glob}} T} + 16 H \eta_{\text{loc}}^2 T_{\text{loc}}^2 \sigma_{\text{glob}}^2 + 32 H \eta_{\text{loc}}^2 T_{\text{loc}}^2 \sigma_{\text{loc}}^2
$$

$$
+ \frac{C^2 \sigma_{\text{DP}}^2}{\xi^2} \sum_{h=1}^{H} R_h^2 d_h + 2\eta_{\text{loc}}^2 T_{\text{loc}}^2 \sum_{h=1}^{H} M_h^2 \left[ 16 L^2 \eta_{\text{loc}}^2 T_{\text{loc}}^2 + \frac{1}{C_h^2} + 4 P_h^2 \right]
$$

$$+ 4 \frac{\eta_{\text{loc}}^2 T_{\text{loc}}^2}{\xi^2 T} \sum_{h=1}^{H} \frac{R_h^2 M_h^2}{C_h^2} \sum_{t=0}^{T-1} \left[ \mathbb{E}_k \left[ \text{Var}_{t_{\text{loc}}} \left( \left\| \mathbf{G}_{h,k}^{(t)} \right\| \right) \right] + \text{Var}_k \left( \mathbb{E}_{t_{\text{loc}}} \left[ \left\| \mathbf{G}_{h,k}^{(t)} \right\| \right] \right) \right].$$

(53)

Choosing $\eta_{\text{glob}} = 1/\sqrt{T}$ and $\eta_{\text{loc}} = 1/\sqrt{T_{\text{loc}} T}$, we get $\eta_{\text{glob}} T = \sqrt{T}$ and $\eta_{\text{loc}}^2 T_{\text{loc}}^2 / T = T_{\text{loc}} / T^2$. Substituting these in the above bound we get

$$\frac{\kappa}{T} \sum_{t=0}^{T-1} \mathbb{E}_{t_{\text{loc}}} \left[ \left\| \nabla \mathscr{L} \left( \boldsymbol{\theta}^{(t)} \right) \right\|^2 \right] \leq \frac{2 \left[ \mathscr{L} \left( \boldsymbol{\theta}^{(0)} \right) - \mathscr{L} \left( \boldsymbol{\theta}^\star \right) \right]}{\sqrt{T}} + 16H \frac{T_{\text{loc}}}{T} \sigma_{\text{glob}}^2 + 32H \frac{T_{\text{loc}}}{T} \sigma_{\text{loc}}^2$$

$$+ \frac{C^2 \sigma_{\text{DP}}^2}{\xi^2} \sum_{h=1}^{H} R_h^2 d_h + 2 \frac{T_{\text{loc}}}{T} \sum_{h=1}^{H} M_h^2 \left[ 16L^2 \frac{T_{\text{loc}}}{T} + \frac{1}{C_h^2} + 4P_h^2 \right]$$

$$+ \frac{4}{\xi^2} \frac{T_{\text{loc}}}{T^2} \sum_{h=1}^{H} \frac{R_h^2 M_h^2}{C_h^2} \sum_{t=0}^{T-1} \left[ \mathbb{E}_k \left[ \text{Var}_{t_{\text{loc}}} \left( \left\| \mathbf{G}_{h,k}^{(t)} \right\| \right) \right] + \text{Var}_k \left( \mathbb{E}_{t_{\text{loc}}} \left[ \left\| \mathbf{G}_{h,k}^{(t)} \right\| \right] \right) \right].$$

(54)

Above can be rewritten as using the big-$\mathcal{O}$ and definition of $\Psi_h^{\text{intra}}$ and $\Psi_h^{\text{inter}}$ as

$$\frac{\kappa}{T} \sum_{t=0}^{T-1} \mathbb{E}_{t_{\text{loc}}} \left[ \left\| \nabla \mathscr{L} \left( \boldsymbol{\theta}^{(t)} \right) \right\|^2 \right] \leq \mathcal{O} \left( \frac{1}{\sqrt{T}} \right) + \mathcal{O} \left( \frac{T_{\text{loc}} \sigma_{\text{glob}}^2}{T} \right) + \mathcal{O} \left( \frac{T_{\text{loc}} \sigma_{\text{loc}}^2}{T} \right)$$

$$+ \mathcal{O} \left( C^2 \sigma_{\text{DP}}^2 \sum_{h=1}^{H} R_h^2 d_h \right) + \mathcal{O} \left( \frac{T_{\text{loc}}}{T} \sum_{h=1}^{H} \frac{M_h^2}{C_h^2} \right) + \mathcal{O} \left( \frac{T_{\text{loc}}}{T} \sum_{h=1}^{H} \frac{R_h^2 M_h^2}{C_h^2} \left[ \Psi_h^{\text{intra}} + \Psi_h^{\text{inter}} \right] \right).$$

(55)

∎

### E.7 Recovering Prior Bounds

**Sublinear Convergence.** Similar to prior works in FL [42, 16, 60, 12, 89, 57, 11] we highlight that Algorithm 1 follows the best known convergence rate of $\mathcal{O} \left( 1/\sqrt{T} \right)$ for non-convex setting. Furthermore, in this section we provide a sketch for recovering the approximate bound for other terms as seen in prior work:

**Federated Averaging [1, 12].** Similar to analysis in [42] (see Remark 1 about Theorem 1 & 2 in [42]), setting $\eta_{\text{glob}} = 1$ does not recover the bound in Federated Averaging. However, starting with the final convergence bound of Theorem 2:

$$\frac{\kappa}{T} \sum_{t=0}^{T-1} \mathbb{E}_{t_{\text{loc}}} \left[ \left\| \nabla \mathscr{L} \left( \boldsymbol{\theta}^{(t)} \right) \right\|^2 \right] \leq \frac{2 \left[ \mathscr{L} \left( \boldsymbol{\theta}^{(0)} \right) - \mathscr{L} \left( \boldsymbol{\theta}^\star \right) \right]}{\eta_{\text{glob}} T} + 16H \eta_{\text{loc}}^2 T_{\text{loc}}^2 \sigma_{\text{glob}}^2 + 32H \eta_{\text{loc}}^2 T_{\text{loc}}^2 \sigma_{\text{loc}}^2$$

$$+ \frac{C^2 \sigma_{\text{DP}}^2}{\xi^2} \sum_{h=1}^{H} R_h^2 d_h + 2 \eta_{\text{loc}}^2 T_{\text{loc}}^2 \sum_{h=1}^{H} M_h^2 \left[ 16L^2 \eta_{\text{loc}}^2 T_{\text{loc}}^2 + \frac{1}{C_h^2} + 4P_h^2 \right]$$

$$+ 4 \frac{\eta_{\text{loc}}^2 T_{\text{loc}}^2}{\xi^2 T} \sum_{h=1}^{H} \frac{R_h^2 M_h^2}{C_h^2} \sum_{t=0}^{T-1} \left[ \mathbb{E}_k \left[ \text{Var}_{t_{\text{loc}}} \left( \left\| \mathbf{G}_{h,k}^{(t)} \right\| \right) \right] + \text{Var}_k \left( \mathbb{E}_{t_{\text{loc}}} \left[ \left\| \mathbf{G}_{h,k}^{(t)} \right\| \right] \right) \right]$$

(56)

and substituting $\eta_{\text{glob}} = 1/\sqrt{T}$, $\eta_{\text{loc}} = 1/(T_{\text{loc}}\sqrt{T})$, $\sigma_{\text{DP}}^2 = 0$ and $C_h \to \infty$, we get,

$$\frac{\kappa}{T} \sum_{t=0}^{T-1} \mathbb{E}_{t_{\text{loc}}} \left[ \left\| \nabla \mathscr{L} \left( \boldsymbol{\theta}^{(t)} \right) \right\|^2 \right] \leq \frac{2 \left[ \mathscr{L} \left( \boldsymbol{\theta}^{(0)} \right) - \mathscr{L} \left( \boldsymbol{\theta}^{\star} \right) \right]}{(1/\sqrt{T})T} + 16H \frac{1}{T_{\text{loc}}^2} \frac{T_{\text{loc}}^2}{T} \sigma_{\text{glob}}^2$$

$$+ 32H \frac{1}{T_{\text{loc}}^2} \frac{T_{\text{loc}}^2}{T} \sigma_{\text{loc}}^2 + 2 \frac{1}{T_{\text{loc}}^2} \frac{T_{\text{loc}}^2}{T} \sum_{h=1}^{H} M_h^2 \left[ 16L^2 \frac{1}{TT_{\text{loc}}^2} T_{\text{loc}}^2 + 4P_h^2 \right]$$

$$= \frac{2 \left[ \mathscr{L} \left( \boldsymbol{\theta}^{(0)} \right) - \mathscr{L} \left( \boldsymbol{\theta}^{\star} \right) \right]}{\sqrt{T}} + \frac{16H\sigma_{\text{glob}}^2}{T} + \frac{32H\sigma_{\text{loc}}^2}{T} + \frac{2}{T} \sum_{h=1}^{H} M_h^2 \left[ 16L^2/T + 4P_h^2 \right]. \tag{57}$$

Above can be rewritten as using the big-$\mathcal{O}$ as:

$$\frac{\kappa}{T} \sum_{t=0}^{T-1} \mathbb{E}_{t_{\text{loc}}} \left[ \left\| \nabla \mathscr{L} \left( \boldsymbol{\theta}^{(t)} \right) \right\|^2 \right] \leq \mathcal{O} \left( \frac{\mathscr{L} \left( \boldsymbol{\theta}^{(0)} \right) - \mathscr{L} \left( \boldsymbol{\theta}^{\star} \right)}{\sqrt{T}} + \frac{\sigma_{\text{glob}}^2}{T} + \frac{\sigma_{\text{loc}}^2}{T} + \frac{1}{T} \right). \tag{58}$$

Similar to Theorem 1 in [12], the above bound convergences at a rate of $\mathcal{O}\left(1/\sqrt{T}\right)$ and $\mathcal{O}\left(1/T\right)$ for optimization term and the update noises $\sigma_{\text{glob}}^2$ and $\sigma_{\text{loc}}^2$. Similar convergence rates are also seen in other works [53, 16].

**Adaptive Federated Optimization [42].**  Starting with the final convergence bound of Theorem 2:

$$\frac{\kappa}{T} \sum_{t=0}^{T-1} \mathbb{E}_{t_{\text{loc}}} \left[ \left\| \nabla \mathscr{L} \left( \boldsymbol{\theta}^{(t)} \right) \right\|^2 \right] \leq \frac{2 \left[ \mathscr{L} \left( \boldsymbol{\theta}^{(0)} \right) - \mathscr{L} \left( \boldsymbol{\theta}^{\star} \right) \right]}{\eta_{\text{glob}} T} + 16H\eta_{\text{loc}}^2 T_{\text{loc}}^2 \sigma_{\text{glob}}^2 + 32H\eta_{\text{loc}}^2 T_{\text{loc}}^2 \sigma_{\text{loc}}^2$$

$$+ C^2 \frac{\sigma_{\text{DP}}^2}{\xi^2} \sum_{h=1}^{H} R_h^2 d_h + 2\eta_{\text{loc}}^2 T_{\text{loc}}^2 \sum_{h=1}^{H} M_h^2 \left[ 16L^2 \eta_{\text{loc}}^2 T_{\text{loc}}^2 + \frac{1}{C_h^2} + 4P_h^2 \right]$$

$$+ 4 \frac{\eta_{\text{loc}}^2 T_{\text{loc}}^2}{\xi^2 T} \sum_{h=1}^{H} \frac{R_h^2 M_h^2}{C_h^2} \sum_{t=0}^{T-1} \left[ \mathbb{E}_k \left[ \mathsf{Var}_{t_{\text{loc}}} \left( \left\| \mathbf{G}_{h,k}^{(t)} \right\| \right) \right] + \mathsf{Var}_k \left( \mathbb{E}_{t_{\text{loc}}} \left[ \left\| \mathbf{G}_{h,k}^{(t)} \right\| \right] \right) \right] \tag{59}$$

and substituting $\eta_{\text{glob}} = 1/\sqrt{T}$, $\eta_{\text{loc}} = 1/\left(T^{3/4} T_{\text{loc}}\right)$, $\sigma_{\text{DP}}^2 = 0$ and $C_h \to \infty$, we get:

$$\frac{\kappa}{T} \sum_{t=0}^{T-1} \mathbb{E}_{t_{\text{loc}}} \left[ \left\| \nabla \mathscr{L} \left( \boldsymbol{\theta}^{(t)} \right) \right\|^2 \right] \leq \frac{2 \left[ \mathscr{L} \left( \boldsymbol{\theta}^{(0)} \right) - \mathscr{L} \left( \boldsymbol{\theta}^{\star} \right) \right]}{(1/\sqrt{T})T} + 16H \frac{T_{\text{loc}}^2}{T_{\text{loc}}^2 T^{3/2}} \sigma_{\text{glob}}^2$$

$$+ 32H \frac{T_{\text{loc}}^2}{T_{\text{loc}}^2 T^{3/2}} \sigma_{\text{loc}}^2 + 2 \frac{T_{\text{loc}}^2}{T_{\text{loc}}^2 T^{3/2}} \sum_{h=1}^{H} M_h^2 \left[ 16L^2 \frac{T_{\text{loc}}^2}{T_{\text{loc}}^2 T^{3/2}} + 4P_h^2 \right]$$

$$= \frac{2 \left[ \mathscr{L} \left( \boldsymbol{\theta}^{(0)} \right) - \mathscr{L} \left( \boldsymbol{\theta}^{\star} \right) \right]}{\sqrt{T}} + \frac{16H\sigma_{\text{glob}}^2}{T^{3/2}} + \frac{32H\sigma_{\text{loc}}^2}{T^{3/2}} + \frac{2}{T^{3/2}} \sum_{h=1}^{H} M_h^2 \left[ 16L^2 T^{-3/2} + 4P_h^2 \right]. \tag{60}$$

Above can be rewritten as using the big-$\mathcal{O}$ as:

$$\frac{\kappa}{T} \sum_{t=0}^{T-1} \mathbb{E}_{t_{\text{loc}}} \left[ \left\| \nabla \mathscr{L} \left( \boldsymbol{\theta}^{(t)} \right) \right\|^2 \right] \leq \mathcal{O} \left( \frac{\mathscr{L} \left( \boldsymbol{\theta}^{(0)} \right) - \mathscr{L} \left( \boldsymbol{\theta}^{\star} \right)}{\sqrt{T}} + \frac{\sigma_{\text{glob}}^2}{T^{3/2}} + \frac{\sigma_{\text{loc}}^2}{T^{3/2}} + \frac{1}{T^{3/2}} \right). \tag{61}$$

Similar to Corollary 1 & 2 in [42], the above bound converges at a rate of $\mathcal{O}\left(1/\sqrt{T}\right)$ and $\mathcal{O}\left(1/T^{3/2}\right)$ for optimization term and the global update noise $\sigma_{\text{glob}}^2$ respectively. However, it follows a faster convergence rate of $\mathcal{O}\left(1/T^{3/2}\right)$ for the local update noise $\sigma_{\text{loc}}^2$ compared to a rate of $\mathcal{O}\left(1/\sqrt{T}\right)$ in [42].

**Understanding Gradient Clipping in Private SGD [60].** Starting with the final convergence bound of Theorem 2:

$$\frac{\kappa}{T}\sum_{t=0}^{T-1}\mathbb{E}_{t_{\text{loc}}}\left[\left\|\nabla\mathscr{L}\left(\boldsymbol{\theta}^{(t)}\right)\right\|^2\right] \leq \frac{2\left[\mathscr{L}\left(\boldsymbol{\theta}^{(0)}\right)-\mathscr{L}\left(\boldsymbol{\theta}^{\star}\right)\right]}{\eta_{\text{glob}}T} + 16H\eta_{\text{loc}}^2 T_{\text{loc}}^2 \sigma_{\text{glob}}^2 + 32H\eta_{\text{loc}}^2 T_{\text{loc}}^2 \sigma_{\text{loc}}^2$$

$$+ \frac{C^2\sigma_{\text{DP}}^2}{\xi^2}\sum_{h=1}^{H}R_h^2 d_h + 2\eta_{\text{loc}}^2 T_{\text{loc}}^2\sum_{h=1}^{H}M_h^2\left[16L^2\eta_{\text{loc}}^2 T_{\text{loc}}^2 + \frac{1}{C_h^2} + 4P_h^2\right]$$

$$+ 4\frac{\eta_{\text{loc}}^2 T_{\text{loc}}^2}{\xi^2 T}\sum_{h=1}^{H}\frac{R_h^2 M_h^2}{C_h^2}\sum_{t=0}^{T-1}\left[\mathbb{E}_k\left[\text{Var}_{t_{\text{loc}}}\left(\left\|\mathbf{G}_{h,k}^{(t)}\right\|\right)\right] + \text{Var}_k\left(\mathbb{E}_{t_{\text{loc}}}\left[\left\|\mathbf{G}_{h,k}^{(t)}\right\|\right]\right)\right]$$

$$\tag{62}$$

and substituting $\eta_{\text{glob}} = 1/\sqrt{T}$, $\eta_{\text{loc}} = 1/(\sqrt{T}T_{\text{loc}})$, $\sigma_{\text{DP}}^2 = \mathcal{O}\left(\frac{\max_{i=1}^{K}(\omega_i)^2 T\ln\frac{1}{\delta}}{\varepsilon^2}\right)$ (per Theorem 1 in [8] and Lemma 1 and Theorem 1 from [46] to guarantee $(\varepsilon,\delta)$-privacy $z^2 \geq const\frac{q^2 T\log 1/\delta}{\varepsilon^2}$, while $\sigma_{\text{DP}} = z\cdot\mathbb{S} = z\cdot\max_{i=1}^{K}\omega_i/q$), $R_h = 1$, $C_h = \frac{C}{\sqrt{H}}$, and $D = \sum_{h=1}^{H}d_h$, we get,

$$\frac{\kappa}{T}\sum_{t=0}^{T-1}\mathbb{E}_{t_{\text{loc}}}\left[\left\|\nabla\mathscr{L}\left(\boldsymbol{\theta}^{(t)}\right)\right\|^2\right] \leq \frac{2\left[\mathscr{L}\left(\boldsymbol{\theta}^{(0)}\right)-\mathscr{L}\left(\boldsymbol{\theta}^{\star}\right)\right]}{(1/\sqrt{T})T} + 16H\frac{1}{T_{\text{loc}}^2}\frac{T_{\text{loc}}^2}{T}\sigma_{\text{glob}}^2$$

$$+ 32H\frac{1}{T_{\text{loc}}^2}\frac{T_{\text{loc}}^2}{T}\sigma_{\text{loc}}^2 + \frac{1}{\xi^2}\mathcal{O}\left(\frac{C^2\max_{i=1}^{K}(\omega_i)^2 T\ln\frac{1}{\delta}}{\varepsilon^2}\right)\sum_{h=1}^{H}d_h$$

$$+ 2\frac{1}{T_{\text{loc}}^2}\frac{T_{\text{loc}}^2}{T}\sum_{h=1}^{H}M_h^2\left[16L^2\frac{1}{TT_{\text{loc}}^2}T_{\text{loc}}^2 + \frac{H}{C^2} + 4P_h^2\right]$$

$$+ 4\frac{T_{\text{loc}}^2}{\xi^2 T^2}\frac{1}{T_{\text{loc}}^2}\sum_{h=1}^{H}\frac{HR_h^2 M_h^2}{C^2}\sum_{t=0}^{T-1}\left[\mathbb{E}_k\left[\text{Var}_{t_{\text{loc}}}\left(\left\|\mathbf{G}_{h,k}^{(t)}\right\|\right)\right] + \text{Var}_k\left(\mathbb{E}_{t_{\text{loc}}}\left[\left\|\mathbf{G}_{h,k}^{(t)}\right\|\right]\right)\right]$$

$$= \frac{2\left[\mathscr{L}\left(\boldsymbol{\theta}^{(0)}\right)-\mathscr{L}\left(\boldsymbol{\theta}^{\star}\right)\right]}{\sqrt{T}} + \frac{16H\sigma_{\text{glob}}^2}{T} + \frac{32H\sigma_{\text{loc}}^2}{T} + \mathcal{O}\left(\frac{DC^2\max_{i=1}^{K}(\omega_i)^2 T\ln\frac{1}{\delta}}{\xi^2\varepsilon^2}\right)$$

$$+ \frac{2}{T}\sum_{h=1}^{H}M_h^2\left[16\frac{L^2}{T} + \frac{H}{C^2} + 4P_h^2\right]$$

$$+ \frac{4}{\xi^2 T^2}\sum_{h=1}^{H}\frac{HR_h^2 M_h^2}{C^2}\sum_{t=0}^{T-1}\left[\mathbb{E}_k\left[\text{Var}_{t_{\text{loc}}}\left(\left\|\mathbf{G}_{h,k}^{(t)}\right\|\right)\right] + \text{Var}_k\left(\mathbb{E}_{t_{\text{loc}}}\left[\left\|\mathbf{G}_{h,k}^{(t)}\right\|\right]\right)\right]. \tag{63}$$

Using the big-$\mathcal{O}$, above can be rewritten as

$$\frac{\kappa}{T}\sum_{t=0}^{T-1}\mathbb{E}_{t_{\text{loc}}}\left[\left\|\nabla\mathscr{L}\left(\boldsymbol{\theta}^{(t)}\right)\right\|^2\right]$$

$$\leq \mathcal{O}\left(\frac{\mathscr{L}\left(\boldsymbol{\theta}^{(0)}\right)-\mathscr{L}\left(\boldsymbol{\theta}^{\star}\right)}{\sqrt{T}} + \frac{\sigma_{\text{glob}}^2}{T} + \frac{\sigma_{\text{loc}}^2}{T} + \frac{DC^2\max_{i=1}^{K}(\omega_i)^2 T\ln\frac{1}{\delta}}{\varepsilon^2} + \frac{1}{T} + \frac{1}{C^2 T}\right). \tag{64}$$

By setting $C = T^{-1/4}$, a similar convergence bound can be recovered up to a constant from Theorem 3.1 in [60] by choosing $\eta_g = 1/T^{1/4}$, $\eta_l = 1/(T^{1/4}Q)$, $C = \eta_l Q$ ($Q$ being analogous to $T_{\text{loc}}$ in our work), and $P = 1$ ($P$ is analogous to $q$, i.e., client sampling proportion in our work) and $\omega_i = 1/K$, though our bound has better rate of convergence for global and local update noise $\mathcal{O}\left(1/T\right)$ compared to $\mathcal{O}\left(1/\sqrt{T}\right)$ in [60].

**Inverse Relationship to Clipping Constant.** While [60] analyzes clipping it does not highlight an inverse relationship with clipping constant $C$ as seen in our work. Similar inverse relationships have also been highlighted in central optimizer analysis [58].

## E.8 Adaptive Optimizers and Per-Layer Clipping: Theorem Under Limited Participation

**Estimator with Bounded Sensitivity for FL with DP**

It is common in several FL works [12, 51, 52, 16] to use weighted averaging of client updates given by:

$$\boldsymbol{\Delta}_h^{(t)} = \left( \sum_{i=1}^{|\mathcal{K}^t|} \omega_{k_i} \right)^{-1} \sum_{i=1}^{|\mathcal{K}^t|} \omega_{k_i} \left( \boldsymbol{\Delta}_{h,k_i}^{(t)} + \mathbf{z}_{h,k_i}^{(t)} \right), \tag{65}$$

where $\mathcal{K}^t$ is a set of sampled users, $\omega_s = \frac{|\mathcal{D}_s|}{\sum_{k=1}^K |\mathcal{D}_k|}$ and $|\mathcal{D}_s|$ represents the cardinality of the data on client $s$. As discussed in prior work on moments accountant for DP [46], this estimator does not have a bounded sensitivity, thus ineligible for guaranteed DP privacy. The unbounded sensitivity can be intuitively seen via the case where all the sampled clients $\mathcal{K}^t$ have low number of data points thus leading to an explosion of the term $\left( \sum_{i=1}^{|\mathcal{K}^t|} \omega_{k_i} \right)^{-1}$. Because of this, our analysis uses the unbiased sampling estimator from [46] which can be expressed as

$$\boldsymbol{\Delta}_h^{(t)} = \frac{1}{q} \sum_{i=1}^{|\mathcal{K}^t|} \omega_{k_i} \left( \boldsymbol{\Delta}_{h,k_i}^{(t)} + \mathbf{z}_{h,k_i}^{(t)} \right) = \frac{1}{q} \sum_{k=1}^K \omega_k \gamma_k^{(t)} \left( \boldsymbol{\Delta}_{h,k}^{(t)} + \mathbf{z}_{h,k}^{(t)} \right), \tag{66}$$

where $q = S/K$, users are sampled i.i.d. with probability $q$ from the population $K$ and $\gamma_k^{(t)} \sim$ Bernoulli($q$) with $\mathbb{E}\left[ \gamma_k^{(t)} \right] = q$. It can be seen that the "unbiasedness" of the estimator results from the fact:

$$\mathbb{E}\left[ \sum_{i=1}^{|\mathcal{K}^t|} \omega_{k_i} \right] = \mathbb{E}\left[ \sum_{k=1}^K \omega_k \gamma_k^{(t)} \right] = \sum_{k=1}^K \omega_k \mathbb{E}\left[ \gamma_k^{(t)} \right] = q \sum_{k=1}^K \omega_k = q. \tag{67}$$

Also, while our algorithm and analysis use this general form of unbiased sampling estimator in Equation 66, our simulation experiments use uniform averaging with $\omega_k = 1/K$.

**Theorem 3.** *Assume* **A1.1**, **A2.1**, **A2.2**, *and* **A3**, $\eta_{\text{glob}} L < 1$ *and* $\kappa = \left[ 1 - 10(1 - \eta_{\text{loc}} T_{\text{loc}})^2 \right] > 0$. *If the trust ratio from Eq. 10 in* LAMB *optimizer is controlled in the Algorithm 1 (global optimizer is* LAMB *and local optimizer is SGD) such that* $r_h^{(t)} \leq R_h$ *and* $\left\| \mathbf{1} - \mathbf{p}_h^{(t)} \right\|_\infty \leq P_h$, $\beta_1 = 0$ *and* $\lambda = 0$ *in* LAMB *optimizer, and clients are i.i.d. sampled with probability* $q$, *then after* $T$ *steps of aggregation the performance of FL with DP, per-layer clipping and layer-wise gradient normalization is characterized by the following upper bound:*

$$\frac{\kappa}{T} \sum_{t=0}^{T-1} \mathbb{E}_{t_{\text{loc}}} \left[ \left\| \nabla \mathscr{L} \left( \boldsymbol{\theta}^{(t)} \right) \right\|^2 \right] \leq \frac{2 \left[ \mathscr{L} \left( \boldsymbol{\theta}^{(0)} \right) - \mathscr{L} \left( \boldsymbol{\theta}^\star \right) \right]}{\eta_{\text{glob}} T} + 20 H \eta_{\text{loc}}^2 T_{\text{loc}}^2 \sigma_{\text{glob}}^2 + 40 H \eta_{\text{loc}}^2 T_{\text{loc}}^2 \sigma_{\text{loc}}^2$$

$$+ \frac{C^2 \sigma_{\text{DP}}^2}{\xi^2} \sum_{h=1}^H R_h^2 d_h + 5 \eta_{\text{loc}}^2 T_{\text{loc}}^2 \sum_{h=1}^H M_h^2 \left[ 8 L^2 \eta_{\text{loc}}^2 T_{\text{loc}}^2 + \frac{1}{2 q C_h^2} + \frac{P_h^2}{q} + \frac{1-q}{q} \right]$$

$$+ 5 \frac{\eta_{\text{loc}}^2 T_{\text{loc}}^2}{q \xi^2 T} \sum_{h=1}^H \frac{R_h^2 M_h^2}{C_h^2} \sum_{t=0}^{T-1} \left[ \mathbb{E}_k \left[ \mathsf{Var}_{t_{\text{loc}}} \left( \left\| \mathbf{G}_{h,k}^{(t)} \right\| \right) \right] + \mathsf{Var}_k \left( \mathbb{E}_{t_{\text{loc}}} \left[ \left\| \mathbf{G}_{h,k}^{(t)} \right\| \right] \right) \right] \tag{68}$$

*where* $k \sim \text{Categorical}(\omega_1, \ldots, \omega_K)$ *and* $\mathbb{E}_{t_{\text{loc}}} [\cdot]$ *denotes the expectation over sampled mini-batch* $\mathscr{B}_k^{(t_{\text{loc}})}$ *every local step* $t_{\text{loc}} = 1, \ldots, T_{\text{loc}}$ *from the client data:* $\boldsymbol{x}_k^{(t,t_{\text{loc}})} \sim \mathcal{D}_k$, $\boldsymbol{x}_k^{(t,t_{\text{loc}})} \in \mathscr{B}_k^{(t_{\text{loc}})}$, $|\mathscr{B}_k^{(t_{\text{loc}})}| = B_k$.

*Proof.* Using the unbiased sampling estimator from Equation 66 we start by bounding $\mathbf{Z}_h$.

**Bounding $\mathbf{Z}_h$ Under Client Sampling**

Under client sampling we have the aggregated clients updates $\mathbf{\Delta}_h^{(t)}$ in Equation 15 defined as:

$$\mathbf{\Delta}_h^{(t)} = \frac{1}{q} \sum_{k=1}^{K} \omega_k \gamma_k^{(t)} \left( \mathbf{\Delta}_{h,k}^{(t)} + \mathbf{z}_{h,k}^{(t)} \right), \tag{69}$$

where $q << 1$ in usual scenario and $\gamma_k^{(t)} \sim \text{Bernoulli}(q)$. The definition of $\mathbf{\Delta}_h^{(t)}$ in Equation 69 thus affects the bound on $\mathbf{Z}_h$ (from Equation 28) as follows:

$$
\begin{aligned}
\mathbf{Z}_h = \mathbb{E}_{t_{\text{loc}}, \gamma_k^{(t)}} \Bigg[ \Bigg\| \nabla \mathscr{L}_h \left( \boldsymbol{\theta}_h^{(t)} \right) \\
- \sum_{k=1}^{K} \omega_k \mathbf{G}_{h,k}^{(t)} + \sum_{k=1}^{K} \omega_k \mathbf{G}_{h,k}^{(t)} \\
- \frac{1}{q} \sum_{k=1}^{K} \omega_k \gamma_k \mathbf{G}_{h,k}^{(t)} + \frac{1}{q} \sum_{k=1}^{K} \omega_k \gamma_k \mathbf{G}_{h,k}^{(t)} \\
- \frac{1}{q} \sum_{k=1}^{K} \omega_k \gamma_k^{(t)} \mathbf{p}_h^{(t)} \odot \mathbf{\Delta}_{h,k}^{(t)} - \frac{1}{q} \sum_{k=1}^{K} \omega_k \gamma_k^{(t)} \mathbf{p}_h^{(t)} \odot \mathbf{z}_{h,k}^{(t)} \\
+ \frac{1}{q} \sum_{k=1}^{K} \omega_k \gamma_k^{(t)} \mathbf{p}_h^{(t)} \odot \widetilde{\mathbf{\Delta}}_{h,k}^{(t)} - \frac{1}{q} \sum_{k=1}^{K} \omega_k \gamma_k^{(t)} \mathbf{p}_h^{(t)} \odot \widetilde{\mathbf{\Delta}}_{h,k}^{(t)} \\
+ \frac{1}{q} \sum_{k=1}^{K} \omega_k \gamma_k^{(t)} \mathbf{p}_h^{(t)} \odot \overline{\mathbf{\Delta}}_{h,k}^{(t)} - \frac{1}{q} \sum_{k=1}^{K} \omega_k \gamma_k^{(t)} \mathbf{p}_h^{(t)} \odot \overline{\mathbf{\Delta}}_{h,k}^{(t)} \Bigg\|^2 \Bigg].
\end{aligned}
\tag{70}
$$

**Bounding Term with DP Noise in $\mathbf{Z}_h$ Under Client Sampling**

Having upper bound on the expectation $\mathbb{E}_{t_{\text{loc}}, \gamma_k^{(t)}} \left[ \mathbf{p}_h^{(t)} \right]_i^2 \leq \frac{R_h^2}{\xi^2}$ and random independent DP noise $\mathbf{z}_{h,k}^{(t)} \sim \mathcal{N}\left( 0, \mathbf{I}_h C^2 \sigma_{\text{DP}}^2 \frac{q}{\sum_{i=1}^{K} w_i^2} \right)$ per theorem condition (thus $\mathbf{p}_h^{(t)}$ and $\mathbf{\Delta}_h^{(t)}$ are independent variables), let's get the upper bound first for:

$$
\begin{aligned}
\mathbb{E}_{t_{\text{loc}}, \gamma_k^{(t)}} \left[ \left\| \sum_{k=1}^{K} \omega_k \gamma_k^{(t)} \mathbf{p}_h^{(t)} \odot \mathbf{z}_{h,k}^{(t)} \right\|^2 \right] &= \mathbb{E}_{t_{\text{loc}}, \gamma_k^{(t)}} \left[ \left\| \mathbf{p}_h^{(t)} \odot \sum_{k=1}^{K} \omega_k \gamma_k^{(t)} \mathbf{z}_{h,k}^{(t)} \right\|^2 \right] \\
&= \sum_{i=1}^{d_h} \mathbb{E}_{t_{\text{loc}}, \gamma_k^{(t)}} \left[ \left[ \mathbf{p}_h^{(t)} \right]_i^2 \left[ \sum_{k=1}^{K} \omega_k \gamma_k^{(t)} \mathbf{z}_{h,k}^{(t)} \right]_i^2 \right] \\
&= \sum_{i=1}^{d_h} \mathbb{E}_{t_{\text{loc}}} \left[ \mathbf{p}_h^{(t)} \right]_i^2 \mathbb{E}_{t_{\text{loc}}, \gamma_k^{(t)}} \left[ \sum_{k=1}^{K} \omega_k \gamma_k^{(t)} \mathbf{z}_{h,k}^{(t)} \right]_i^2 \leq \frac{R_h^2}{\xi^2} d_h \mathbb{E}_{t_{\text{loc}}, \gamma_k^{(t)}} \left[ \sum_{k=1}^{K} \omega_k \gamma_k^{(t)} \mathbf{z}_{h,k}^{(t)} \right]_0^2 \\
&= \frac{R_h^2}{\xi^2} d_h C^2 \sigma_{\text{DP}}^2 \frac{q}{\sum_{k=1}^{K} \omega_k^2} \sum_{k=1}^{K} \mathbb{E}_{\gamma_k^{(t)}} \left[ \omega_k^2 (\gamma_k^{(t)})^2 \right] = \frac{R_h^2}{\xi^2} d_h C^2 \sigma_{\text{DP}}^2 q^2.
\end{aligned}
\tag{71}
$$

**Bounding $\mathbf{Z}_h$ with $\mathbf{Y}_1, \mathbf{Y}_2, \mathbf{Y}_3, \mathbf{Y}_4$ and $\mathbf{W}$ Terms Under Client Sampling**

Given Equations 29, 33, 30 and 31 (we use the fact that DP noise $\mathbf{z}_{h,k}^{(t)}$ is zero-mean independent variable), we can bound $\mathbf{Z}_h$ in the following way:

$$
\mathbf{Z}_h \leq \underbrace{5\,\mathbb{E}_{t_{\text{loc}}}\left[\sum_{k=1}^{K}\omega_k\left\|\nabla\mathscr{L}_h\left(\boldsymbol{\theta}_h^{(t)}\right)-\mathbf{G}_{h,k}^{(t)}\right\|^2\right]}_{\mathbf{Y}_1}
$$

$$
+\underbrace{5\,\mathbb{E}_{t_{\text{loc}},\gamma_k^{(t)}}\left[\left\|\sum_{k=1}^{K}\omega_k\mathbf{G}_{h,k}^{(t)}-\frac{1}{q}\sum_{k=1}^{K}\omega_k\gamma_k^{(t)}\mathbf{G}_{h,k}^{(t)}\right\|^2\right]}_{\mathbf{W}}
$$

$$
+5\frac{1}{q^2}\sum_{k=1}^{K}\omega_k\mathbb{E}_{t_{\text{loc}},\gamma_k^{(t)}}\left[\left\|\gamma_k^{(t)}\left(\mathbf{G}_{h,k}^{(t)}-\mathbf{p}_h^{(t)}\odot\overline{\boldsymbol{\Delta}}_{h,s}^{(t)}\right)\right\|^2\right]
$$

$$
+5\frac{1}{q^2}\sum_{k=1}^{K}\omega_k\mathbb{E}_{t_{\text{loc}},\gamma_k^{(t)}}\left[\left\|\gamma_k^{(t)}\mathbf{p}_h^{(t)}\odot\left(\boldsymbol{\Delta}_{h,k}^{(t)}-\widetilde{\boldsymbol{\Delta}}_{h,k}^{(t)}\right)\right\|^2\right]
$$

$$
+5\frac{1}{q^2}\sum_{k=1}^{K}\omega_k\mathbb{E}_{t_{\text{loc}},\gamma_k^{(t)}}\left[\left\|\gamma_k^{(t)}\mathbf{p}_h^{(t)}\odot\left(\widetilde{\boldsymbol{\Delta}}_{h,k}^{(t)}-\overline{\boldsymbol{\Delta}}_{h,k}^{(t)}\right)\right\|^2\right]+\frac{R_h^2}{\xi^2}d_h C^2\sigma_{\text{DP}}^2
$$

$$
\leq\left\{\mathbb{E}\left[\gamma_k^{(t)}\right]^2=q;\gamma_k^{(t)}\text{ is independent variable from the gradients and deltas}\right\}
$$

$$
\leq 5\mathbf{Y}_1+5\mathbf{W}+5\frac{1}{q}\sum_{k=1}^{K}\omega_k\underbrace{\mathbb{E}_{t_{\text{loc}}}\left[\left\|\left(\mathbf{G}_{h,k}^{(t)}-\mathbf{p}_h^{(t)}\odot\overline{\boldsymbol{\Delta}}_{h,s}^{(t)}\right)\right\|^2\right]}_{\mathbf{Y}_2}
$$

$$
+5\frac{1}{q}\sum_{k=1}^{K}\omega_k\underbrace{\mathbb{E}_{t_{\text{loc}}}\left[\left\|\mathbf{p}_h^{(t)}\odot\left(\boldsymbol{\Delta}_{h,k}^{(t)}-\widetilde{\boldsymbol{\Delta}}_{h,k}^{(t)}\right)\right\|^2\right]}_{\mathbf{Y}_3}
$$

$$
+5\frac{1}{q}\sum_{k=1}^{K}\omega_k\underbrace{\mathbb{E}_{t_{\text{loc}}}\left[\left\|\mathbf{p}_h^{(t)}\odot\left(\widetilde{\boldsymbol{\Delta}}_{h,k}^{(t)}-\overline{\boldsymbol{\Delta}}_{h,k}^{(t)}\right)\right\|^2\right]}_{\mathbf{Y}_4}+\frac{R_h^2}{\xi^2}d_h C^2\sigma_{\text{DP}}^2. \tag{72}
$$

$$
\mathbf{W}=\mathbb{E}_{t_{\text{loc}},\gamma_k^{(t)}}\left[\left\|\sum_{k=1}^{K}\omega_k\left(\mathbf{G}_{h,k}^{(t)}-\frac{1}{q}\gamma_k^{(t)}\mathbf{G}_{h,k}^{(t)}\right)\right\|^2\right]
$$

$$
\overset{\text{Eq. 29}}{\leq}\frac{1}{q^2}\sum_{k=1}^{K}\omega_k\mathbb{E}_{t_{\text{loc}},\gamma_k^{(t)}}\left[\left\|(q-\gamma_k^{(t)})\mathbf{G}_{h,k}^{(t)}\right\|^2\right]
$$

$$
=\frac{1}{q^2}\sum_{k=1}^{K}\omega_k\mathbb{E}_{\gamma_k^{(t)}}\left[\left(q-\gamma_k^{(t)}\right)^2\right]\mathbb{E}_{t_{\text{loc}}}\left[\left\|\mathbf{G}_{h,k}^{(t)}\right\|^2\right]=\frac{1-q}{q}\sum_{k=1}^{K}\omega_k\mathbb{E}_{t_{\text{loc}}}\left[\left\|\mathbf{G}_{h,k}^{(t)}\right\|^2\right]
$$

$$
\overset{\text{Eq. 20}}{\leq}\frac{1-q}{q}\eta_{\text{loc}}^2 T_{\text{loc}}^2 M_h^2 \tag{73}
$$

Reusing bounds for $\mathbf{Y}_1$, $\mathbf{Y}_2$, $\mathbf{Y}_3$, $\mathbf{Y}_4$ from the proof of Theorem 2 and having $\kappa=\left[1-10(1-\eta_{\text{loc}}T_{\text{loc}})^2\right]$, we get:

$$
\frac{\kappa}{T}\sum_{t=0}^{T-1}\mathbb{E}_{t_{\text{loc}}}\left[\left\|\nabla\mathscr{L}\left(\boldsymbol{\theta}^{(t)}\right)\right\|^2\right]\leq\frac{2\left[\mathscr{L}\left(\boldsymbol{\theta}^{(0)}\right)-\mathscr{L}\left(\boldsymbol{\theta}^{\star}\right)\right]}{\eta_{\text{glob}}T}+20H\eta_{\text{loc}}^2 T_{\text{loc}}^2\sigma_{\text{glob}}^2+40H\eta_{\text{loc}}^2 T_{\text{loc}}^2\sigma_{\text{loc}}^2
$$

$$+ \frac{C^2 \sigma_{\mathrm{DP}}^2}{\xi^2} \sum_{h=1}^{H} R_h^2 d_h + 5 \eta_{\mathrm{loc}}^2 T_{\mathrm{loc}}^2 \sum_{h=1}^{H} M_h^2 \left[ 8 L^2 \eta_{\mathrm{loc}}^2 T_{\mathrm{loc}}^2 + \frac{1}{2 q C_h^2} + \frac{P_h^2}{q} + \frac{1-q}{q} \right]$$

$$+ 5 \frac{\eta_{\mathrm{loc}}^2 T_{\mathrm{loc}}^2}{q \xi^2 T} \sum_{h=1}^{H} \frac{R_h^2 M_h^2}{C_h^2} \sum_{t=0}^{T-1} \left[ \mathbb{E}_k \left[ \mathsf{Var}_{t_{\mathrm{loc}}} \left( \left\| \mathbf{G}_{h,k}^{(t)} \right\| \right) \right] + \mathsf{Var}_k \left( \mathbb{E}_{t_{\mathrm{loc}}} \left[ \left\| \mathbf{G}_{h,k}^{(t)} \right\| \right] \right) \right]. \quad (74)$$

■

**Another Estimator for Limited Participation**

In prior work [46] it was shown that another estimator can be used for weighted averaging of client updates under client sampling:

$$\boldsymbol{\Delta}_h^{(t)} = \frac{1}{\max(q_{\min}, \sum_{i=1}^{|\mathcal{K}^t|} \omega_{k_i})} \sum_{i=1}^{|\mathcal{K}^t|} \omega_{k_i} \left( \boldsymbol{\Delta}_{h,k_i}^{(t)} + \mathbf{z}_{h,k_i}^{(t)} \right), \quad (75)$$

This estimator is not unbiased compared to the one we use in Algorithm 1 and [46] gives the differential privacy guarantees for it too. To obtain the convergence bound for this estimator similar to Theorems 2 and 3 we can use the fact that ($q_{\min} \le q$)

$$\left\| \boldsymbol{\Delta}_h^{(t)} \right\| \le \frac{1}{q_{\min}} \left\| \sum_{i=1}^{|\mathcal{K}^t|} \omega_{k_i} \left( \boldsymbol{\Delta}_{h,k_i}^{(t)} + \mathbf{z}_{h,k_i}^{(t)} \right) \right\|. \quad (76)$$

Having this bound we can repeat the same steps as we did for the unbiased estimator from Algorithm 1 and get similar asymptotic bound as in Theorem 3 but with change of $q$ to $q_{\min}$ and different sensitivity bound for DP noise.

## F   Empirical Analysis: Data and Central Models Training

**Data**   We perform all experiments using two datasets of audio-transcription pairs: LibriSpeech [61] and Common Voice v13.0 [62]. These two datasets are read speech but differ in other properties, like data diversity, noise conditions, speaker variation, and speaker distribution. We not only present results with English locale in LibriSpeech and Common Voice v13.0 but also complement them with results on French and German locale from Common Voice v13.0. For LibriSpeech data, the original 16kHz sampling rate is maintained, while for Common Voice we downsampled every audio to 16kHz sampling rate.

Every split of LS and CV has a separate set of speakers as well as every validation and test sets have entirely different speakers from the train. Validation data are used to tune all hyper-parameters and to select the best models based on the word error rate (WER), while the test sets are used only for final evaluation. Statistics on the number of speakers and the number of minutes per speaker are given in Figure 1 for both LS and CV datasets and their subsets. The statistics show that CV data are much more heterogeneous than LS as highlighted by [20]. CV data thus enable a more realistic scenario for testing FL and FL with DP. The most realistic scenario for FL uses a small central dataset to train a seed model (e.g. *LS-100*), and a larger dataset from a different distribution for FL (e.g. *CV-en-train*). All training subsets used in the empirical analysis and their statistics are listed in Table 2.

**Token Set**   [82] showed that for data from different domains, character tokens are more suited than word-pieces. Since in this paper we consider settings with data from different domains, the token set used in all our experiments is composed of English characters (a-z), augmented with a word boundary token, hyphen and apostrophe, resulting in a total of 29 characters. For French and German, common non-English characters are included as well.

Table 2: Speaker statistics for LibriSpeech (LS) and Common Voice (CV) train sets and their subsets.

| Subset | # hours | # speakers | # minutes per speaker | | | |
|---|---|---|---|---|---|---|
| | | | mean | std | min | max |
| LS-100 | 100.6 | 251 | 24.1 | 2.7 | 5.5 | 25.2 |
| LS-360 | 363.6 | 921 | 23.7 | 3.2 | 1.9 | 25.3 |
| LS-500 | 496.9 | 1,166 | 25.6 | 5.9 | 3.0 | 30.3 |
| LS-860 | 860.5 | 2,087 | 24.7 | 5.1 | 1.9 | 30.3 |
| LS-960 | 961.1 | 2,338 | 24.7 | 4.9 | 1.9 | 30.3 |
| CV-en-train | 1593.7 | 34,753 | 2.8 | 32.7 | 0.02 | 5,049.6 |
| CV-en-train-10 | 149.5 | 3,475 | 2.6 | 17.3 | 0.03 | 755.1 |
| CV-en-train-90 | 1444.2 | 31,278 | 2.8 | 34.0 | 0.02 | 5,049.6 |
| CV-en-train-05 | 79.5 | 1,737 | 2.7 | 15.8 | 0.03 | 508.3 |
| CV-en-train-95 | 1514.2 | 33,016 | 2.7 | 33.4 | 0.02 | 5,049.6 |
| CV-fr-train | 727.9 | 6,856 | 6.4 | 57.2 | 0.04 | 3081.2 |
| CV-fr-train-10 | 47.6 | 685 | 4.2 | 13.6 | 0.07 | 235.1 |
| CV-fr-train-90 | 680.3 | 6,171 | 6.6 | 60.2 | 0.04 | 3081.2 |
| CV-de-train | 852.8 | 7,127 | 7.2 | 89.2 | 0.03 | 6249.9 |
| CV-de-train-10 | 52.2 | 712 | 4.4 | 11.4 | 0.04 | 120.8 |
| CV-de-train-90 | 800.6 | 6,415 | 7.5 | 94.0 | 0.03 | 6249.9 |

**Data preprocessing**  For CV English, transcriptions are normalized similarly as for LS by (i) lower casing; (ii) removing punctuation while preserving hyphen; and (iii) converting non-English characters into English ones with `unidecode`[6] package. For CV French and German, we do not remove non-English characters and we retain single quotes.

**Model**  We start our experimentation with the state-of-the-art model on *LS-100* from [63]: (i) 1D convolution to perform striding (kernel of 7 with stride of 3); (ii) a transformer encoder with 36 layers, post-LayerNorm, 4 attention heads, an embedding dimension of 768, an MLP dimension of 3072, a dropout and layer drop [90] of 0.3; and (iii) a linear layer to map to the target vocabulary. The resulting model has 255M trainable parameters. We focus only on a CTC model as it contains only the encoder part, is simpler to train in practice compared to Seq2Seq or Transducer models, and is less likely to over-fit to the language model [25].

**Positional Embedding**  To reduce model training time by a factor of approximately 2-3 and to reduce the memory footprint, we use CAPE positional embedding [91] instead of relative positional embedding [92]; both models perform similarly.

**SpecAugment**  SpecAugment [64] is activated from the very first step of training. Two frequency masks with frequency mask parameter $F = 30$, ten time masks with maximum time-mask ratio $p = 0.1$ and time mask parameter $T = 50$ are used; time warping is not used.

**Training**  We train models on 8 GPUs (A100 80GB), and use a dynamic batch size of $\sim 240$s audio per GPU. For all central models training, we use LARS optimizer with the learning rate of 0.5 (for models fine-tuned from seed models trained on *CV-\*-train-10* we use 0.2) without a warmup period. Training is done for up to 300k-600k steps until full convergence with step-wise (by 2x) learning rate decay every 50k steps started at 40k-330k depending on the model.

# G  Empirical Analysis: Federated Learning without Differential Privacy

## G.1  Hyper-parameters

All dropout and layer drop are fixed to 0.3 We train each client with a dynamic batch size of total 120s of audio (CV) or 360s of audio (LS). In Figures 2 and 3 we use the same LR and LR decay schedule for all seed models regardless of the cohort size or the data used to train a seed model. Optimal hyper-parameters (e.g. LR) are likely to depend on the quality of the seed model and cohort size. Thus, the results could likely be further improved by tuning the LR and its decay schedule for

---

[6]`https://pypi.org/project/Unidecode`.

Table 3: Results (WER %) on LS. All runs use exponential decay for central LR starting at iteration 1,000, decay rate 0.6, and transition steps 500 (w/o seed model) or 250 (w/ seed model). Local learning rate is 0.4 (w/o seed model) or 0.2 (w/ seed model). Central learning rate is 0.006 (w/o seed model) or 0.003 (w/ seed model). The number of central steps is $T = 2$k and the number of local epochs is 10.

| Data | seed: None; train: LS-960 | | | | | | seed: LS-100; train: LS-860 | | | | | | seed: CV-en; train: LS-960 | | | | | |
|---|---|---|---|---|---|---|---|---|---|---|---|---|---|---|---|---|---|---|
| | seed | 8 | 16 | 32 | 64 | central | seed | 8 | 16 | 32 | 64 | central | seed | 8 | 16 | 32 | 64 | central |
| dev-clean | 100.0 | 6.6 | 4.8 | 4.0 | 3.3 | 2.7 | 6.2 | 3.3 | 3.1 | 2.9 | 2.7 | 2.7 | 16.5 | 4.0 | 3.6 | 3.3 | 2.9 | 3.1 |
| test-clean | 100.0 | 6.7 | 5.1 | 4.2 | 3.4 | 2.8 | 6.7 | 3.4 | 3.2 | 3.0 | 2.9 | 2.9 | 15.5 | 4.3 | 3.8 | 3.5 | 3.2 | 3.2 |
| dev-other | 100.0 | 17.2 | 13.5 | 11.1 | 8.8 | 6.7 | 19.2 | 9.4 | 8.5 | 8.1 | 7.7 | 6.9 | 25.2 | 10.5 | 9.6 | 8.8 | 8.1 | 7.5 |
| test-other | 100.0 | 17.5 | 13.7 | 11.1 | 8.8 | 6.8 | 19.5 | 9.0 | 8.3 | 7.6 | 7.1 | 6.8 | 25.9 | 10.3 | 9.4 | 8.6 | 7.8 | 7.2 |

Table 4: Results (WER %) on CV. We use exponential decay for central LR starting at $t = 1,000$ (w/o seed model) or $t = 750$ (w/ seed model), decay rate 0.6, and transition steps 500 (w/o seed model) or 750 (w/ seed model) with $T = 2$k total central steps and 10 local epochs. Local (central) LR is 0.4 (0.006) (w/o seed model) or 0.2 (0.002) (w/ seed model).

| Seed | Data | Eval. | seed WER | cohort size WER | | | | | | central WER |
|---|---|---|---|---|---|---|---|---|---|---|
| | | | | 8 | 16 | 32 | 64 | 128 | 256 | |
| None | CV-en | dev | 100.0 | 62.9 | 51.9 | 41.3 | 32.9 | 27.2 | 21.3 | 15.1 |
| | | test | 100.0 | 66.7 | 56.5 | 46.3 | 38.0 | 31.9 | 25.7 | 18.2 |
| CV-en-05 | CV-en-95 | dev | 31.3 | 26.6 | 24.3 | 22.7 | 21.2 | 19.8 | 18.2 | 15.2 |
| | | test | 36.4 | 31.6 | 28.9 | 27.0 | 25.4 | 23.8 | 22.1 | 18.3 |
| CV-en-10 | CV-en-90 | dev | 23.0 | 20.3 | 18.9 | 17.7 | 16.7 | 15.7 | 14.8 | 14.5 |
| | | test | 27.9 | 24.4 | 22.8 | 21.5 | 20.1 | 19.1 | 18.0 | 17.6 |
| LS-100 | CV-en | dev | 54.7 | 24.5 | 22.2 | 20.1 | 18.4 | 16.8 | 15.6 | 14.7 |
| | | test | 61.2 | 28.8 | 26.3 | 23.9 | 22.0 | 20.2 | 18.9 | 17.8 |
| LS-960 | CV-en | dev | 27.0 | 19.7 | 18.1 | 16.9 | 15.6 | 14.5 | 13.7 | 14.1 |
| | | test | 31.5 | 23.5 | 21.6 | 20.2 | 18.8 | 17.6 | 16.6 | 17.2 |

Table 5: Impact of randomizing the distribution of data across users for LS measured by WER (%). Parameter settings are described in Table 3. While the original train data are non-IID, IID (columns with "IID") versions of *LS-960* and *LS-860* are created by choosing a user id uniformly and randomly from the set of user ids for each data point in the corresponding dataset.

| Data | seed: None; train: LS-960 | | | | | | seed: LS-100; train: LS-860 | | | | | | seed: CV-en; train: LS-960 | | | | | |
|---|---|---|---|---|---|---|---|---|---|---|---|---|---|---|---|---|---|---|
| | seed | 8 | 8-IID | 16 | 16-IID | central | seed | 8 | 8-IID | 16 | 16-IID | central | seed | 8 | 8-IID | 16 | 16-IID | central |
| dev-clean | 100.0 | 6.6 | 5.9 | 4.8 | 4.5 | 2.7 | 6.2 | 3.3 | 3.3 | 3.1 | 3.0 | 2.7 | 16.5 | 4.0 | 3.9 | 3.6 | 3.5 | 3.1 |
| test-clean | 100.0 | 6.7 | 6.0 | 5.1 | 4.7 | 6.7 | 2.8 | 3.4 | 3.3 | 3.2 | 3.1 | 2.9 | 15.5 | 4.3 | 4.1 | 3.8 | 3.7 | 3.2 |
| dev-other | 100.0 | 17.2 | 14.0 | 13.5 | 11.2 | 6.7 | 19.1 | 9.4 | 8.1 | 8.5 | 7.4 | 6.9 | 25.2 | 10.5 | 9.5 | 9.6 | 8.8 | 7.5 |
| test-other | 100.0 | 17.5 | 14.0 | 13.7 | 10.9 | 6.8 | 19.5 | 9.0 | 7.9 | 8.3 | 7.2 | 6.8 | 25.9 | 10.3 | 9.3 | 9.4 | 8.4 | 7.2 |

each cohort size and seed model separately. Furthermore, we can improve models by longer training exceeding 2k central steps as shown in ablations in Appendix G.8, Table 14.

## G.2 Detailed Results for English

Table 3 details the results for LS from Figure 2 and Table 4 details the results for CV from Figure 3. Table 5 details the results for randomized LS dataset (IID) from Figure 4 (left and middle). Table 6 details the results for randomized CV dataset (IID) from Figure 4 (right).

## G.3 Impact of Model Architecture on FL Performance in ASR

Table 7 compares several model architectures for the trivial FL scenario with cohort size 1 and 64k central iterations on *LS-100*. Cohort size of 1 is impractical but it eliminates the impact of federated averaging. The learning rates and learning rate decay schedules are tuned for each architecture. During preliminary FL experiments we have observed that pre-LayerNorm models often perform better than post-LayerNorm ones. It is of note that without a linear central learning rate warmup, we

Table 6: Impact of randomizing the distribution of data across users for CV measured by WER (%). Parameter settings are described in Table 4. While the original train data are non-IID, the IID (columns with "IID") version of *CV-en-train* is created by choosing a user id uniformly and randomly from the set of user ids for each data point in the corresponding dataset.

| Seed | Data | Eval. | seed WER | cohort size WER | | | | central WER |
|---|---|---|---|---|---|---|---|---|
| | | | | 16 | 16-IID | 32 | 32-IID | |
| None | CV-en | dev | 100.0 | 51.9 | 50.2 | 41.3 | 40.9 | 15.1 |
| | | test | 100.0 | 56.5 | 55.0 | 46.3 | 45.8 | 18.2 |
| LS-100 | CV-en | dev | 54.7 | 22.2 | 21.1 | 20.1 | 19.1 | 14.7 |
| | | test | 61.2 | 26.3 | 25.0 | 23.9 | 22.7 | 17.8 |

Table 7: Comparison (WER, %) between pre-LayerNorm and post-LayerNorm architectures in transformer for trivial FL scenario with cohort size $S = 1$ and central steps $T = 64$k on *LS-100*. pre-LayerNorm models perform best and their training is robust with respect to hyper-parameters such as the learning schedule. Central models are trained according to Appendix F. FL models use exponential learning rate decay, LAMB as central and SGD as local optimizers.

| Model | Warmup | dev-clean | dev-other | test-clean | test-other |
|---|---|---|---|---|---|
| Central pre-LayerNorm | 0 | 5.9 | 18.9 | 6.4 | 19.2 |
| FL pre-LayerNorm | 0 | 5.6 | 17.7 | 5.9 | 17.9 |
| Central post-LayerNorm | 0 | 8.1 | 25.0 | 8.6 | 25.6 |
| FL post-LayerNorm | 1000 | 5.9 | 17.5 | 6.3 | 18.0 |

Table 8: Comparison (WER, %) of various server optimizers on *LS-960* with and without a seed model. For LAMB, the results and parameters are the same as those in Table 3 (note that these are sub-optimal because for simplicity we use the same learning rate and learning rate decay schedule for each configuration regardless of the cohort size and all runs with seed models use the same configuration). For all other optimizers, the central learning rate and the learning rate decay schedule are tuned separately for each combination of cohort size and seed model.

| Seed | Data | Cohort size | Central optimizer | LR | dev-clean $T = 0$ | dev-clean $T = 2$k | test-clean $T = 0$ | test-clean $T = 2$k | dev-other $T = 0$ | dev-other $T = 2$k | test-other $T = 0$ | test-other $T = 2$k |
|---|---|---|---|---|---|---|---|---|---|---|---|---|
| None | LS-960 | 8 | LAMB | 0.006 | 100.0 | 6.6 | 100.0 | 6.7 | 100.0 | 17.2 | 100.0 | 17.5 |
| | | | LARS | 0.7 | 100.0 | 13.7 | 100.0 | 14.1 | 100.0 | 30.9 | 100.0 | 31.6 |
| | | | Adam | 0.001 | 100.0 | 14.1 | 100.0 | 14.6 | 100.0 | 30.4 | 100.0 | 31.0 |
| None | LS-960 | 16 | LAMB | 0.006 | 100.0 | 4.8 | 100.0 | 5.1 | 100.0 | 13.5 | 100.0 | 13.7 |
| | | | LARS | 0.7 | 100.0 | 10.5 | 100.0 | 11.0 | 100.0 | 25.9 | 100.0 | 25.9 |
| | | | Adam | – | - | - | - | - | - | - | - | - |
| CV-en | LS-960 | 8 | LAMB | 0.003 | 16.5 | 4.0 | 15.5 | 4.3 | 25.2 | 10.5 | 25.9 | 10.3 |
| | | | LARS | 1.2 | 16.5 | 4.2 | 15.5 | 4.4 | 25.2 | 10.6 | 25.9 | 10.6 |
| | | | Adam | 0.012 | 16.5 | 4.3 | 15.5 | 4.3 | 25.2 | 10.7 | 25.9 | 10.5 |

were unable to train reasonable FL models with post-LayerNorm. Our experiments showed that FL models with pre-LayerNorm are easier to train, they do not require a central learning rate warmup, and they are generally more robust with respect to hyper-parameters. These observations are similar to prior works on transformers central training [55, 31]. That is why we use the pre-LayerNorm configuration for all experiments in the paper. It is interesting that for this trivial FL scenario FL models outperforms centrally trained models. However, when we switch to larger *LS-960* dataset, this does not hold anymore.

## G.4 Impact of Server Optimizer on FL Performance in ASR

Table 8 compares the LAMB optimizer [49] used as the central optimizer in all FL runs presented so far with Adam [93] and LARS [65] on several configurations for *LS-960* dataset. The results on *LS-960* indicate that LAMB performs significantly better than LARS and Adam without a seed model, and it performs slightly better than LARS and Adam with a seed model. Adam performs slightly better than LARS.

Table 9: Comparison (WER, %) of various optimizers on *CV-en* with and wihout seed models. For LAMB, the results and parameters are the same as those in Table 4 (note that these are sub-optimal because for simplicity we use the same learning rate and learning rate decay schedule for each configuration regardless of the cohort size and all runs with seed models use the same configuration). For all other optimizers, the central learning rate and the learning rate decay schedule are tuned separately for each combination of cohort size and seed model.

| Seed | Data | Cohort size | Central optimizer | LR | dev $T = 0$ | dev $T = 2$k | test $T = 0$ | test $T = 2$k |
|---|---|---|---|---|---|---|---|---|
| None | CV-en | 8 | LAMB | 0.006 | 100.0 | 62.9 | 100.0 | 66.7 |
| | | | LARS | 3.4 | 100.0 | 70.4 | 100.0 | 73.8 |
| | | | Adam | 0.0005 | 100.0 | 68.9 | 100.0 | 72.2 |
| | | | AdaGrad | 0.003 | 100.0 | 84.3 | 100.0 | 86.2 |
| | | | SGD | 2.8 | 100.0 | 83.8 | 100.0 | 86.0 |
| None | CV-en | 16 | LAMB | 0.006 | 100.0 | 51.9 | 100.0 | 56.5 |
| | | | LARS | 2.6 | 100.0 | 57.6 | 100.0 | 62.0 |
| | | | Adam | 0.0005 | 100.0 | 57.7 | 100.0 | 62.1 |
| | | | AdaGrad | 0.002 | 100.0 | 82.1 | 100.0 | 84.5 |
| | | | SGD | 3.0 | 100.0 | 84.5 | 100.0 | 86.6 |
| CV-en-10 | CV-en-90 | 8 | LAMB | 0.002 | 23.0 | 19.4 | 27.9 | 23.5 |
| | | | LARS | 0.3 | 23.0 | 18.7 | 27.9 | 22.6 |
| | | | Adam | 0.004 | 23.0 | 18.9 | 27.9 | 22.9 |
| | | | AdaGrad | 0.016 | 23.0 | 19.4 | 27.9 | 23.6 |
| | | | SGD | 1.6 | 23.0 | 20.9 | 27.9 | 25.4 |
| CV-en-10 | CV-en-90 | 16 | LAMB | 0.002 | 23.0 | 18.3 | 27.9 | 22.1 |
| | | | LARS | 0.4 | 23.0 | 18.0 | 27.9 | 21.8 |
| | | | Adam | 0.006 | 23.0 | 18.3 | 27.9 | 22.1 |
| | | | AdaGrad | 0.015 | 23.0 | 19.1 | 27.9 | 23.2 |
| | | | SGD | 1.6 | 23.0 | 20.8 | 27.9 | 25.2 |
| CV-en-10 | CV-en-90 | 32 | LAMB | 0.002 | 23.0 | 17.3 | 27.9 | 21.0 |
| | | | LARS | 0.6 | 23.0 | 17.3 | 27.9 | 20.9 |
| | | | Adam | 0.006 | 23.0 | 17.5 | 27.9 | 21.1 |
| CV-en-10 | CV-en-90 | 64 | LAMB | 0.002 | 23.0 | 16.7 | 27.9 | 20.1 |
| | | | LARS | 0.5 | 23.0 | 16.6 | 27.9 | 20.1 |
| | | | Adam | 0.008 | 23.0 | 16.4 | 27.9 | 20.0 |

Table 9 compares LAMB with Adam, AdaGrad [94], LARS, and SGD [66] on several configurations for *CV-en* dataset. The results on CV show that without seed models, LAMB performs significantly better than all other optimizers but with seed models, LAMB is sometimes outperformed slightly by LARS and Adam. SGD, AdaGrad and Adam are outperformed by LAMB and LARS in almost all scenarios.

During hyper-parameter tuning, some adaptive optimizers (e.g., Adam) often became unstable and the training diverged, especially without a well performing seed model. Furthermore, the optimal parameters of these optimizers oftentimes vary significantly between, e.g., the cohort sizes, indicating that they are less robust than LAMB in our setting.

The robustness of LAMB across all scenarios and its stability are the main reasons for choosing LAMB as the central optimizer for most of the experiments in the paper. However, the results in Table 9 suggest that some of the models could be further improved with more hyper-parameters tuning and choosing the best optimizer for each case. Also, [56] showed that tuning other optimizer parameters, e.g. $\varepsilon$ in Adam, can significantly improve FL model training for ASR. However, in this paper we restrict ourselves to tuning only the learning rate and learning rate schedule; the remaining parameters were set to their default values from `optax` library[7].

We have not completed an extensive evaluation of other optimizers for local training to keep it efficient (no state, no additional memory, no extra computations): SGD as a local optimizer is robust and efficient in our experiments. However, preliminary experiments show that LARS and LAMB are well suited candidates for replacing SGD as the local optimizer and will likely outperform SGD.

For completeness, here we provide more details on optimizer tuning. For both *LS-960* and *CV-en-train* without a seed model, we tuned the central LR for LAMB between 0.001 and 0.009, and the

---

[7]https://optax.readthedocs.io/en/latest/

Table 10: Results (WER, %) on CV for English, French and German. Configurations are identical to those in Figure 3 and Table 4 regardless of the language.

| Seed | Data | Eval. | seed WER | cohort size WER | | | | | central WER |
|---|---|---|---|---|---|---|---|---|---|
| | | | | 8 | 16 | 32 | 64 | 128 | |
| None | CV-en | dev | 100.0 | 62.9 | 51.9 | 41.3 | 32.9 | 27.2 | 15.1 |
| | | test | 100.0 | 66.7 | 56.5 | 46.3 | 38.0 | 31.9 | 18.2 |
| None | CV-fr | dev | 100.0 | 34.7 | 25.4 | 18.8 | 15.0 | 12.6 | 10.7 |
| | | test | 100.0 | 38.7 | 29.1 | 21.6 | 17.5 | 14.8 | 12.2 |
| None | CV-de | dev | 100.0 | 30.1 | 22.8 | 16.1 | 11.7 | 9.5 | 7.7 |
| | | test | 100.0 | 32.8 | 25.5 | 18.3 | 13.4 | 10.9 | 8.8 |
| CV-en-10 | CV-en-90 | dev | 23.0 | 20.3 | 18.9 | 17.7 | 16.7 | 15.7 | 14.5 |
| | | test | 27.9 | 24.4 | 22.8 | 21.5 | 20.1 | 19.1 | 17.6 |
| CV-fr-10 | CV-fr-90 | dev | 24.0 | 15.6 | 14.3 | 13.2 | 12.0 | 11.2 | 10.8 |
| | | test | 27.5 | 18.1 | 16.6 | 15.3 | 14.0 | 13.1 | 12.6 |
| CV-de-10 | CV-de-90 | dev | 18.6 | 12.8 | 11.4 | 10.2 | 9.1 | 8.1 | 8.1 |
| | | test | 21.2 | 14.7 | 13.1 | 11.7 | 10.5 | 9.3 | 9.2 |

local LR for SGD from $0.2$ to $0.6$. We have done the same for one selected seed model for each dataset. Additionally, we tried several learning rate schedules, including constant rate, step decay, and exponential decay on several configurations. After the initial experiments, we chose one configuration for each dataset (LS, CV) without a seed model and one configuration for each dataset (LS, CV) with a seed model, and we ran the remaining experiments with the chosen configurations. The initial tuning was done on smaller cohort sizes. For other optimizers discussed in this section, we tuned the key parameters until a locally optimal value was found for central LR for each presented experiment, and we considered 4 variations of the exponential decay rate for each LR value.

### G.5 Detailed Results for CV French and German

Table 10 shows the results of FL on CV for French and German languages, and for comparison it provides the corresponding results on CV for English. To demonstrate that the settings used for English language were robust, we did not tune any parameters for French and German, and simply used the exact same configuration that was used in the corresponding training on English language.

The results show that even though French and German have considerably smaller datasets, the training is apparently considerably easier and WERs are significantly smaller than for English whether or not a seed model is used. This is likely due to the degree of consistency between the orthography and phonology as was discussed for example in [95, 96, 97, 98]; German and French have stronger orthography-to-phonology consistency than English. Furthermore, the results for French are considerably better than those presented by [20]. As French and German data are smaller, for the same cohort size and central steps we do more epochs over data for French and German than for English CV. Thus, FL training can match the central training with smaller cohort size for both French and German compared to English. It is of note that French and German turn out to be easier also for FL with DP as shown in Appendix H.6, Table 18.

### G.6 Impact of SpecAugment

In all experiments so far, we used SpecAugment [64] activated from the very first step of training as was also common in most prior works. Table 11 shows the results with and without SpecAugment for several configurations analyzed in this paper on LS data. These results confirm that SpecAugment improves WER in all the cases.

However, Table 12 shows that SpecAugment appears to have a negative impact on the trained models for CV (English), especially for FL training without a seed model and small cohort sizes. This is surprising as prior works reported only improved results with SpecAugment for transformer models. These results also reveal another difference between benchmarks on LS and on CV.

It is possible that the results with SpecAugment on CV would improve if SpecAugment was turned on later in the training and its parameters were tuned for each scenario separately. Nonetheless, since in

Table 11: Results (WER, %) on LS with and without SpecAugment [64]. Configurations are identical to those in Figure 2 and Table 3 except the SpecAugment schedule as noted in the table.

| Seed | Data | SpecAugment | Cohort size | dev-clean | | test-clean | | dev-other | | test-other | |
|---|---|---|---|---|---|---|---|---|---|---|---|
| | | | | $T=0$ | $T=2k$ | $T=0$ | $T=2k$ | $T=0$ | $T=2k$ | $T=0$ | $T=2k$ |
| None | LS-960 | ✓ | 8 | 100.0 | 6.6 | 100.0 | 6.7 | 100.0 | 17.2 | 100.0 | 17.5 |
| None | LS-960 | ✗ | 8 | 100.0 | 6.6 | 100.0 | 6.8 | 100.0 | 19.3 | 100.0 | 19.4 |
| None | LS-960 | ✓ | 16 | 100.0 | 4.8 | 100.0 | 5.1 | 100.0 | 13.5 | 100.0 | 13.7 |
| None | LS-960 | ✗ | 16 | 100.0 | 5.4 | 100.0 | 5.5 | 100.0 | 16.5 | 100.0 | 16.5 |
| LS-100 | LS-860 | ✓ | 8 | 6.2 | 3.3 | 6.7 | 3.4 | 19.1 | 9.4 | 19.5 | 9.0 |
| LS-100 | LS-860 | ✗ | 8 | 6.2 | 3.3 | 6.7 | 3.3 | 19.2 | 10.2 | 19.5 | 9.8 |
| LS-100 | LS-860 | ✓ | 16 | 6.2 | 3.1 | 6.7 | 3.2 | 19.1 | 8.5 | 19.5 | 8.3 |
| LS-100 | LS-860 | ✗ | 16 | 6.2 | 3.2 | 6.7 | 3.2 | 19.1 | 9.9 | 19.5 | 9.5 |
| CV-en | LS-960 | ✓ | 8 | 16.6 | 4.0 | 15.5 | 4.3 | 25.2 | 10.5 | 25.9 | 10.3 |
| CV-en | LS-960 | ✗ | 8 | 16.6 | 3.8 | 15.5 | 4.1 | 25.2 | 11.5 | 25.9 | 11.2 |
| CV-en | LS-960 | ✓ | 16 | 16.6 | 3.6 | 15.5 | 3.8 | 25.2 | 9.6 | 25.9 | 9.4 |
| CV-en | LS-960 | ✗ | 16 | 16.6 | 3.5 | 15.5 | 3.8 | 25.2 | 10.9 | 25.9 | 10.6 |

Table 12: Results (WER, %) on CV with and without SpecAugment [64]. Configurations are identical to those in Figure 3 and Table 4 except the SpecAugment schedule as noted in the table.

| Seed | Data | SpecAugment | Cohort size | dev | | test | |
|---|---|---|---|---|---|---|---|
| | | | | $T=0$ | $T=2k$ | $T=0$ | $T=2k$ |
| None | CV-en | ✓ | 8 | 100.0 | 62.9 | 100.0 | 66.7 |
| None | CV-en | ✗ | 8 | 100.0 | 52.3 | 100.0 | 57.5 |
| None | CV-en | ✓ | 16 | 100.0 | 51.9 | 100.0 | 56.5 |
| None | CV-en | ✗ | 16 | 100.0 | 42.2 | 100.0 | 47.9 |
| None | CV-en | ✓ | 32 | 100.0 | 41.3 | 100.0 | 46.3 |
| None | CV-en | ✗ | 32 | 100.0 | 33.8 | 100.0 | 39.3 |
| CV-en-10 | CV-en-90 | ✓ | 8 | 23.0 | 20.3 | 27.9 | 24.4 |
| CV-en-10 | CV-en-90 | ✗ | 8 | 23.0 | 19.9 | 27.9 | 24.3 |
| CV-en-10 | CV-en-90 | ✓ | 16 | 23.0 | 18.9 | 27.9 | 22.8 |
| CV-en-10 | CV-en-90 | ✗ | 16 | 23.0 | 18.3 | 27.9 | 22.4 |
| CV-en-10 | CV-en-90 | ✓ | 32 | 23.0 | 17.7 | 27.9 | 21.5 |
| CV-en-10 | CV-en-90 | ✗ | 32 | 23.0 | 17.1 | 27.9 | 21.2 |
| LS-100 | CV-en | ✓ | 8 | 54.7 | 24.5 | 61.2 | 28.8 |
| LS-100 | CV-en | ✗ | 8 | 54.7 | 23.3 | 61.2 | 27.9 |
| LS-100 | CV-en | ✓ | 16 | 54.7 | 22.2 | 61.2 | 26.3 |
| LS-100 | CV-en | ✗ | 16 | 54.7 | 21.0 | 61.2 | 25.4 |
| LS-100 | CV-en | ✓ | 32 | 54.7 | 20.1 | 61.2 | 23.9 |
| LS-100 | CV-en | ✗ | 32 | 54.7 | 19.0 | 61.2 | 23.0 |
| LS-960 | CV-en | ✓ | 8 | 27.0 | 19.7 | 31.5 | 23.5 |
| LS-960 | CV-en | ✗ | 8 | 27.0 | 19.5 | 31.5 | 23.5 |
| LS-960 | CV-en | ✓ | 16 | 27.0 | 18.1 | 31.5 | 21.6 |
| LS-960 | CV-en | ✗ | 16 | 27.0 | 17.8 | 31.5 | 21.6 |
| LS-960 | CV-en | ✓ | 32 | 27.0 | 16.9 | 31.5 | 20.2 |
| LS-960 | CV-en | ✗ | 32 | 27.0 | 16.4 | 31.5 | 20.2 |

most scenarios SpecAugment either improved models or the differences were marginal, for simplicity, we use SpecAugment in all experiments in this paper.

### G.7 Performance of FedProx in FL for ASR

[11] proposed FedProx to alleviate the impact of heterogeneous data on FL performance. Since the results presented earlier in Tables 5 and 6 suggested that heterogeneous data pose a challenge for FL also in our training, we also evaluate the impact of FedProx on model quality in ASR. For each configuration, we use FedProx with the regularization weight $\mu \in \{0.00001, 0.0001, 0.001, 0.01, 0.1, 1.0\}$ and chose the best result, as suggested by [11].

Table 13 presents the results of using FedProx in several scenarios on LS and CV datasets presented earlier in Tables 3 and 4. The results show FedProx improves model performance (WER is decreased) in 8 out of 10 training configurations tested, although in most cases the improvement is marginal.

Table 13: Results (WER, %) of FedProx on selected configurations on LS (top) and CV (English) (bottom) datasets. All parameters except for FedProx $\mu$ are identical to those in Tables 3 and 4. Parameter $\mu \in \{0.00001, 0.0001, 0.001, 0.01, 0.1, 1.0\}$ is tuned separately for every case and the best result is provided for each base configuration.

| Seed | Data | fedprox $\mu$ | Cohort size | dev-clean | | test-clean | | dev-other | | test-other | |
|---|---|---|---|---|---|---|---|---|---|---|---|
| | | | | $T=0$ | $T=2$k | $T=0$ | $T=2$k | $T=0$ | $T=2$k | $T=0$ | $T=2$k |
| None | LS-960 | 0 | 8 | 100.0 | 6.6 | 100.0 | 6.7 | 100.0 | 17.2 | 100.0 | 17.5 |
| None | LS-960 | 0.1 | 8 | 100.0 | 6.4 | 100.0 | 6.7 | 100.0 | 17.5 | 100.0 | 17.5 |
| None | LS-960 | 0 | 16 | 100.0 | 4.8 | 100.0 | 5.1 | 100.0 | 13.5 | 100.0 | 13.7 |
| None | LS-960 | 0.1 | 16 | 100.0 | 4.9 | 100.0 | 5.1 | 100.0 | 13.4 | 100.0 | 13.5 |
| LS-100 | LS-860 | 0 | 8 | 6.2 | 3.3 | 6.7 | 3.4 | 19.1 | 9.4 | 19.5 | 9.0 |
| LS-100 | LS-860 | 0.0001 | 8 | 6.2 | 3.3 | 6.7 | 3.5 | 19.1 | 9.3 | 19.5 | 9.0 |
| LS-100 | LS-860 | 0 | 16 | 6.2 | 3.1 | 6.7 | 3.2 | 19.1 | 8.5 | 19.5 | 8.3 |
| LS-100 | LS-860 | 1.0 | 16 | 6.2 | 3.0 | 6.7 | 3.2 | 19.1 | 8.6 | 19.5 | 8.3 |

| Seed | Data | fedprox $\mu$ | Cohort size | Central LR | dev | | test | |
|---|---|---|---|---|---|---|---|---|
| | | | | | $T=0$ | $T=2$k | $T=0$ | $T=2$k |
| None | CV-en | 0 | 8 | 0.006 | 100.0 | 62.9 | 100.0 | 66.7 |
| None | CV-en | 0.01 | 8 | 0.006 | 100.0 | 63.4 | 100.0 | 67.4 |
| None | CV-en | 0 | 16 | 0.006 | 100.0 | 51.9 | 100.0 | 56.5 |
| None | CV-en | 0.0001 | 16 | 0.006 | 100.0 | 51.0 | 100.0 | 55.8 |
| None | CV-en | 0 | 32 | 0.006 | 100.0 | 41.3 | 100.0 | 46.3 |
| None | CV-en | 0.0001 | 32 | 0.006 | 100.0 | 40.0 | 100.0 | 44.9 |
| LS-100 | CV-en | 0 | 8 | 0.002 | 54.7 | 24.5 | 61.2 | 28.8 |
| LS-100 | CV-en | 0.1 | 8 | 0.002 | 54.7 | 24.3 | 61.2 | 28.7 |
| LS-100 | CV-en | 0 | 16 | 0.002 | 54.7 | 22.2 | 61.2 | 26.3 |
| LS-100 | CV-en | 1e-05 | 16 | 0.002 | 54.7 | 22.0 | 61.2 | 26.3 |
| LS-100 | CV-en | 0 | 32 | 0.002 | 54.7 | 20.1 | 61.2 | 23.9 |
| LS-100 | CV-en | 0.1 | 32 | 0.002 | 54.7 | 20.1 | 61.2 | 23.9 |

In one of the remaining cases there is no change and only in one case the results with FedProx are considerably worse than without it.

It is surprising how the optimal value of the key FedProx parameter $\mu$ changes considerably between the various scenarios. This suggests that it would make sense to evaluate adaptive $\mu$ as suggested by [11]. We leave the use of adaptive $\mu$ and the investigation of how FedProx may improve FL training robustness (e.g. with respect to the number of local epochs or steps) for future work.

We also tried limiting the number of batches processed on each client [12] and normalizing users' deltas sent to the server [13] but neither approach improved the results. See Table 6 in Appendix H.2 for the results on limiting the number of batches (steps) processed for each client.

### G.8 Extending the Number of Central FL Iterations

Table 14 shows that even though most FL models were stopped after 2k central steps, letting these models to train longer would further improve performance. However, due to the communication complexity for each central step, it is best to use a moderate number of central steps and maximize utility of the training by optimizing the parameters for local, on-device training, cohort sizes, and other key FL parameters.

### G.9 Impact of Under-Trained Seed Models

Table 15 shows that choosing a better seed model improves performance across the board. Furthermore, the results presented previously in Table 4 show that using a seed model trained on more data improves FL performance, even if the data used to train seed models are from a different domain.

Table 14: Results (WER, %) on selected FL configurations on CV obtained after $T = 4$k central steps and their comparison to those obtained after $T = 2$k central steps. All parameters are identical to those in Table 4.

| Seed | Data | Cohort size | dev | | | test | | |
|---|---|---|---|---|---|---|---|---|
| | | | $T = 0$ | $T = 2$k | $T = 4$k | $T = 0$ | $T = 2$k | $T = 4$k |
| None | CV-en | 16 | 100.0 | 51.9 | 43.3 | 100.0 | 56.5 | 48.3 |
| None | CV-en | 32 | 100.0 | 41.3 | 34.0 | 100.0 | 46.3 | 38.9 |
| None | CV-en | 64 | 100.0 | 32.9 | 27.3 | 100.0 | 38.0 | 32.0 |
| CV-en-10 | CV-en-90 | 16 | 23.0 | 18.9 | 17.8 | 27.9 | 22.8 | 21.4 |
| CV-en-10 | CV-en-90 | 32 | 23.0 | 17.7 | 16.9 | 27.9 | 21.5 | 20.4 |
| CV-en-10 | CV-en-90 | 64 | 23.0 | 16.7 | 16.0 | 27.9 | 20.1 | 19.4 |
| LS-100 | CV-en | 16 | 54.7 | 22.2 | 19.9 | 61.2 | 26.3 | 23.7 |
| LS-100 | CV-en | 32 | 54.7 | 20.1 | 18.2 | 61.2 | 23.9 | 21.8 |
| LS-100 | CV-en | 64 | 54.7 | 18.4 | 16.8 | 61.2 | 22.0 | 20.2 |

Table 15: Impact of under-trained seed models on WER of the final model for CV dataset with *LS-100* seed and cohort size of 32. The under-trained seed models are obtained from the first 70 steps of the baseline central training used to generate the actual seed model. The parameters for the experiments without seed models and the one with the high quality seed model are the same as in Table 4. The parameters for the seeds of lower quality are the same as those without a seed model.

| Seed | dev | | test | |
|---|---|---|---|---|
| | $T = 0$ | $T = 2$k | $T = 0$ | $T = 2$k |
| None | 100.0 | 39.9 | 100.0 | 44.7 |
| LS-100 (30 steps) | 98.9 | 37.7 | 100.0 | 42.8 |
| LS-100 (50 steps) | 83.2 | 32.8 | 87.8 | 37.8 |
| LS-100 (70 steps) | 75.9 | 33.3 | 81.1 | 38.2 |
| LS-100 (full) | 54.7 | 20.1 | 61.2 | 23.9 |

Table 16: Results (WER %) on CV for different cohort sizes. We use exponential decay for central LR starting at $t = 750$, decay rate 0.6, and transition steps 750 with $T = 2$k total central steps. Local (central) LR is 0.2 (0.002). All models are trained with the same hyper-parameters, only the cohort size is varied. for left half of Table with cohort size from 8 to 256 we use 10 local epochs, while for the right half of the Table we use 10 local steps to scale efficiently on GPU to 256-5120 cohort sizes. Central models are trained either with the batch discussed in Section F or with 3x batch size, shown in brackets (all other hyper-parameters are the same as in Section F).

| Seed | Data | Eval. | seed WER | cohort size WER | | | | | | | | | | | | | central WER |
|---|---|---|---|---|---|---|---|---|---|---|---|---|---|---|---|---|---|
| | | | | 8 | 16 | 32 | 64 | 128 | 256 | 256 | 512 | 1024 | 2048 | 3072 | 4096 | 5120 | |
| LS-100 | CV-en | dev | 54.7 | 24.5 | 22.2 | 20.1 | 18.4 | 16.8 | 15.6 | 15.6 | 16.8 | 15.7 | 14.9 | 14.5 | 14.3 | 14.1 | 14.7 (12.7) |
| | | test | 61.2 | 28.8 | 26.3 | 23.9 | 22.0 | 20.2 | 18.9 | 18.6 | 20.0 | 18.9 | 17.8 | 17.4 | 17.2 | 16.9 | 17.8 (15.6) |
| LS-960 | CV-en | dev | 27.0 | 19.7 | 18.1 | 16.9 | 15.6 | 14.5 | 13.7 | 18.0 | 14.6 | 13.9 | 13.6 | 13.0 | 12.8 | 12.7 | 14.1 (12.0) |
| | | test | 31.5 | 23.5 | 21.6 | 20.2 | 18.8 | 17.6 | 16.6 | 21.5 | 17.6 | 16.7 | 16.3 | 15.7 | 15.5 | 15.4 | 17.2 (14.8) |

## G.10 Scaling to Larger Cohorts

We further scale the cohort size to check limitations on the cohort size and scaling of FL: for efficient GPU utilization we switch from 10 local epochs to 10 local steps for cohort sizes of 256 to 5120, while keeping all other hyper-parameters the same. Results are shown in Table 16: FL scales to larger cohort sizes lowering further WER. There is also observed degradation by switching from local epochs to local steps especially for a stronger seed model, likely due to overfitting to the seed model's data, which are out-of-domain data with respect to the FL data.

## H Empirical Analysis: Federated Learning with Differential Privacy

### H.1 Differential Privacy Noise Discussion

There are different equivalent formulations how the noise can be added to the clients' deltas to introduce DP, which can cause confusion about the noise scale, and how the moments accountant is applied. Having Algorithm 1, step 12 can be defined as follows:

1. Noise is added on the client level: $\boldsymbol{\Delta}^t = 1/q \sum_{k \in \mathcal{K}^t} \omega_k \left[ \boldsymbol{\Delta}_k^t + \mathcal{N}(0, IC^2\sigma_{client}^2) \right]$. We use this definition with $\sigma_{client} = \sigma \cdot \sqrt{\frac{q}{\sum_{k=1}^{K} \omega_k^2}}$. It was also used by [55].

2. Noise is added on the server level after averaging clients' deltas: $\boldsymbol{\Delta}^t = 1/q \left[ \sum_{k \in \mathcal{K}^t} \omega_k \boldsymbol{\Delta}_k^t \right] + \mathcal{N}(0, IC^2\sigma_{avg}^2)$. This is the definition used by [46].

3. Noise is added on the server level after summation but before normalization to the number of clients (used by [8]): $\boldsymbol{\Delta}^t = 1/(qK) \left[ \left( \sum_{k \in \mathcal{K}^t} K\omega_k \boldsymbol{\Delta}_k^t \right) + \mathcal{N}(0, IC^2\sigma_{sum}^2) \right]$.

Different variants of noise are connected with each other via $\sigma_{sum} = \sigma_{avg} \cdot qK = \sigma_{avg} \cdot S$ and $\sigma_{client} = \sigma_{avg} \cdot \sqrt{\frac{q}{\sum_{k=1}^{K} \omega_k^2}}$. Then we can compute that $\sigma_{DP} = \sigma_{avg}$ in this notation from Algorithm 1.

Throughout the paper we use $\omega_k = \frac{1}{K}$ and the moments accountant implementation from `opacus` [99] which works with $\sigma_{sum}$ noise definition. To re-scale noise added to each client in order to be consistent with `opacus`, we re-scale it by multiplying by the cohort size $S$. Thus, we get Theorem 1 where $z$ is defined as $z = \sigma_{sum}$ and, finally, we get the bound on $\sigma_{DP}$ via Theorem 1 from [8] which gives bound on $z^2 \geq const \frac{q^2 T \log 1/\delta}{\varepsilon^2}$ in our notation. In all experiments with FL with DP, we use the same privacy budget for every training step.

### H.2 Large Cohort Training Implementation

Our initial FL implementation processed the clients in each cohort sequentially, potentially parallelizing the training for each client using multiple GPUs. For each client, we train a local model for a given number of epochs. However, this approach does not scale well to training with large cohorts, e.g. 1,024, which were necessary for experiments with FL with DP.

That is why we implemented another version where every client is trained on 1 GPU and we train the models for several clients in parallel utilizing all available GPUs (e.g. with 32 GPUs we can process 32 clients in parallel). To do that efficiently with highly imbalanced data like CV where some clients have much more data than others, we restrict the training on every client to a pre-defined number of training steps (batches processed) instead of epochs. Switching from a fixed number of epochs to a fixed number of training steps per client was previously reported to improve performance in the presence of data heterogeneity [12].

Since we always use dynamic batching for efficient implementation and the average number of minutes of audio per client in CV is 2.5, FL training with 10 local epochs and total dynamic batch of 2 minutes per client can be approximated with 10 local steps and the same batch size. This configuration is used in all FL with DP experiments.

Unlike reported by [12], we did not observe improved performance after switching to the number of local steps but instead observed degradation in performance: see Figure 6 for the results on one configuration of CV with *LS-100* seed model. However, it is of note that the differences will likely get smaller with larger cohort sizes. From results in Figure 6, we expect that more local compute that would be feasible in a real deployment, should lead to better results than what we get in our experiments for FL with DP.

### H.3 Empirical Analysis

For FL training with the large cohort size of 1,024, the client delta norms are already bounded due to the local clipping (see Algorithm 1, line 10) done in each step of the local training for every client (see Figure 7). Local clipping is a necessary part of the training because otherwise the local training

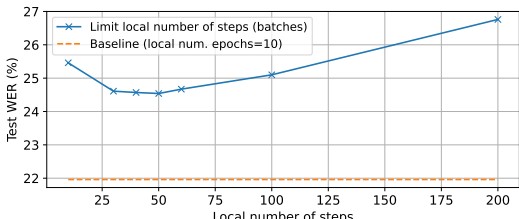

Figure 6: Comparison of WER for FL training between local number of steps (solid) and local number of epochs (dashed). Training is done on *CV-en-train* with a seed model pre-trained centrally on *LS-100*. The cohort size is $S = 64$, total number of central steps is $T = 2k$, and all other parameters are set the same as in the corresponding configuration in Figure 3.

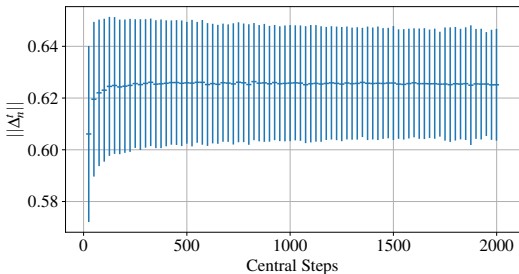

Figure 7: Client's delta norm averaged per clients throughout FL training with the cohort size of $S = 1,024$ on *CV-en-train* from a seed model trained on *LS-100*. We use exponential decay for central LR starting at $t = 750$, decay rate 0.6, and transition steps 750 with $T = 2k$ total central steps and 10 local steps. Local (central) LR is 0.2 (0.002).

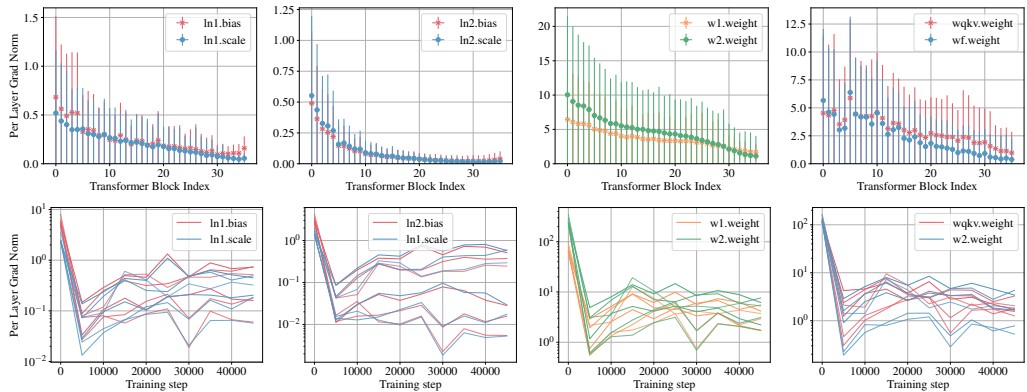

Figure 8: Central training from scratch on *CV-en-train* and its per layer gradients norm: (top) averaged across training steps and (bottom) showed per layer along the training. The model is trained with LARS optimizer and the learning rate of 0.5. The norms of the per-layer gradients are balanced differently compared to models trained with FL or with FL and DP in Figure 10, e.g., LayerNorm gradients do not dominate over MLP and attention gradients.

of the transformer model would not converge [31, 30]. This is similar to the standard recipe for the central training of transformer models.

As discussed in Section 4.3, we varied the clipping bound $C$ for clients' deltas and did not observe any impact of it on the final performance even when $C = 10^{-8}$. We also did not observe the difference between training with the full precision (float32) or training with the reduced precision (bloat16). The LAMB optimizer's $\xi = 10^{-6}$, thus it was a leading term in the denominator during optimization when clipping $C < 10^{-6}$.

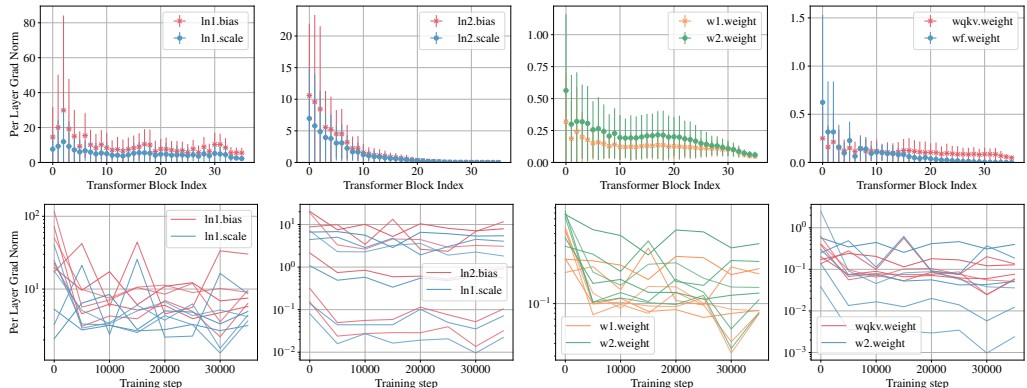

Figure 9: Central training on *CV-en-train* from the *LS-100* seed model and its per layer gradients norm: (top) averaged across training steps and (bottom) showed per layer along the training. The model is trained with LARS optimizer and the learning rate of 0.5. The norms of the per-layer gradients are balanced similarly to models trained with FL or with FL and DP in Figure 10: LayerNorm gradients do dominate over MLP and attention gradients.

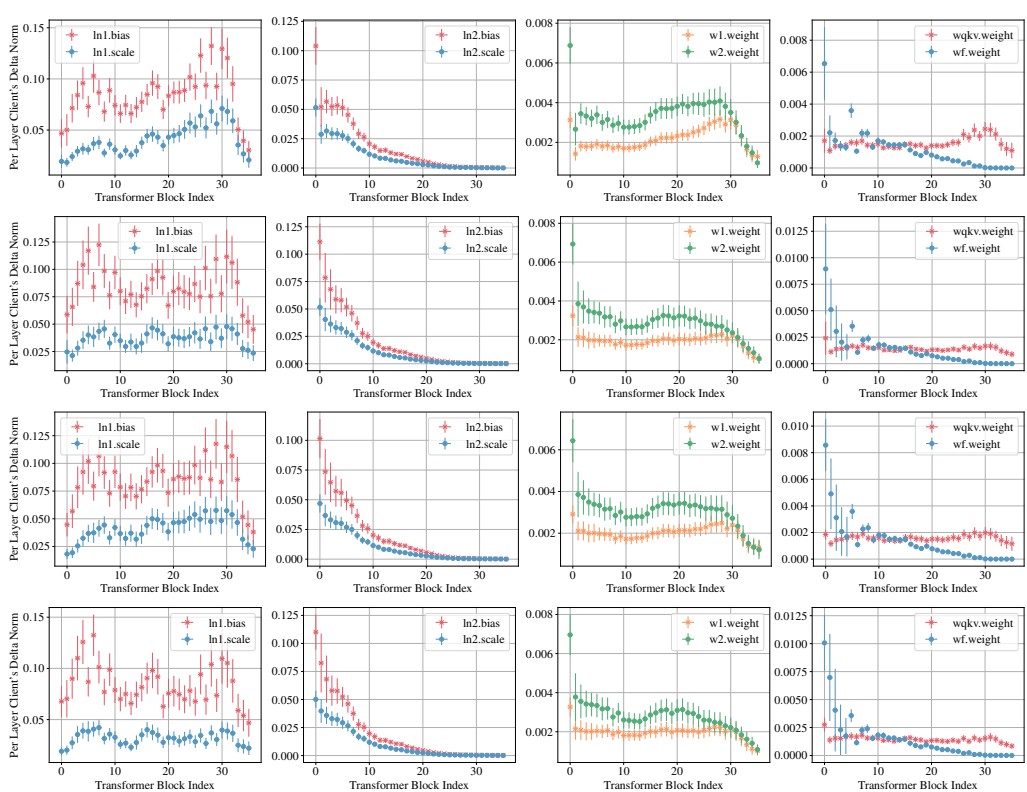

Figure 10: Client delta norms computed per layer in the model. We average the statistics across all clients and central steps, and plot the mean and standard deviation. The model is trained with (first row) global clients' deltas clipping $C = 10^{-2}$ and $\sigma_{\mathrm{DP}} = 0$, (second row) global clients' deltas clipping $C = 10^{-6}$ and $\sigma_{\mathrm{DP}} = 0$, (third row) per-layer clients' deltas clipping (Definition 3, "uniform") $C = 10^{-3}$ and $\sigma_{\mathrm{DP}} = 0$, and (fourth row) per-layer clients' deltas clipping (Definition 3, "uniform") $C = 10^{-2}$ and $\sigma_{\mathrm{DP}} = 3 \cdot 10^{-6}$. The rest of the training configuration is the same as in Figure 5. A transformer block consists of attention parameters (wqkv and wf), MLP (w1 and w2), LayerNorm applied to input of attention (ln1) or MLP (ln2). The statistics are consistent with the training with global clipping (Algorithm 1) in Figure 5.

We assume that the seed model is trained centrally without DP[8] (e.g. *LS-100*) after which FL with DP is run on *CV-en-train* by initializing FL model with the seed model. When we add DP noise to the training alongside with the clipping of clients' deltas, we also did not observe any difference in the training dynamic and final performance (WER) as long as $C\sigma_{\text{DP}}$ remained constant (e.g., halving the clipping bound $C$ and halving the noise $\sigma_{\text{DP}}$ would produce a nearly identical model). We hypothesise that this is the outcome of (i) above observation that clipping does not affect training; and (ii) using LAMB as a central optimizer, which performs LARS per-layer scaling, and scales both the noise as well as the signal in the same way.

As discussed in Section 4.3, we observe clients' deltas imbalance across different layers of the transformer model (see Figure 5). The first layers (1-10 transformer blocks) have higher delta norms than the last layers (20-36 transformer blocks) for LayerNorm in MLP part and attention final linear projection. This is the opposite behaviour than observed in the deep models, e.g. by [100]. Also, LayerNorms in general have an order of magnitude larger clients' deltas norms than those for MLP and attention. We checked if FL is the source of this deltas imbalance by looking into central training. Central training from scratch on *CV-en-train*, Figure 8, has per layer gradients that behave differently from the clients' deltas in FL or FL with DP training. However, when we compare central training on *CV-en-train* from the same *LS-100* seed model, we will see that per layer gradients behave similarly to the clients' deltas in FL or FL with DP training (see Figure 9).

The smallest delta norms are still non-zero and are order of $10^{-4}$ for LayerNorm (ln2) and $10^{-6}$ for attention (wf) which are re-scaled later by LAMB central optimizer to have the same gradient magnitude across layers. This also highlights necessity of using adaptive optimizers on the server side because otherwise a part of the network will not be trained at all. A similar behavior to the one from Figure 5 can be observed (i) with or without DP noise; and (ii) with global clipping or per-layer clipping of clients' deltas (see Figure 10).

## H.4    Detailed Results

Comparison for both loss and word error rate (WER) for different values of DP noise and global vs "uniform" per-layer clipping is given in Figure 11, and comparison between "uniform" and "dim" per-layer clipping is given in Figure 12. Training dynamic is shown in Figure 13 for global clipping and in Figure 14 for per-layer clipping. For the per-layer clipping setting we can increase DP noise till $\sigma_{\text{DP}} = 100 \cdot 10^{-6}$ and get similar performance as with global clipping but DP noise $\sigma_{\text{DP}} = 3 \cdot 10^{-6}$. The former is preferable as it has better $(\varepsilon, \delta)$-DP guarantees, detailed results of which are shown in Table 17.

## H.5    Per-Layer Clipping Analysis

To understand which part of the transformer is most affected by DP noise, we train a model by adding DP noise only to a particular group of parameters for both global clipping and per-layer "uniform" clipping (see Figure 15): in this case DP guarantees *do not hold*, however we do this for the sake of analysis. We can see that adding DP noise to the parameters of MLP layers drastically reduces model performance, while adding it to other parameters changes WER of the model only marginally. This holds for both types of clipping we apply on clients' deltas.

As per-layer clipping "dim" performed the best in our experiments (see Table 1), we analyse the effect of DP noise for this configuration in depth in Figure 16. First, the results are consistent with Figure 15 in that MLP layers are the most susceptible parts of the transformer, e.g. even if we add DP noise to all layers except MLP ones, we see only small degradation in model performance (middle plot in Figure 15). Second, if we add DP noise with $\sigma_{\text{DP}}$ to all layers but MLP layers get DP noise with $\sigma_{\text{DP}}/2$, we see a significant improvement in the model performance (right plot in Figure 15). The latter suggests that we could redistribute the clipping budget across layers to further alleviate the effect of DP noise during training.

Further experiments with per-layer clipping as $C_i = C\sqrt{\frac{\alpha_i d_i}{\sum_{h=1}^{H} \alpha_h d_h}}$ where $d_i$ is the dimension of the $i$-th layer, $i = 1, \ldots, H$, and $\alpha_i = 1$ for all layers except MLP and $\alpha_i = \beta$ for all MLP layers with $\beta \in \{1.5, 2, 3, 10\}$ did not improve results.

---

[8]We presume that these data are either public or do not require privacy protection.

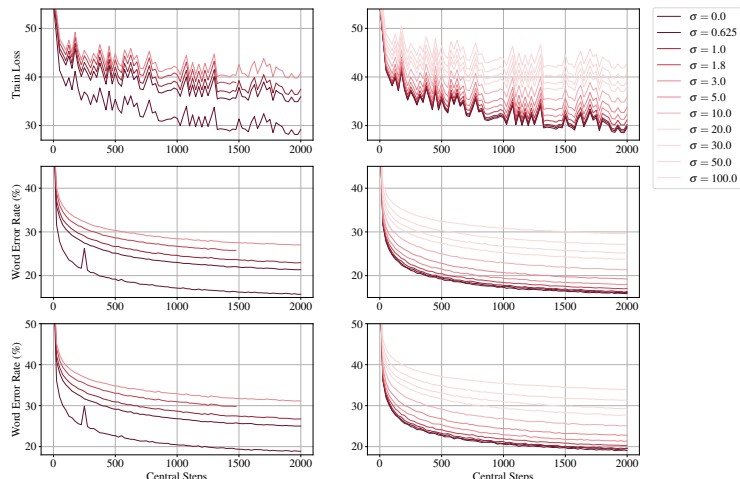

Figure 11: Loss (top) and word error rate (WER) measured on *CV-en-dev* (middle) and *CV-en-test* (bottom) sets for different values of DP noise $\sigma_{\mathrm{DP}}$ (scale is set to $10^{-6}$). We apply clipping of $10^{-2}$ either globally (left, Algorithm 1) or per-layer (right, Definition 3, "uniform") with $T = 2$k central steps and $L = 1{,}024$ cohort size. The rest of the training configuration is the same as in Figure 7. The seed model is trained on *LS-100*.

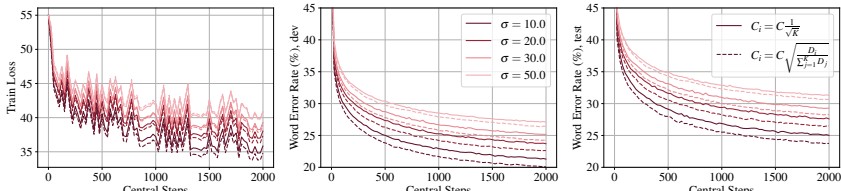

Figure 12: Loss (left) and word error rate (WER) measured on *CV-en-dev* (middle) and *CV-en-test* (right) sets for different values of DP noise $\sigma_{\mathrm{DP}}$ (scale is set to $10^{-6}$). We apply clipping of $10^{-2}$ per-layer (Definition 3, "uniform" and "dim") with $T = 2$k central steps and $S = 1{,}024$ cohort size. The rest of the training configuration is the same as in Figure 7. The seed model is trained on *LS-100*.

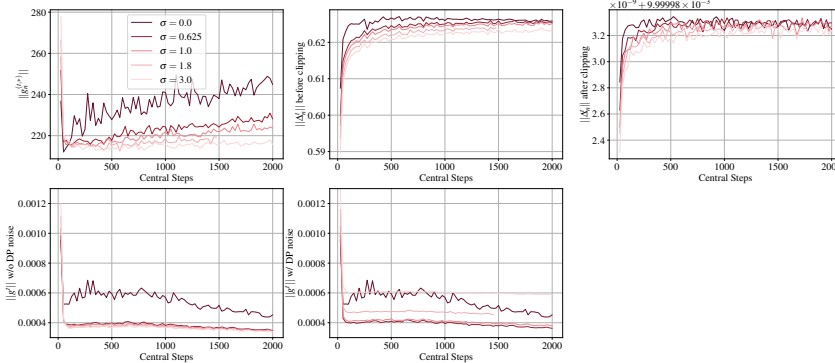

Figure 13: Training dynamic of models from Figure 11 with different DP noise $\sigma_{\mathrm{DP}}$ (scale is set to $10^{-6}$), global clipping of $10^{-2}$ and $T = 2$k central steps. The seed model is trained on *LS-100*: (top, left) client gradients norm during local training (averaged across clients in the cohort); (top, middle) client's delta norm before clipping; (top, right) client's delta norm after clipping; (bottom, left) server gradients norm before DP noise is added per clients' deltas; (bottom, middle) server gradients norm after DP noise is added per clients' deltas.

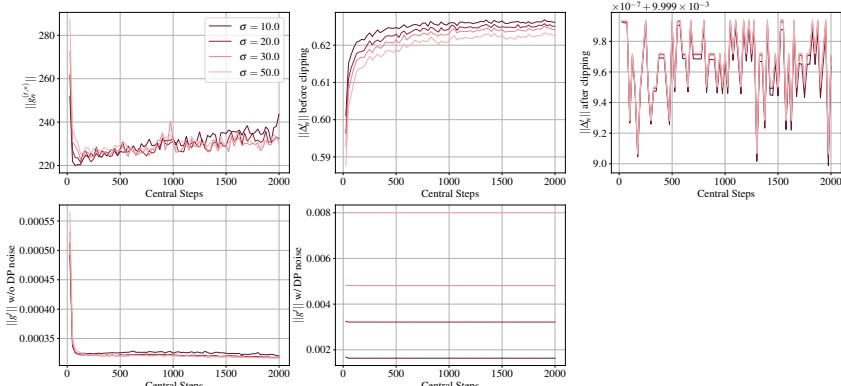

Figure 14: Training dynamic of models from Figure 12 with different DP noise $\sigma_{\mathrm{DP}}$ (scale is set to $10^{-6}$), per-layer clipping of $10^{-2}$ (Definition 3, "dim") and $T = 2\mathrm{k}$ central steps. The seed model is trained on *LS-100*: (top, left) client gradients norm during local training (averaged across clients in the cohort); (top, middle) client's delta norm before clipping; (top, right) client's delta norm after clipping; (bottom, left) server gradients norm before DP noise is added per clients' deltas; (bottom, middle) server gradients norm after DP noise is added per clients' deltas.

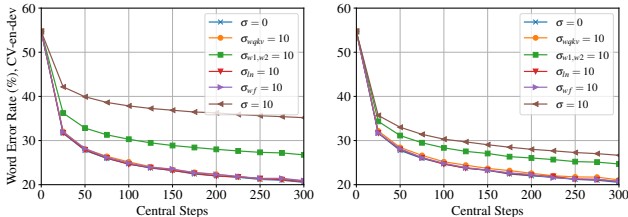

Figure 15: WER of models trained on *CV-en-train* and evaluated on *CV-en-dev* for different values of DP noise $\sigma_{\mathrm{DP}}$ (scale is set to $10^{-6}$). We add either DP noise to all parameters in the model ($\sigma_{\mathrm{DP}} = 10$), or no DP noise ($\sigma_{\mathrm{DP}} = 0$), or DP noise to the specific group of parameters: to attention ($\sigma_{\mathrm{DP},wqkv} = 10$), to MLP ($\sigma_{\mathrm{DP},w1,w2} = 10$), to LayerNorms ($\sigma_{\mathrm{DP},ln} = 10$), to attention final projection ($\sigma_{\mathrm{DP},wf} = 10$). We apply clipping of $10^{-2}$ either globally (left, Algorithm 1) or per-layer (right, Definition 3, "uniform") with $T = 2\mathrm{k}$ central steps and $S = 1{,}024$ cohort size. The rest of the training configuration is the same as in Figure 7. The seed model is trained on *LS-100*.

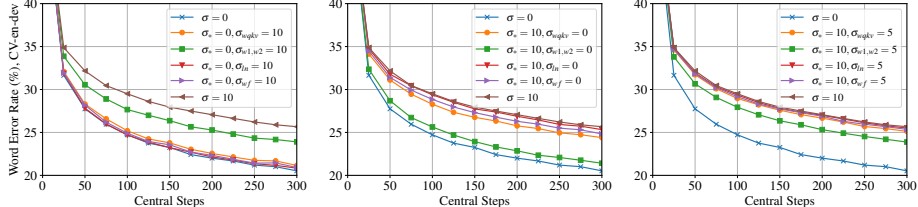

Figure 16: WER of models trained on *CV-en-train* and evaluated on *CV-en-dev* for different values of DP noise $\sigma_{\mathrm{DP}}$ (scale is set to $10^{-6}$). We apply per-layer clipping of $10^{-2}$ (Definition 3, "dim") with $T = 2\mathrm{k}$ central steps and $S = 1{,}024$ cohort size. The rest of the training configuration is the same as in Figure 7. The seed model is trained on *LS-100*. We add either DP noise to all parameters in the model ($\sigma_{\mathrm{DP}} = 10$), or no DP noise ($\sigma_{\mathrm{DP}} = 0$). We also add DP noise (left) to the specific group of parameters only; (middle) to all parameters except the specific group of parameters; (right) to all parameters but the DP noise with $\sigma_{\mathrm{DP}}/2 = 5$ to the specific group of parameters.

## H.6 Federated Learning with Differential Privacy for French and German

We run out of the box experiments for FL with DP for French and German CV data using the same configuration as for English (training parameters are given in Figure 7). Seed models are trained

Table 17: Extended results of Table 1 for FL with DP and a model pre-trained on *LS-100* (∼100h) used as central data and afterwards fine-tuned on *CV-en-train* (∼1.6k hours) used as clients data. We report added noise $\mathcal{N}(0, IC^2\sigma_{\mathrm{DP}}^2 qK)$ per client and CV dev and test WERs (%) for two clipping variants with clipping bound $C$: global and per layer "uniform" ("dim"). Total number of users $K$, expected number of users sampled per central step $S = qK$, and the number of central steps $T$ are given. We set $\delta = 10^{-9}$ and report $\varepsilon$ for which $(\varepsilon, \delta)$-DP holds for a given $S$ and $K$ using the moments accountant of [8]. For scaling $S$ and $K$ where it is practically intractable to run model training (marked "-"), we extrapolate $(\varepsilon, \delta)$-DP assuming training dynamic remains unchanged thus similar WER will be obtained. Central training gives 14.7%/17.8% WER on dev/test. $\varepsilon$ should be below 10 to be practically useful (marked with blue).

| $z$ | $\sigma_{\mathrm{DP}}(\cdot 10^{-6})$ | $C$ | $S$ | $K$ | $q = S/K$ | $T$ | $\varepsilon$ | order | global clipping dev WER (%) | test WER (%) | per-layer clipping dev WER (%) | test WER (%) |
|---|---|---|---|---|---|---|---|---|---|---|---|---|
| - | - | - | 0 | 34,753 | 0 | 0 | 0 | - | 54.7 | 61.2 | 54.7 | 61.2 |
| 0.1024 | 100.0 | 0.01 | 1,024 | 34,753 | 0.0295 | 2,006 | $3.3\cdot10^4$ | 1.1 | - | - | 29.6 | 33.9 |
| 1.024 | 100.0 | 0.01 | 10,240 | 347,530 | 0.0295 | 2,006 | $1.3\cdot10^1$ | 4.0 | - | - | - | - |
| 5.12 | 100.0 | 0.01 | 51,200 | 1,737,650 | 0.0295 | 2,006 | $1.6\cdot10^0$ | 25 | - | - | - | - |
| 0.0512 | 50.0 | 0.01 | 1,024 | 34,753 | 0.0295 | 2,006 | $3.5\cdot10^5$ | 1.1 | - | - | 27.1 (26.4) | 31.3 (30.6) |
| 0.512 | 50.0 | 0.01 | 10,240 | 347,530 | 0.0295 | 2,006 | $7.2\cdot10^1$ | 1.5 | - | - | - | - |
| 2.56 | 50.0 | 0.01 | 51,200 | 1,737,650 | 0.0295 | 2,006 | $3.5\cdot10^0$ | 10.0 | - | - | - | - |
| 0.03072 | 30.0 | 0.01 | 1,024 | 34,753 | 0.0295 | 2,006 | $1.1\cdot10^6$ | 1.1 | - | - | 25.2 (24.2) | 29.3 (28.2) |
| 0.3072 | 30.0 | 0.01 | 10,240 | 347,530 | 0.0295 | 2,006 | $3.7\cdot10^2$ | 1.1 | - | - | - | - |
| 1.536 | 30.0 | 0.01 | 51,200 | 1,737,650 | 0.0295 | 2,006 | $6.5\cdot10^0$ | 7.0 | - | - | - | - |
| 0.02048 | 20.0 | 0.01 | 1,024 | 34,753 | 0.0295 | 2,006 | $2.6\cdot10^6$ | 1.1 | - | - | 23.7 (22.6) | 27.6 (26.5) |
| 1.024 | 20.0 | 0.01 | 51,200 | 1,737,650 | 0.0295 | 2,006 | $1.3\cdot10^0$ | 4.0 | - | - | - | - |
| 2.048 | 20.0 | 0.01 | 102,400 | 3,475,300 | 0.0295 | 2,006 | $4.5\cdot10^0$ | 9.0 | - | - | - | - |
| 0.01024 | 10.0 | 0.01 | 1,024 | 34,753 | 0.0295 | 2,006 | $1.1\cdot10^7$ | 1.1 | 30.7 | 35.2 | 21.3 (20.1) | 25.0 (23.7) |
| 0.512 | 10.0 | 0.01 | 51,200 | 1,737,650 | 0.0295 | 2,006 | $7.2\cdot10^1$ | 1.5 | - | - | - | - |
| 0.512 | 10.0 | 0.01 | 51,200 | 17,376,500 | 0.00295 | 2,034 | $1.3\cdot10^1$ | 3.0 | - | - | - | - |
| 1.024 | 10.0 | 0.01 | 102,400 | 3,475,300 | 0.0295 | 2,006 | $1.3\cdot10^1$ | 4.0 | - | - | - | - |
| 2.048 | 10.0 | 0.01 | 204,800 | 6,950,600 | 0.0295 | 2,006 | $4.5\cdot10^0$ | 9.0 | - | - | - | - |
| 2.048 | 10.0 | 0.01 | 204,800 | 69,506,000 | 0.00295 | 2,006 | $7.5\cdot10^{-1}$ | 25.0 | - | - | - | - |
| 0.00512 | 5.0 | 0.01 | 1,024 | 34,753 | 0.0295 | 2,006 | $4.2\cdot10^7$ | 1.1 | - | - | 19.2 | 22.7 |
| 0.512 | 5.0 | 0.01 | 102,400 | 3,475,300 | 0.0295 | 2,006 | $7.2\cdot10^1$ | 1.5 | - | - | - | - |
| 1.024 | 5.0 | 0.01 | 204,800 | 6,950,600 | 0.0295 | 2,006 | $1.3\cdot10^1$ | 4.0 | - | - | - | - |
| 1.024 | 5.0 | 0.01 | 204,800 | 69,506,000 | 0.00295 | 2,034 | $2.1\cdot10^0$ | 10.0 | - | - | - | - |
| 1.024 | 5.0 | 0.01 | 204,800 | 695,060,000 | 0.000295 | 3,390 | $1.2\cdot10^0$ | 15.0 | - | - | - | - |
| 0.003072 | 3.0 | 0.01 | 1,024 | 34,753 | 0.0295 | 2,006 | $1.2\cdot10^8$ | 1.1 | 27.0 | 31.1 | 17.9 (17.1) | 21.2 (20.4) |
| 0.3072 | 3.0 | 0.01 | 102,400 | 3,475,300 | 0.0295 | 2,006 | $3.7\cdot10^2$ | 1.1 | - | - | - | - |
| 0.6144 | 3.0 | 0.01 | 204,800 | 6,950,600 | 0.0295 | 2,006 | $4.2\cdot10^1$ | 2.0 | - | - | - | - |
| 0.6144 | 3.0 | 0.01 | 204,800 | 69,506,000 | 0.00295 | 2,034 | $7.2\cdot10^0$ | 3.0 | - | - | - | - |
| 0.6144 | 3.0 | 0.01 | 204,800 | 695,060,000 | 0.000295 | 3,390 | $3.7\cdot10^0$ | 6.0 | - | - | - | - |
| 0.0018432 | 1.8 | 0.01 | 1,024 | 34,753 | 0.0295 | 2,006 | $4.5\cdot10^8$ | 1.5 | 25.8 | 29.2 | 17.0 | 20.2 |
| 0.18432 | 1.8 | 0.01 | 102,400 | 3,475,300 | 0.0295 | 2,006 | $2.3\cdot10^4$ | 1.5 | - | - | - | - |
| 0.36864 | 1.8 | 0.01 | 204,800 | 6,950,600 | 0.0295 | 2,006 | $2.7\cdot10^2$ | 1.5 | - | - | - | - |
| 0.36864 | 1.8 | 0.01 | 204,800 | 69,506,000 | 0.00295 | 2,034 | $4.5\cdot10^1$ | 1.5 | - | - | - | - |
| 0.36864 | 1.8 | 0.01 | 204,800 | 695,060,000 | 0.000295 | 3,390 | $1.6\cdot10^1$ | 2.5 | - | - | - | - |
| 0.001024 | 1.0 | 0.01 | 1,024 | 34,753 | 0.0295 | 2,006 | $1.1\cdot10^9$ | 1.1 | 22.9 | 26.7 | 16.2 (16.0) | 19.5 (19.3) |
| 0.1024 | 1.0 | 0.01 | 102,400 | 3,475,300 | 0.0295 | 2,006 | $3.2\cdot10^4$ | 1.1 | - | - | - | - |
| 0.2048 | 1.0 | 0.01 | 204,800 | 6,950,600 | 0.0295 | 2,006 | $1.1\cdot10^3$ | 1.1 | - | - | - | - |
| 0.2048 | 1.0 | 0.01 | 204,800 | 69,506,000 | 0.00295 | 2,034 | $2.7\cdot10^2$ | 1.1 | - | - | - | - |
| 0.2048 | 1.0 | 0.01 | 204,800 | 695,060,000 | 0.000295 | 3,390 | $9.4\cdot10^1$ | 1.3 | - | - | - | - |
| 0.0006144 | 0.625 | 0.01 | 1,024 | 34,753 | 0.0295 | 2,006 | $4.0\cdot10^9$ | 1.5 | 21.3 | 25.0 | 16.1 | 19.3 |
| 0.06144 | 0.625 | 0.01 | 102,400 | 3,475,300 | 0.0295 | 2,006 | $3.8\cdot10^5$ | 1.5 | - | - | - | - |
| 0.12288 | 0.625 | 0.01 | 204,800 | 6,950,600 | 0.0295 | 2,006 | $7.9\cdot10^4$ | 1.5 | - | - | - | - |
| - | 0 | 0.001 | 1,024 | 34,753 | 0.0295 | 2,000 | inf | - | 15.7 | 18.9 | 15.9 | 19.1 |
| - | 0 | 0.01 | 1,024 | 34,753 | 0.0295 | 2,000 | inf | - | 15.7 | 18.9 | 15.9 | 19.1 |
| - | 0 | 0.1 | 1,024 | 34,753 | 0.0295 | 2,000 | inf | - | 15.7 | 18.9 | 15.7 | 19.0 |
| - | 0 | 1.0 | 1,024 | 34,753 | 0.0295 | 2,000 | inf | - | 15.7 | 18.9 | 15.7 | 18.9 |

on *CV-fr-train-10* and *CV-de-train-10*, while *CV-fr-train-90* and *CV-de-train-90* are used for further FL with DP training. We get similar results as for English, see Table 18. With the same DP noise $\sigma_{\mathrm{DP}} = 3 \cdot 10^{-6}$, we are able to closely match the model trained without DP noise ($\sigma_{\mathrm{DP}} = 0$) with only a small WER degradation: (i) for French from 15.2% to 16.0% WER while guaranteeing $(5.5, 10^{-9})$-DP, and (ii) for German from 11.0% to 12.0% WER while guaranteeing $(5.4, 10^{-9})$-DP; assuming the training effectiveness remains the same if we extrapolate to ∼50M clients with the cohort size of ∼250k. Moreover, we can also increase DP noise to $\sigma_{\mathrm{DP}} = 10^{-5}$, getting 17.9% WER with $(1.9, 10^{-9})$-DP for French and 13.9% WER with $(1.8, 10^{-9})$-DP for German by scaling only to ∼16M clients with the cohort size of ∼250k, assuming the training effectiveness remains the same. The latter is a realistic scenario for mid/low resource languages.

For both French and German we observe that per-layer clipping is not as effective as for English and we get only marginal improvements over global clipping. We have checked that the seed model quality and the seed model being out-of-domain are the not the sources of this discrepancy in results

Table 18: Results for FL with DP and a model pre-trained on *CV-fr-train-10/CV-de-train-10* ($\sim$50h) used as central data and afterwards fine-tuned on (top/bottom) *CV-fr-train-90/CV-de-train-90* ($\sim$700-800 hours) used as clients data. We report added noise $\mathcal{N}(0, IC^2\sigma_{DP}^2qK)$ per client and CV dev and test WERs (%) for two clipping variants with clipping bound $C$: global and per layer "dim". Total number of users $K$, expected number of users sampled per central step $S = qK$, and the number of central steps $T$ are given. We set $\delta = 10^{-9}$ and report $\varepsilon$ for which $(\varepsilon, \delta)$-DP holds for a given $S$ and $K$ using the moments accountant of [8]. For scaling $S$ and $K$ where it is practically intractable to run model training (marked "-"), we extrapolate $(\varepsilon, \delta)$-DP assuming training dynamic remains unchanged thus similar WER will be obtained. Central training gives 10.8%/12.6% WER for French and 8.1%/9.2% WER for German on dev/test. $\varepsilon$ should be below 10 to be practically useful (marked with blue).

| $z$ | $\sigma_{DP}(\cdot 10^{-6})$ | $C$ | $S$ | $K$ | $q = S/K$ | $T$ | $\varepsilon$ | order | global clipping | | per-layer clipping "dim" | |
|---|---|---|---|---|---|---|---|---|---|---|---|---|
| | | | | | | | | | dev WER (%) | test WER (%) | dev WER (%) | test WER (%) |
| - | - | - | 0 | 6,171 | 0 | 0 | 0 | - | 24.0 | 27.5 | 24.0 | 27.5 |
| 0.01024 | 10.0 | 0.01 | 1,024 | 6,171 | 0.1660 | 2,002 | $1.1 \cdot 10^7$ | 1.3 | - | - | 15.6 | 17.9 |
| 2.56 | 10.0 | 0.01 | 256,000 | 1,542,750 | 0.1660 | 2,002 | $2.4 \cdot 10^1$ | 3.0 | - | - | - | - |
| 2.56 | 10.0 | 0.01 | 256,000 | 15,427,500 | 0.0166 | 2,013 | $1.9 \cdot 10^0$ | 20.0 | - | - | - | - |
| 0.003072 | 3.0 | 0.01 | 1,024 | 6,171 | 0.1660 | 2,002 | $1.2 \cdot 10^8$ | 1.1 | 14.1 | 16.2 | 13.9 | 16.0 |
| 0.768 | 3.0 | 0.01 | 256,000 | 1,542,750 | 0.1660 | 2,002 | $1.8 \cdot 10^2$ | 3.0 | - | - | - | - |
| 0.768 | 3.0 | 0.01 | 256,000 | 15,427,500 | 0.0166 | 2,013 | $1.4 \cdot 10^1$ | 3.0 | - | - | - | - |
| 0.768 | 3.0 | 0.01 | 256,000 | 46,282,500 | 0.00553 | 1,991 | $5.5 \cdot 10^0$ | 5.0 | - | - | - | - |
| - | 0 | 0.01 | 1,024 | 6,171 | 0.1660 | 2,000 | inf | - | 13.2 | 15.2 | 13.2 | 15.2 |
| - | - | - | 0 | 6,415 | 0 | 0 | 0 | - | 18.6 | 21.2 | 18.6 | 21.2 |
| 0.01024 | 10.0 | 0.01 | 1,024 | 6,415 | 0.1596 | 2,002 | $1.1 \cdot 10^7$ | 1.1 | - | - | 12.3 | 13.9 |
| 2.56 | 10.0 | 0.01 | 256,000 | 1,603,750 | 0.1596 | 2,002 | $2.3 \cdot 10^1$ | 3.0 | - | - | - | - |
| 2.56 | 10.0 | 0.01 | 256,000 | 16,037,500 | 0.01596 | 2,016 | $1.8 \cdot 10^0$ | 20.0 | - | - | - | - |
| 0.003072 | 3.0 | 0.01 | 1,024 | 6,415 | 0.1596 | 2,002 | $1.2 \cdot 10^8$ | 1.1 | 10.7 | 12.1 | 10.5 | 12.0 |
| 0.768 | 3.0 | 0.01 | 256,000 | 1,603,750 | 0.1596 | 2,002 | $1.7 \cdot 10^2$ | 1.5 | - | - | - | - |
| 0.768 | 3.0 | 0.01 | 256,000 | 16,037,500 | 0.01596 | 2,016 | $1.4 \cdot 10^1$ | 4.0 | - | - | - | - |
| 0.768 | 3.0 | 0.01 | 256,000 | 48,112,500 | 0.00532 | 2,068 | $5.4 \cdot 10^0$ | 5.0 | - | - | - | - |
| - | 0 | 0.01 | 1,024 | 6,415 | 0.1596 | 2,000 | inf | - | 9.7 | 11.0 | 9.7 | 11.0 |

Table 19: Ablation for FL with DP and a model pre-trained either on *LS-960/CV-en-train-10* used as central data and afterwards fine-tuned on (top/bottom) *CV-en-train/CV-en-train-90*. We report added noise $\mathcal{N}(0, IC^2\sigma_{DP}^2qK)$ per client and CV dev and test WERs (%) for two clipping variants with clipping bound $C$: global and per layer "dim". Total number of users $K$, expected number of users sampled per central step $S = qK$, and the number of central steps $T$ are given. Central training gives 14.1%/17.2% WER for training from *LS-960* seed and 14.5%/17.6% for training from *CV-en-train-10* seed on dev/test. All the remaining parameters are the same as in Table 17.

| Seed | Data | $\sigma_{DP}(\cdot 10^{-6})$ | $C$ | $S$ | $K$ | $q = S/K$ | $T$ | global clipping | | per-layer clipping "dim" | |
|---|---|---|---|---|---|---|---|---|---|---|---|
| | | | | | | | | dev WER (%) | test WER (%) | dev WER (%) | test WER (%) |
| LS-960 | - | - | - | - | 0 | 34,753 | 0 | 0 | 27.0 | 31.5 | 27.0 | 31.5 |
| LS-960 | CV-en-train | 30 | 0.01 | 256 | 34,753 | 0.0074 | 2000 | 22.5 | 26.1 | 18.7 | 22.2 |
| CV-10 | - | - | - | - | 0 | 34,753 | 0 | 0 | 23.0 | 27.9 | 23.0 | 27.9 |
| CV-10 | CV-en-train-90 | 30 | 0.01 | 256 | 31,278 | 0.0082 | 2000 | 20.8 | 25.1 | 18.7 | 22.6 |

between languages: if we change the seed model for English to a better out-of-domain *LS-960* seed or to a better in-domain *CV-en-train-10* seed, we still observe a drastic improvement from per-layer clipping compared to global clipping (see Tables 19 and 20, and Figure 17).

First, there is a discrepancy in gradients balance across layers for the central model training for English, French and German with *CV-\*-train-10* seed models. The training of the English model has the issue we discussed above that LayerNorms dominate the attention and MLP, which translates to the similar behavior for FL and FL with DP training. However, French and German models do not have the same imbalance issue as English and, moreover, similar behavior holds for the central training, FL and FL with DP for French and German (see Figures 18, 20, 19 and 21). We attribute the later to the properties of the languages as discussed in Appendix G.5.

One factor that we cannot exclude from the above analysis is the user sampling $q = S/K$, which is significantly higher for French and German (16%) than for English ($< 1\%$) due to a smaller number of speakers in the French and German datasets. Further investigation is needed to evaluate larger

Table 20: Results for FL with DP and a model pre-trained on *LS-960* (∼1000h) used as central data and afterwards fine-tuned on *CV-en-train* (∼1.6k hours) used as clients data. We report added noise $\mathcal{N}(0, IC^2\sigma_{\mathrm{DP}}^2 qK)$ per client and CV dev and test WERs (%) for two clipping variants with clipping bound $C$: global and per layer "dim". Total number of users $K$, expected number of users sampled per central step $S = qK$, and the number of central steps $T$ are given. We set $\delta = 10^{-9}$ and report $\varepsilon$ for which $(\varepsilon, \delta)$-DP holds for a given $S$ and $K$ using the moments accountant of [8]. For scaling $S$ and $K$ where it is practically intractable to run model training (marked "-"), we extrapolate $(\varepsilon, \delta)$-DP assuming training dynamic remains unchanged thus similar WER will be obtained. Central training gives 14.1%/17.2% WER on dev/test. $\varepsilon$ should be below 10 to be practically useful (marked with blue).

| $z$ | $\sigma_{\mathrm{DP}}(\cdot 10^{-6})$ | $C$ | $S$ | $K$ | $q = S/K$ | $T$ | $\varepsilon$ | order | global clipping | | per-layer clipping | |
|---|---|---|---|---|---|---|---|---|---|---|---|---|
| | | | | | | | | | dev WER (%) | test WER (%) | dev WER (%) | test WER (%) |
| - | - | - | 0 | 34,753 | 0 | 0 | 0 | - | 27.0 | 31.5 | 27.0 | 31.5 |
| 0.03072 | 30.0 | 0.01 | 1,024 | 34,753 | 0.0295 | 2,006 | $1.1\cdot10^6$ | 1.1 | 22.5 | 26.1 | 18.7 | 22.2 |
| 0.3072 | 30.0 | 0.01 | 10,240 | 347,530 | 0.0295 | 2,006 | $3.7\cdot10^2$ | 1.1 | - | - | - | - |
| 1.536 | 30.0 | 0.01 | 51,200 | 1,737,650 | 0.0295 | 2,006 | $6.5\cdot10^0$ | 7.0 | - | - | - | - |
| 0.01024 | 10.0 | 0.01 | 1,024 | 34,753 | 0.0295 | 2,006 | $1.1\cdot10^7$ | 1.1 | 20.5 | 24.1 | 16.5 | 19.7 |
| 0.512 | 10.0 | 0.01 | 51,200 | 1,737,650 | 0.0295 | 2,006 | $7.2\cdot10^1$ | 1.5 | - | - | - | - |
| 0.512 | 10.0 | 0.01 | 51,200 | 17,376,500 | 0.00295 | 2,034 | $1.3\cdot10^1$ | 3.0 | - | - | - | - |
| 1.024 | 10.0 | 0.01 | 102,400 | 3,475,300 | 0.0295 | 2,006 | $1.3\cdot10^1$ | 4.0 | - | - | - | - |
| 2.048 | 10.0 | 0.01 | 204,800 | 6,950,600 | 0.0295 | 2,006 | $4.5\cdot10^0$ | 9.0 | - | - | - | - |
| 2.048 | 10.0 | 0.01 | 204,800 | 69,506,000 | 0.00295 | 2,006 | $7.5\cdot10^{-1}$ | 25.0 | - | - | - | - |
| 0.003072 | 3.0 | 0.01 | 1,024 | 34,753 | 0.0295 | 2,006 | $1.2\cdot10^8$ | 1.1 | 18.1 | 21.6 | 14.9 | 17.8 |
| 0.3072 | 3.0 | 0.01 | 102,400 | 3,475,300 | 0.0295 | 2,006 | $3.7\cdot10^2$ | 1.1 | - | - | - | - |
| 0.6144 | 3.0 | 0.01 | 204,800 | 6,950,600 | 0.0295 | 2,006 | $4.2\cdot10^1$ | 2.0 | - | - | - | - |
| 0.6144 | 3.0 | 0.01 | 204,800 | 69,506,000 | 0.00295 | 2,034 | $7.2\cdot10^0$ | 3.0 | - | - | - | - |
| 0.6144 | 3.0 | 0.01 | 204,800 | 695,060,000 | 0.000295 | 3,390 | $3.7\cdot10^0$ | 6.0 | - | - | - | - |
| - | 0 | 0.01 | 1,024 | 34,753 | 0.0295 | 2,000 | inf | - | 13.9 | 16.7 | 14.0 | 16.8 |

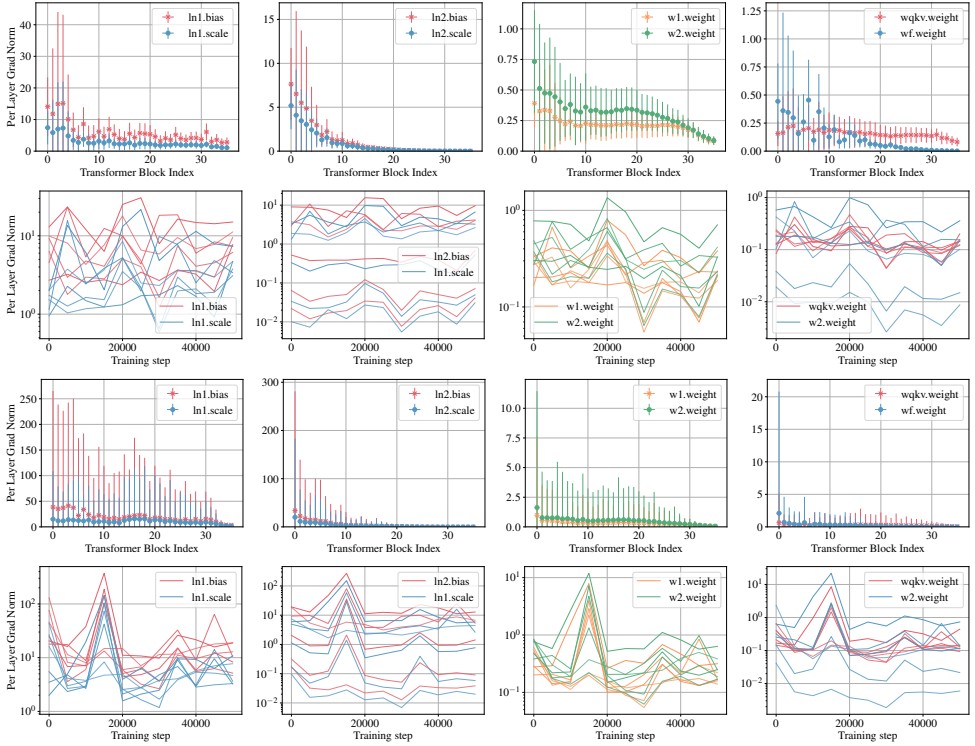

Figure 17: (first and second rows) Central training on *CV-en-train* from the *LS-960* seed model and (third and fourth rows) Central training on *CV-en-train-90* from the *CV-en-train-10* seed model and their per layer gradients norm: (first, third rows) averaged across training steps and (second, fourth) showed per layer along the training. The model is trained with LARS optimizer and the learning rate of 0.5/0.2. LayerNorm gradients **do** dominate over MLP and attention gradients.

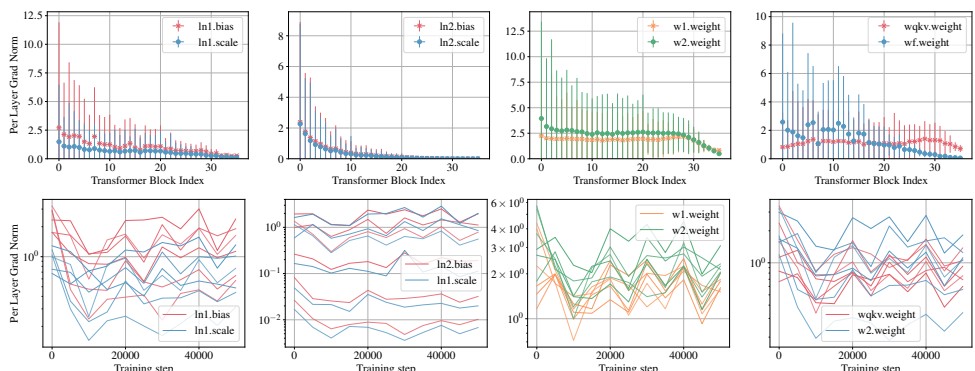

Figure 18: Central training on *CV-fr-train-90* from the *CV-fr-train-10* seed model and its per layer gradients norm: (top) averaged across training steps and (bottom) showed per layer along the training. The model is trained with LARS optimizer and the learning rate of 0.2. The norms of the per-layer gradients are balanced similarly to models trained with FL or with FL and DP in Figure 19: LayerNorm gradients **do not** dominate over MLP and attention gradients.

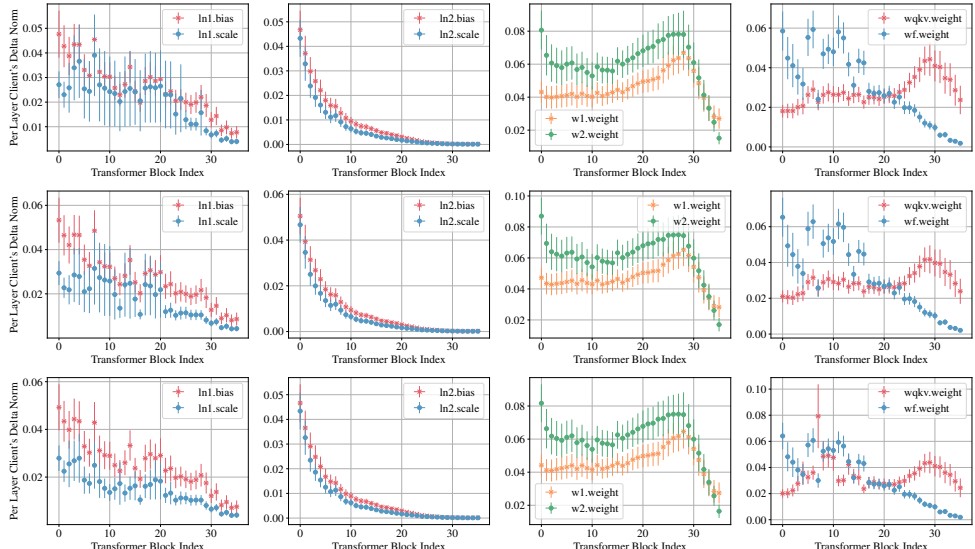

Figure 19: Client delta norms computed per layer in the French model trained on *CV-fr-train-90* from a seed *CV-fr-train-10* model. We average the statistics across all clients and central steps, and plot the mean and standard deviation. The model is trained with (first row) global clients' deltas clipping $C = 10^{-2}$ and $\sigma_{\mathrm{DP}} = 0$, (second row) global clients' deltas clipping $C = 10^{-2}$ and $\sigma_{\mathrm{DP}} = 3 \cdot 10^{-6}$, (third row) per-layer clients' deltas clipping (Definition 3, "dim") $C = 10^{-2}$ and $\sigma_{\mathrm{DP}} = 3 \cdot 10^{-6}$. The rest of the training configuration is the same as in Figure 5. A transformer block consists of attention parameters (wqkv and wf), MLP (w1 and w2), LayerNorm applied to input of attention (ln1) or MLP (ln2).

datasets with a larger number of speakers for French and German (as we need a large cohort size to alleviate the impact of DP noise), and to probe other languages.

[101] also used per-layer clipping but for NLP domain and observed the difference in the gradient norms of different transformer layers. However, per-layer clipping did not outperform the global clipping for training with DP (there was no FL component) in many settings. We would like to highlight the main differences with our study for ASR domain: i) our architecture is encoder-based model trained with a sequence loss (CTC), while [101] use decoder-based (causal) model trained with cross-entropy loss; ii) Tables 3 and 4 of [101] show that per-layer clipping significantly improves results for GLUE tasks, thus it is task dependent; iii) [101] fine-tune pre-trained model for

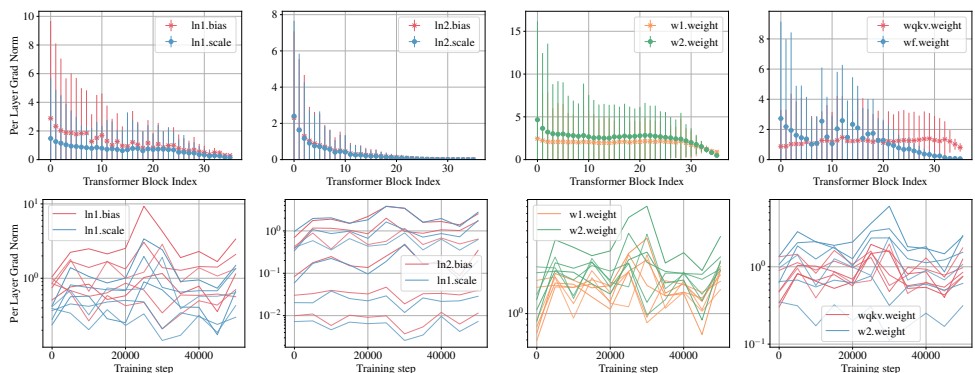

Figure 20: Central training on *CV-de-train-90* from the *CV-de-train-10* seed model and its per layer gradients norm: (top) averaged across training steps and (bottom) showed per layer along the training. The model is trained with LARS optimizer and the learning rate of 0.2. The norms of the per-layer gradients are balanced similarly to models trained with FL or with FL and DP in Figure 21: LayerNorm gradients **do not** dominate over MLP and attention gradients.

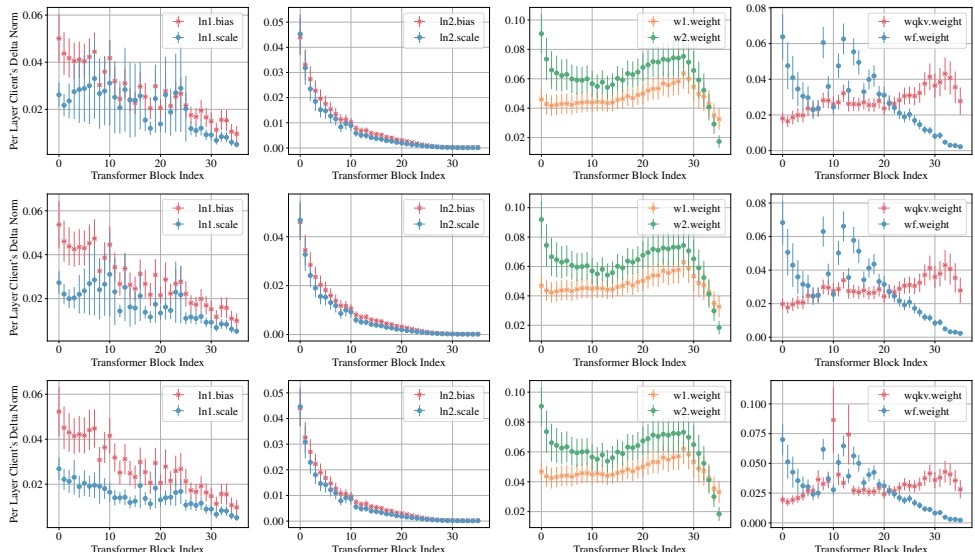

Figure 21: Client delta norms computed per layer in the German model trained on *CV-de-train-90* from a seed *CV-de-train-10* model. We average the statistics across all clients and central steps, and plot the mean and standard deviation. The model is trained with (first row) global clients' deltas clipping $C = 10^{-2}$ and $\sigma_{\mathrm{DP}} = 0$, (second row) global clients' deltas clipping $C = 10^{-2}$ and $\sigma_{\mathrm{DP}} = 3 \cdot 10^{-6}$, (third row) per-layer clients' deltas clipping (Definition 3, "dim") $C = 10^{-2}$ and $\sigma_{\mathrm{DP}} = 3 \cdot 10^{-6}$. The rest of the training configuration is the same as in Figure 5. A transformer block consists of attention parameters (wqkv and wf), MLP (w1 and w2), LayerNorm applied to input of attention (ln1) or MLP (ln2).

a downstream task with another objective (this can affect the contribution of different parts of the model) while in ASR we keep it the same. Moreover, our theoretical results (Theorem 2) show that per-layer clipping can help to improve convergence in case of higher level of heterogeneity.

### H.7 Per-Layer Clipping for Different Model Sizes

We further evaluate effectiveness of the per-layer clipping for different model sizes. We take the baseline model we used before with 36 layers, 768 embedding and 3072 MLP dimension (244M parameters), set its layer drop to 0.1 and consider the following models: *narrow* with 114M parameters (reduce embedding to 512 and MLP dimension to 2048), *wide* with 450M parameters (increase

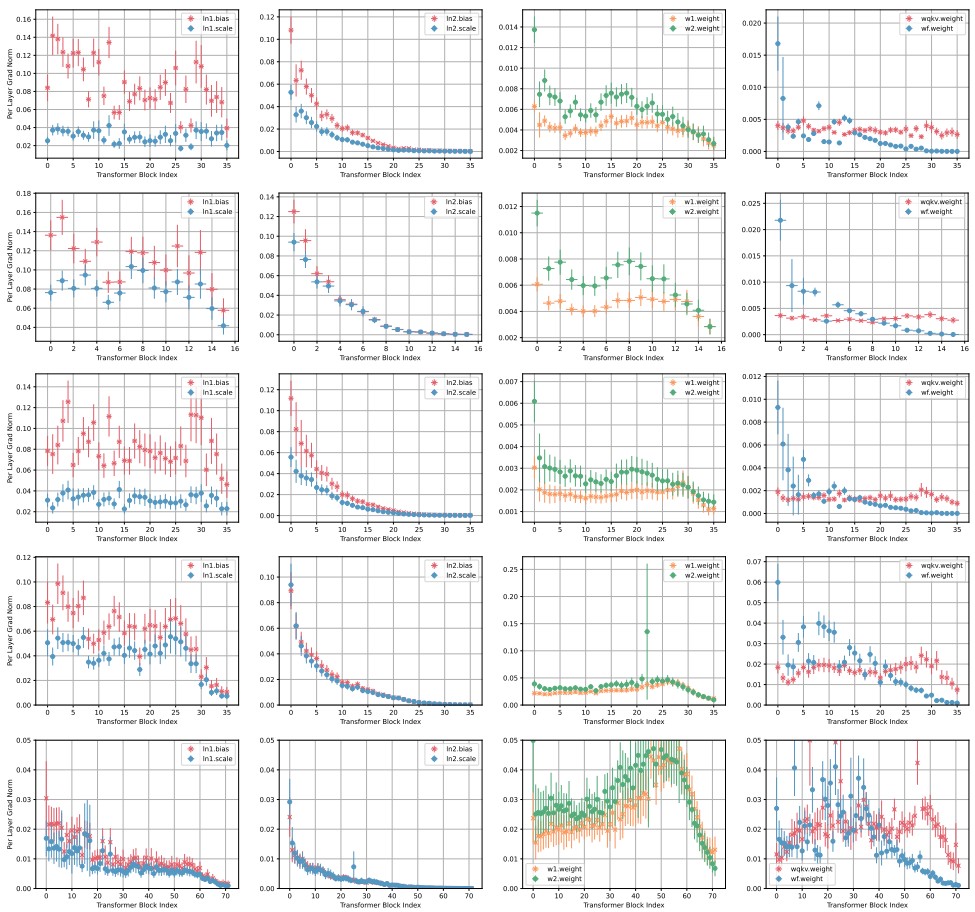

Figure 22: Client delta norms computed per layer in the narrow (row 1), shallow (row 2), baseline (row 3), wide (row 4) and deep (row 5) models trained on *CV* from a seed *LS-100* model. We average the statistics across all clients and central steps, and plot the mean and standard deviation. All models are trained with global clients' deltas clipping $C = 10^{-2}$ and $\sigma_{\mathrm{DP}} = 10 \cdot 10^{-6}$. A transformer block consists of attention parameters (wqkv and wf), MLP (w1 and w2), LayerNorm applied to input of attention (ln1) or MLP (ln2).

embedding to 1024 and MLP dimension to 4096), *shallow* with 114M parameters (reduce only number of layers to 16) and *deep* with 510M parameters (increase depth to 72 layers). All models are trained with the same hyperparameters as the baseline model – we only change the model architecture as discussed (with layer drop set 0.1 for all models including the baseline). There are few takeaways and observations from the results (all comparisons are provided on test set), shown in Table 21:

1. Per-layer clipping consistently outperforms global clipping for different model sizes.

2. For per-layer clipping, as model size increases, the model performance in FL with DP degrades more compared to FL. This holds for both increasing model size via width and depth. Degradation for increasing model width is smaller compared to model depth. These results are in line with our theoretical results.

3. For global clipping, as model size increases, the model performance in FL with DP degrades more compared to FL. This holds for both increasing model size via width and depth. However, for larger model size (wide and deep) we see significant performance improvement – we hypothesize that it is due to the lower gradient imbalance between layer normalization and FC layers, see Figures 22 and 23 for global and per-layer clipping. Model sizes $> 500$M we leave for the future exploration and highlight the need to study larger models considering model size limitations and aforementioned results in the current work.

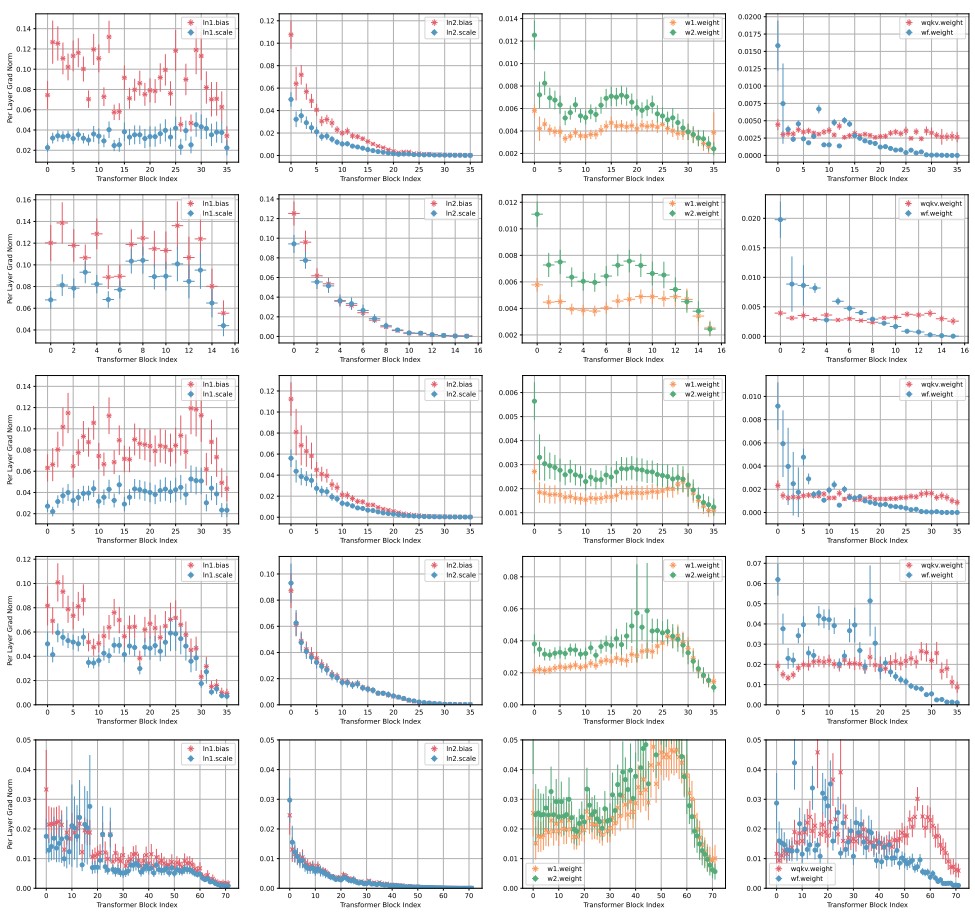

Figure 23: Client delta norms computed per layer in the narrow (row 1), shallow (row 2), baseline (row 3), wide (row 4) and deep (row 5) models trained on *CV* from a seed *LS-100* model. We average the statistics across all clients and central steps, and plot the mean and standard deviation. All models are trained with per-layer (Definition 3, "uniform") clients' deltas clipping $C = 10^{-2}$ and $\sigma_{\mathrm{DP}} = 10 \cdot 10^{-6}$. A transformer block consists of attention parameters (wqkv and wf), MLP (w1 and w2), LayerNorm applied to input of attention (ln1) or MLP (ln2).

Table 21: Ablation for FL and FL with DP with a model pre-trained on *LS-100* used as central data and afterwards fine-tuned *CV-en-train*. We report added noise $\mathcal{N}(0, IC^2\sigma_{\mathrm{DP}}^2 qK)$ per client and CV dev and test WERs (%) for two clipping variants with clipping bound $C = 0.01$: global and per layer "uniform". Total number of users $K = 34,753$, expected number of users sampled per central step $S = qK = 1024$, and the number of central steps $T = 2000$ are given. We also show relative degradation in performance for test set if we switch from FL to FL+DP for a specific configuration.

| Model | $\sigma_{\mathrm{DP}}(\cdot 10^{-6})$ | global clipping | | | per-layer clipping "uniform" | | |
|---|---|---|---|---|---|---|---|
| | | dev WER (%) | test WER (%) | rel. % ↓ | dev WER (%) | test WER (%) | rel. % |
| narrow | 0 | 15.2 | 18.2 | - | - | - | - |
| | 10 | 27.5 | 31.7 | 74.2 | 19.5 | 23.2 | 27.5 |
| baseline | 0 | 14.7 | 17.6 | - | - | - | - |
| | 10 | 29.9 | 34.6 | 96.6 | 19.7 | 23.3 | 32.4 |
| wide | 0 | 13.7 | 16.6 | - | - | - | - |
| | 10 | 20.8 | 24.7 | 48.8 | 20.0 | 23.7 | 42.8 |
| shallow | 0 | 16.3 | 19.8 | - | - | - | - |
| | 10 | 30.6 | 35.1 | 77.3 | 20.9 | 24.8 | 25.3 |
| baseline | 0 | 14.7 | 17.6 | - | - | - | - |
| | 10 | 29.9 | 34.6 | 96.6 | 19.7 | 23.3 | 32.4 |
| deep | 0 | 14.2 | 17.2 | - | - | - | - |
| | 10 | 21.7 | 25.7 | 49.4 | 22.4 | 26.4 | 53.5 |

# I Compute Resources

In Table 22 we show the summary of used compute of the main training configurations for benchmarks of FL and FL with DP for transparency and setting proper expectations for the community.

Table 22: Compute for the main expeirments we run for FL and FL with DP. For all experiments we use LAMB as the central optimizer and SGD as the local optimizer.

| Seed | Data | Model | Client Total Batch Size | Cohort Size $S$ | Local | Central Steps $T$ | # GPUs A100 80GB | Runtime (h) | Total GPU (h) |
|------|------|-------|-------------------------|------------------|-------|-------------------|-------------------|-------------|---------------|
| CV-en-train | LS-960 | FL | 6min | 8 | 10 epochs | 2000 | 2 | 53 | 106 |
| CV-en-train | LS-960 | FL | 6min | 16 | 10 epochs | 2000 | 2 | 103 | 206 |
| CV-en-train | LS-960 | FL | 6min | 32 | 10 epochs | 2000 | 2 | 191 | 382 |
| CV-en-train | LS-960 | FL | 6min | 64 | 10 epochs | 2000 | 4 | 278 | 1,112 |
| LS-960 | CV-en-train | FL | 2min | 16 | 10 epochs | 2000 | 2 | 42 | 84 |
| LS-960 | CV-en-train | FL | 2min | 32 | 10 epochs | 2000 | 2 | 62 | 124 |
| LS-960 | CV-en-train | FL | 2min | 64 | 10 epochs | 2000 | 2 | 98 | 196 |
| LS-960 | CV-en-train | FL | 2min | 128 | 10 epochs | 2000 | 2 | 169 | 338 |
| LS-960 | CV-en-train | FL | 2min | 256 | 10 epochs | 2000 | 4 | 304 | 1,216 |
| LS-100 | CV-en-train | FL | 2min | 1,024 | 10 steps | 2000 | 32 | 34 | 1,088 |
| LS-100 | CV-en-train | FL + DP | 2min | 1,024 | 10 steps | 2000 | 32 | 35 | 1,120 |
| LS-100 | CV-en-train | FL + DP | 2min | 256 | 10 steps | 2000 | 16 | 18 | 288 |
| CV-de-train-10 | CV-de-train-90 | FL | 2min | 1,024 | 10 steps | 2000 | 16 | 66 | 1,056 |
| CV-de-train-10 | CV-de-train-90 | FL + DP | 2min | 1,024 | 10 steps | 2000 | 16 | 67 | 1,072 |
| CV-fr-train-10 | CV-fr-train-90 | FL | 2min | 1,024 | 10 steps | 2000 | 16 | 60 | 960 |
| CV-fr-train-10 | CV-fr-train-90 | FL + DP | 2min | 1,024 | 10 steps | 2000 | 16 | 61 | 976 |
| CV-fr-train-10 | CV-fr-train-90 | FL + DP | 2min | 1,024 | 10 steps | 2000 | 64 | 18 | 1,152 |

# J Contributions

The overall vision for enabling differentially private federated learning in ASR was conceived by Martin Pelikan, Sheikh Shams Azam, Jan "Honza" Silovsky, and Tatiana Likhomanenko, who identified the gap in current research and defined the problem scope. The work on Differential Privacy was done in consultation with Vitaly Feldman and Kunal Talwar and the theoretical work was done in consultation with Christopher G Brinton and Kunal Talwar. Specific contributions of the authors can be attributed as:

- **Algorithm Design.** The design of algorithm including per-layer clipping and layer-wise gradient normalization was led by Martin Pelikan, Sheikh Shams Azam, Jan "Honza" Silovsky, and Tatiana Tatiana Likhomanenko in consultation with Vitaly Feldman and Kunal Talwar.

- **Implementation and Experimental Results.** The FL with DP training pipeline was developed by Martin Pelikan and Tatiana Likhomanenko. Martin Pelikan carried out the extensive experiments for FL evaluating the effects of data heterogeneity, optimizer settings, and initialization strategies, while Tatiana Likhomanenko carried out the extensive experiments for FL with DP evaluating DP and different clipping strategies. All was done in consultation with Sheikh Shams Azam, Jan "Honza" Silovsky, Vitaly Feldman and Kunal Talwar.

- **Theoretical Convergence Analysis.** The theoretical analysis of per-layer clipping and layer-wise adaptive optimizer was led by Sheikh Shams Azam and Tatiana Likhomanenko. The FL convergence analysis was done in consultation with Christopher G. Brinton and the analysis of DP in the bound was done in consultation with Kunal Talwar. Kunal Talwar double checked all derivations in the final proof.

- **Writing and Paper Preparation** The manuscript was jointly written by Martin Pelikan, Sheikh Shams Azam, and Tatiana Likhonamanenko. It was edited and reviewed by all other authors.

