# OpenReview forum: "Enabling Differentially Private Federated Learning for Speech Recognition: Benchmarks, Adaptive Optimizers, and Gradient Clipping"
_NeurIPS.cc/2025/Conference — NeurIPS 2025 poster_

### Official Review · Reviewer_R1Qm · 2025-06-06

**Clarity:** 3
**Significance:** 3
**Originality:** 4
**Rating:** 5
**Confidence:** 4

**Summary:**

This paper addresses the challenge of applying Federated Learning with Differential Privacy to Automatic Speech Recognition, a task complicated by the heterogeneity of large transformer models. The authors establish the first practical benchmark for FL with DP in end-to-end ASR, demonstrating that per-layer clipping and layer-wise gradient normalization can mitigate gradient heterogeneity and clipping bias. Their empirical results show that FL with DP is viable under strong privacy guarantees, achieving competitive performance with only a small degradation in WER.

**Questions:**

1. How does the proposed method scale to even larger models?
2. Could you provide more insights into how different DP budgets affect WER in practical scenarios?
3. Does the FL+DP approach ever match centralized training performance, or is there always a gap?

**Ethical Concerns:**

["NO or VERY MINOR ethics concerns only"]

**Final Justification:**

The rebuttal provided by the authors has addressed my concerns well, and I have decided to maintain my positive evaluation.

**Limitations:**

Yes.

**Paper Formatting Concerns:**

I didn't find any formatting issues.

**Quality:**

4

**Strengths And Weaknesses:**

Strengths:
This paper establishes the first benchmark of FL+DP in large-scale ASR and makes an important contribution. The proposed layer-by-layer pruning and gradient normalization techniques effectively balance the model performance and privacy protection, and the demonstration process is relatively detailed. This work is highly original and demonstrates that FL and DP perform differently on large models compared to small models.

Weaknesses:
The computational overhead of the proposed methods may limit scalability to larger models or real-time applications.
Additionally, the relatively high error rates of baseline models on open-source datasets like Librispeech suggest that more recent ASR models should be used as baselines to better validate the effectiveness of FL+DP on larger ASR systems.

---

> ### Author Rebuttal · Authors · 2025-07-30
>
> Dear Reviewer R1Qm,
>
> We deeply appreciate your positive assessment of our work and your recognition of its importance as "the first benchmark of FL+DP in large-scale ASR." We address each of your points below.
>
>
> > "The computational overhead of the proposed methods may limit scalability to larger models or real-time applications"
>
>
> You raise an important practical concern about computational overhead. We acknowledge this limitation while emphasizing that our work addresses a more fundamental challenge:
>
> - Prior works such as Gao et. al 2022 [20] pointed out that training FL-ASR was infeasible. Some improvements were made by Azam et al. 2023 [56] to train a 120M Conformer model, but the training of FL with DP for ASR was still infeasible. Our primary contribution is establishing that such training is possible and studying the underlying reasons why this infeasibility existed.
> - We deliberately focused on the feasibility first, with efficiency optimization being a natural next step that merits dedicated research. We have already seen important insights from contemporary research in the domains of (i) parameter-efficient methods, (ii) gradient compression and quantization (iii) selective layer updates that might be helpful in this regard. However, each of these directions deserves substantial research, which we defer to future improvements that can be studied on top of our findings.
>
> ---
>
> > "the relatively high error rates of baseline models on open-source datasets like Librispeech suggest that more recent ASR models should be used as baselines"
>
> We believe that this assessment might be originating from the confusion between the model performance in the centralized and federated learning cases. For context, we compare against Whisper small (244M parameters, see Radford et al. 2022 D.2.2 - similar size to our model):
>
> - For LibriSpeech test-other, Whisper small achieves 7.6% WER. In comparison, our centralized baseline achieves 6.8% WER on LibriSpeech test-other (Table 3, Appendix). SOTA models trained only on LibriSpeech achieve 5-6% WER for CTC models w/o a beam search and LM and are close to 3% with a beam search and LM - our model is ~3.5% when the beam search and LM are integrated. Our FL baseline on LibriSpeech test-other achieves 7.1% WER (only 0.3% degradation from centralized baseline). For context, prior work, Azam et al. 2023 [56], achieves 10.29% WER for FL setting with CTC-AED model.
> - For Common-Voice English, Whisper small achieves 14.5% WER. In comparison, our centralized baseline achieves 18.2% WER (Table 4, Appendix) without a beam search and LM. Our FL baseline on Common Voice achieves 25.7% WER. For context, no prior work in FL-ASR, including Gao et al. 2022 [20], uses Common Voice English.
> - Additionally, Gao et al. 2022 [20] are able to train a model using FL on Common-Voice French and achieve 20.99% WER. In comparison, we achieve 13.1% WER (Table 10, Appendix) on Common-Voice French, while Whisper small is reported as 22.7%.
>
> ---
>
> > "How does the proposed method scale to even larger models?"
>
> While we were not able to train a larger model within the rebuttal timeframe, we hope that the following table comparing against a model with 114M parameters can shed some light on the phenomenon. We can observe that (i) the larger 250M parameter model is more affected by uniform clipping, and (ii) it also benefits more from the per-layer clipping.
>
> |Model Size|T|Uniform Clipping|Per-Layer Clipping|Rel. Improv.|
> |---|---|---:|---:|---:|
> |114M|2k|29.5|21.3|27.8\%|
> |250M|2k|31.1|20.4|34.4\%|
>
> This is in line with our theoretical analysis (Theorem 2) which emphasizes that per-layer clipping benefits model with greater depth H and layer-wise gradient heterogeneity (Ψ$_h$ terms) which is more prominent in deeper models (Chan et al. [23], Wang et al. [24]; discussed in Sec. 3). This confirms that **larger models benefit more** from our approach. This is intuitive since more gradient heterogeneity across layers can be expected in deeper models, thus benefiting from per-layer intervention, i.e., per-layer clipping and layer-wise adaptive optimization.
>
> ---
>
> > "Could you provide more insights into how different DP budgets affect WER in practical scenarios?"
>
>
> We provide a detailed analysis in Table 17 (Appendix). There exists a **privacy-utility trade-off** between DP noise and WER, wherein larger DP noise leads to worse WER. However, we point to the trend that better privacy-utility trade-offs are realized by per-layer clipping when compared to constant clipping
>
> Practically, the target epsilon value often depends on the application, wherein a smaller epsilon ε ≤ 3 (i.e., more privacy) is preferred for training with sensitive data such as healthcare, and a slightly larger epsilon ε = 3-10 might be acceptable in other domains such as object detection, etc.
>
> Furthermore, population size dramatically impacts achievable privacy (Table 1). This is in line with prior works such as McMahan et al. 2018 [46].
>
> ---
>
> > "Does the FL+DP approach ever match centralized training performance, or is there always a gap?"
>
>
> This touches on fundamental aspects of privacy-preserving ML. In general, if the amount of data is similar between the centralized and FL training, then centralized training acts as an upper bound on FL training performance. This is due to the following:
>
> - FL training observes more heterogeneous training as data cannot be shuffled, similar to centralized training.
> - DP adds random noise to the training, which creates an irreducible gap in performance.
>
> ---
>
> **Summary**
>
> We thank you for recognizing this work as making "an important contribution" with "high originality." Your questions about practical deployment are exactly the right ones to ask as we move FL+DP from research to production. While computational challenges exist, this work addresses the fundamental challenge of the feasibility of training.

---

> > ### Comment · Reviewer_R1Qm · 2025-08-04
> >
> > Thank you for the clarification. I will keep my rating.

---

### Official Review · Reviewer_TNaa · 2025-06-12

**Clarity:** 3
**Significance:** 2
**Originality:** 2
**Rating:** 2
**Confidence:** 3

**Summary:**

This paper proposes to use federated learning with differential privacy for ASR. The authors investigate a Transformer-based ASR model trained with CTC loss under various federated learning settings and evaluate its performance on two datasets.

**Questions:**

It is unclear why the proposed model is characterized as a 'large model' in the context of ASR. Many recent ASR studies employ models with over 250 M parameters. A large amount of papers on ASR utilize models larger than 250M parameters. Clarifying this with respect to existing literature would strengthen the positioning of your work.

**Ethical Concerns:**

["NO or VERY MINOR ethics concerns only"]

**Final Justification:**

My original concerns remain largely unresolved and they are very important for this paper to be accepted. The authors did not revise or meaningfully address core concerns raised in the initial review. Their response did not demonstrate sufficient willingness to improve the paper based on my feedback for model architecture, dataset, novelty and model scale. Due to unresolved major concerns regarding methodology, dataset evaluation, and novelty, I recommend rejecting this paper.

**Limitations:**

Given that the paper is positioned as a benchmark study, a more comprehensive experimental evaluation is expected, including a wider range of backbone models and datasets. Relying solely on LibriSpeech and CommonVoice is insufficient to robustly support the claims made since more ASR works report their results on more than two datasets. It is recommended that the authors incorporate additional ASR datasets to enhance the generalizability and credibility of the findings.


In Table 2, the reported WERs for both datasets are relatively high, especially for LibriSpeech, where much lower WERs are typically achieved in recent literature. This suboptimal performance appears to stem from the choice of model architecture and the limited training data scale. While the proposed method shows improvements, the overall weak ASR performance may hinder its impact in the field. I recommend the authors include more recent and competitive ASR models in their experiments to better demonstrate the effectiveness of their approach.

**Quality:**

2

**Strengths And Weaknesses:**

Quality: The general quality is good but can be improved further with better experiment design. Regarding ASR model selection, the use of a transformer with CTC—while effective in earlier work—is relatively outdated given recent advances in end-to-end ASR, including models utilizing both CTC and sequence-to-sequence (S2S) losses, as well as the emergence of speech foundation models and speech language models (LLMs).

Clarity: The paper is well-written with a clear structure. The integration of federated learning with ASR presents a promising research direction and offers valuable insights for future work. The in-depth analysis of data heterogeneity and the application of differential privacy in federated learning is particularly insightful and inspiring.

Significance: In the context of ASR, it is unclear what specific contributions have been made to address speech recognition challenges beyond the integration of existing federated learning and differential privacy techniques. The current approach appears more aligned with an engineering implementation rather than a novel methodological advancement, which may limit its suitability for a NeurIPS publication.

Originality: The novelty of the work appears limited, as the use of federated learning with differential privacy has been explored in prior research. It remains unclear what specific technical contributions are made from both the federated learning and differential privacy perspectives. Clarifying these aspects would strengthen the paper’s contribution.

---

> ### Author Rebuttal · Authors · 2025-07-30
>
> Dear Reviewer TNaa,
>
> We thank you for the thoughtful and detailed feedback. We greatly appreciate the points raised and would like to clarify a few aspects of our work, as some key elements of our contributions and the FL+DP setting may have been misunderstood. We address each point below with the hope of resolving these potential concerns.
>
>
> > "the use of a transformer with CTC—while effective in earlier work—is relatively outdated given recent advances in end-to-end ASR, including models utilizing both CTC and sequence-to-sequence (S2S) losses"
>
> We offer a different perspective that CTC is still a highly relevant and widely adopted approach in the literature:
>
> - NVIDIA's FastConformer-CTC achieves competitive results on the OpenASR leaderboard (2024) (see [HuggingFace Open ASR Leaderboard](https://huggingface.co/spaces/hf-audio/open_asr_leaderboard))
> - Google's USM employs CTC for their 2B parameter universal speech model [Zhang et al., 2023; Wang et al., 2024]
> - Meta's MMS uses CTC for their massively multilingual model covering 1,100+ languages [Pratap et al., 2023]
>
> Additionally, our choice of ASR model is deliberate since we aimed to solve the ASR problem with on-device capability. Speech LLMs may still be infeasible due to device constraints and are still behind cascaded models (see “Spirit-LM: Interleaved spoken and written language model,” Transactions of the Association for Computational Linguistics, 2025; “Qwen2. 5-omni technical report,” arXiv:2503.20215, 2025; Voicebench: Benchmarking llm-based voice assistants; arXiv:2410.17196, 2024; MMAU: A massive multitask audio understanding and reasoning benchmark, arXiv:2410.19168, 2024). Foundation models of similar size are not comparable on ASR tasks, thus making the current ASR models more suitable for on-device applications.
>
> ---
>
> > "it is unclear what specific contributions have been made to address speech recognition challenges beyond the integration of existing federated learning and differential privacy techniques"
>
> Below, we clarify the aspects of our work that are beyond just the integration of existing methods:
>
> 1. We observe that a naive integration of FL with DP does not converge in ASR. We arrive at a practical recipe to circumvent the convergence problems by using additional interventions: (i) per-layer clipping and (ii) layer-wise normalization.
> 2. We demonstrate a clear departure from existing knowledge about per-layer clipping — previous work [McMahan et al., 2018] found no benefit for LSTMs — by providing empirical evidence supporting its effectiveness.
> 3. We also theoretically analyze the effect of per-layer clipping together with layer-wise adaptive optimization on the convergence dynamics. Our theoretical analysis goes beyond existing FL literature to understand the impact of heterogeneity, depth, and optimizer dynamics on convergence.
> 4. We “establish the first benchmark of FL+DP in large-scale ASR” (also echoed by Reviewer R1Qm) and further study the framework to explore the impact of: (i) different initialization strategies and (ii) in-domain and out-of-domain pre-training initialization.
>
> Thus, in addition to the engineering implementation, our work also provides fundamental insights into optimization dynamics, theoretical guarantees, and practical techniques that advance the field. Furthermore, as echoed by Reviewer UQPF, the "proposed approach can potentially be applied to other domains apart from ASR and can have a significant impact."
>
> ---
>
> > "the reported WERs for both datasets are relatively high, especially for LibriSpeech, where much lower WERs are typically achieved"
>
>
> We believe that this assessment might be originating from the confusion between the model performance in the centralized and federated learning cases. For context, we compare against Whisper small (244M parameters, see Radford et al. 2022 D.2.2 - similar size to our model):
>
> - For LibriSpeech test-other, Whisper small achieves 7.6% WER. In comparison, our centralized baseline achieves 6.8% WER on LibriSpeech test-other (Table 3, Appendix). SOTA models trained only on LibriSpeech achieve 5-6% WER for CTC models w/o a beam search and LM and are close to 3% with a beam search and LM - our model is ~3.5% when the beam search and LM are integrated. Our FL baseline on LibriSpeech test-other achieves 7.1% WER (only 0.3% degradation from centralized baseline). For context, prior work, Azam et al. 2023 [56], achieves 10.29% WER for FL setting with CTC-AED model.
> - For Common-Voice English, Whisper small achieves 14.5% WER. In comparison, our centralized baseline achieves 18.2% WER (Table 4, Appendix) without a beam search and LM. Our FL baseline on Common Voice achieves 25.7% WER. For context, no prior work in FL-ASR, including Gao et al. 2022 [20], uses Common Voice English.
> - Additionally, Gao et al. 2022 [20] are able to train a model using FL on Common-Voice French and achieve 20.99% WER. In comparison, we achieve 13.1% WER (Table 10, Appendix) on Common-Voice French, while Whisper small is reported as 22.7%.
>
> ---
>
> > "Relying solely on LibriSpeech and CommonVoice is insufficient to robustly support the claims"
>
> We understand the concern and wanted to clarify that our choice for FL simulations is constrained by the requirements of speaker metadata:
>
> - Datasets WITH speaker IDs (suitable for FL): LibriSpeech (\~2k speakers), CommonVoice (\~35k speakers), VCTK (\~100 speakers), TED-LIUM (\~1.2k speakers), etc.
> - Datasets WITHOUT speaker IDs (unsuitable for FL): People's Speech, GigaSpeech, SPGISpeech, etc.
>
> The requirement for speaker ID to simulate realistic user partitions severely limits options. Among suitable datasets, LibriSpeech and CommonVoice provide the largest speaker diversity (see Table 2, Appendix).
>
> |Dataset|# speakers|minutes/speaker|domain|
> |---|---|---|---|
> |LibriSpeech|~2.3k|min: 2, max: 30|read-speech|
> |CV english|~35k|min:0.02, max: ~5k|conversational|
> |CV french|~6.9k|min:0.04, max: ~3k|conversational|
> |CV german|~7.1k|min:0.03, max: ~6k|conversational|
>
> This diversity ensures our methods are tested across the wide spectrum of FL challenges. Our evaluation spans 4 datasets (LS + 3 CV languages), whereas most prior FL-ASR works typically evaluate on 1-2 datasets [19, 20, 56].
>
> ---
>
> > "It is unclear why the proposed model is characterized as a 'large model' in the context of ASR"
>
>
> The characterization of "large" is **relative to FL literature**, not centralized ASR. Typical FL models in literature are <30M parameters (e.g., FedAvg [1], FedProx [11]). Prior work in FL-ASR has scaled up to ~120M parameters [19] without DP. Thus, by comparison, our model with 250M parameters is more than 2x larger than the prior works in the domain and ~10x larger than conventional FL works. This scaling is particularly significant in the domain of DP because DP noise scales with model dimension. Additionally, successfully training such models in FL+DP was previously considered infeasible (Gao et. al [20]).
>
> ---
>
> **Summary**
>
> We thank the reviewer for their thoughtful feedback and hope the responses above address their concerns. We will incorporate these clarifications in the final draft of the paper. We request the reviewer to reconsider their score in light of these clarifications and would be happy to provide further details if helpful.

---

> > ### Comment · Reviewer_TNaa · 2025-08-04
> >
> > Thank you for your reply. I will keep my rating and wait for your updated version.

---

> > > ### Author Response · Authors · 2025-08-04
> > >
> > > Dear Reviewer TNaa,
> > >
> > > We would like to mention that unfortunately, this NeurIPS, the rules have changed and we cannot upload a revision during the rebuttal. We also wanted to emphasize that **nearly all the concerns raised in your review are addressed in the paper/Appendix.** We will highlight these more/bring it forward to the main paper.
> > > - our contributions are listed in detail in the main text (lines 46-65)
> > > - usage of CTC loss and encoder models is discussed in Appendix G (lines 1188-1190) and Appendix F4, F5 + speech community is well aware that CTC is still a strong model and speech LLM is behind for many speech tasks
> > > - characterization of "large" being relative to FL literature is discussed in Appendix F2, lines 1085-1103
> > > - data and their characteristics are discussed in the main text (lines 206-223), Appendix F6 (lines 1133-1158), Appendix G (lines 1160-1174), and Table 2 in Appendix. We will add reference to other datasets with smaller number of speakers or w/o any speaker information in the camera ready paper. However, it is a well known in speech community.
> > > - WER of baseline models: we have already discussed and compared with prior works including [20, 56] in the main text (lines 227-228, 319-334) and Appendix H5 (1273-1274).
> > >
> > > We hope this gives you a reasonably good idea of what our revised paper will look like.  Please let us know if you have any additional concerns.

---

> > ### Comment · Reviewer_TNaa · 2025-08-06
> >
> > Thank you for your response. However, after careful consideration of your clarifications, my original concerns remain largely unresolved:
> >
> > 1. Model Architecture Choice: While you have provided examples where CTC-based architectures remain relevant, my primary concern was specifically about employing CTC without exploring integration with S2S losses, which has been widely acknowledged in recent ASR advances. Your explanation highlights industry adoption but does not justify why your approach specifically avoids leveraging recent successful S2S+CTC hybrid methodologies, which could further strengthen performance and generalizability.
> >
> > 2. Dataset: Your justification regarding dataset constraints is understood. However, as emphasized in my initial review, relying solely on LibriSpeech and CommonVoice remains insufficient for establishing robust claims required at top-tier conferences. Although these datasets offer some speaker diversity, they are limited in linguistic and acoustic diversity compared to the broader range of available ASR benchmarks. Expanding evaluations, even within available FL-compatible datasets, would significantly enhance the robustness and impact of your findings.
> >
> > 3. Novelty: Your additional clarifications on contributions, particularly regarding per-layer clipping, offer valuable insights. Nonetheless, these appear primarily as incremental methodological adjustments rather than substantial breakthroughs. Your theoretical contributions and practical insights, while beneficial, need stronger evidence or benchmarks clearly demonstrating significant performance improvements relative to simpler integrations of FL and DP.
> >
> > 4. Large models: Your definition of 'large' as relative within FL contexts is noted. However, a parameter count alone is insufficient to justify labeling a model as "large," especially when evaluating ASR performance.

---

> > > ### Author Response · Authors · 2025-08-07
> > >
> > > Dear Reviewer TNaa,
> > >
> > > We appreciate your continued engagement. We'd like to address your specific concerns with additional clarifications:
> > >
> > > 1. Regarding Novelty: We believe that our claim to novelty is in line with the NeurIPS conference guidelines (see https://neurips.cc/Conferences/2025/CallForPapers) that _"encourages in-depth analysis of existing methods that provide new insights in terms of their limitations or behaviour"_. Our work directly fulfills this criteria given no prior work has successfully trained competitive ASR models with FL+DP -- not because researchers haven't tried to "integrate" these techniques, but since standard integration fundamentally fails to converge for large models. Our contribution is thus not incremental as it **enables a previously impossible capability** and provides **fundamental theoretical understanding** of why it works. This is a **qualitative breakthrough** compared to prior works going from (i) non-converging thus unusable models towards deployable models and (ii) achieving stronger WERs under the same privacy budgets.
> > >
> > > 2. Regarding Model Architecture Choice: Our prior response was clarifying the concern regarding "CTC is relatively outdated" which we believe is addressed. We understand your remaining concern is regarding "exploring CTC without integration with S2S". Competitive ASR models can be trained with multiple candidate losses including CTC-AED, Transducer Loss, etc. apart from CTC only. The jury is still out on whether one of the losses is fundamentally better than the other. Given that the main goal of our work was to enable FL with DP of ASR models **we chose one of the widely-used losses** and spent the majority of our training and analysis resources on analyzing the **convergence obstacles arising out of FL+DP** training (which was previously not possible) which is the main contribution.
> > >
> > > 3. Regarding Dataset: We believe that the reviewers main critique of using more datasets is about "establishing robust claims required at top-conferences". While we agree that "more is better" when it comes to evaluation, we believe that **our evaluation should not be considered inadequate** given the evaluation on 4 distinct datasets (3 from CV + LS) characterized by extreme data heterogeneity among clients ranging from 0.02 to 5k mins/client apart from several initialization strategies, in-domain and out-of-domain pre-training, etc. Additionally, as presented to Reviewer UQPF and R1Qm, we also demonstrated our **insights hold for different model sizes**, which further strengthens the robustness of our evaluation.
> > >
> > > 4. Regarding Large Models: Our 250M parameter models are objectively **10x larger** than typical FL research (<30M) and **2x larger** than prior FL-ASR work (120M). In the DP context, this scale difference is particularly significant because DP noise scales with model dimensionality. We **request the reviewer to clarify** what additional justification they would like to see beyond objective parameter counts and comparison to prior work.

---

### Official Review · Reviewer_UQPF · 2025-06-15

**Clarity:** 3
**Significance:** 3
**Originality:** 3
**Rating:** 5
**Confidence:** 2

**Summary:**

This work proposes a mechanism for end-to-end training of ASR models that centers on Differentially Private Federated Learning (DP-FL). At the core the proposed approaches utilizes layer wise clipping and gradient normalization to take a step towards DP-FL. The approach is backed by theoretical and empirical studies and analysis to demonstrate its strength.

**Questions:**

Can you give an example of user level D and D' sets that are adjacent? Each user has their own unique speech characteristics and adding/removing all their data can potentially impact the information  contained inside a dataset.

It has been highlighted that training with larger models can be challenging for DP. Therefore it would be beneficial to have an ablation study that demonstrates the impact of the proposed techniques on models of different sizes.


The rows marked blue in Table 1. They seem to be extrapolated (perhaps I missed something). To increase confidence in the approach it would be great to have at least one blue ($\epsilon$ below 10) row with non extrapolated numbers.

It seems that even small change in $\sigma_{DP}$ can impact performance. What are the author's thoughts on such an impact (and on $\epsilon$) if the number of steps is increased.


The per layer clipping in the context of DP-FL does not seem specific to ASR. Are there any characteristics that are especially more useful for ASR ?

**Ethical Concerns:**

["NO or VERY MINOR ethics concerns only"]

**Final Justification:**

This work proposes a well structured approach that is supported by theoretical and empirical analysis. Apart from ASR, the proposed approach can also be applied to other domains further demonstrating the quality and benefits of it. However, there are some weaknesses such as practical usefulness of the demonstrated $\epsilon$ and along with the variety of datasets.

The authors provided several clarifications through their rebuttal which can help mitigate some of the weaknesses. Furthermore, they also explained the practical limitations that impact some of the important weaknesses.

The score is based on these strengths and weaknesses including the impact of practical limitations.

**Limitations:**

yes

**Quality:**

3

**Strengths And Weaknesses:**

This work proposes a well structured approach and provides theoretical and empirical support/evidences for it. A well known approach of gradient clipping is utilized in the given context of DP-FL and demonstrated to be useful. Empirical evaluation and analysis through the domain of end to end ASR shows strengths of this technique. The proposed approach can potentially be applied to other domains apart from ASR and can have significant impact. The manuscript is well written and provides clear and well rounded, good quality proposal and analysis.

However, there is room for improvement. In the main results , experiments with practically useful $\epsilon$ for DP seem to be extrapolated. Ideally, the proposed approach would be combined with tractable numbers for these $\epsilon$ values. In addition to this, the proposed work can be further enriched through a more thorough ablation study and utilization of a larger variety of ASR datasets. Furthermore, the concept of per layer clipping for DP-FL seems more generic and can have applications for more than just the ASR domain. Therefore, demonstrating its impact on properties specific to ASR would have been beneficial.

---

> ### Author Rebuttal · Authors · 2025-07-30
>
> Dear Reviewer UQPF,
>
> We sincerely thank you for your thorough and constructive review of our work. We address each of your concerns below.
>
>
> > "The rows marked blue in Table 1. They seem to be extrapolated... To increase confidence in the approach, it would be great to have at least one blue (ε below 10) row with non-extrapolated numbers."
>
>
> We acknowledge this concern. The extrapolation to a practical ε value at the scale required for blue rows would require simulating millions of users with cohort sizes of \~200k. However, we cannot run such simulations since there are no public datasets that can support this scale.
>
> **Our extrapolation is based on the moments accountant** (Abadi et al. 2016 [8], McMahan et al. 2018 [46]) method, which is a standard practice in FL and DP (for vision, text, etc.) due to the absence of such large-scale public data. The extrapolation estimates epsilon when population and cohort size increase while preserving the privacy noise-to-signal ratio.
>
> We validated the above assumption by simulating different cohort sizes with ~35k users in Common-Voice (Table 16, Appendix):
>
> |Cohort Size|Dev|Test|
> |---:|---:|---:|
> |1024|13.9|16.7|
> |2048|13.6|16.3|
> |4096|12.8|15.5|
> |5120|12.7|15.4|
>
> These results demonstrate the consistent performance trend supporting the validity of the extrapolation.
>
> ---
>
> > "the proposed work can be further enriched through a more thorough ablation study and utilization of a larger variety of ASR datasets"
>
> Our choice for FL simulations is constrained by the requirements of speaker metadata:
>
> - Datasets WITH speaker IDs (suitable for FL): LibriSpeech (\~2k speakers), CommonVoice (\~35k speakers), VCTK (\~100 speakers), TED-LIUM (\~1.2k speakers), etc.
> - Datasets WITHOUT speaker IDs (unsuitable for FL): People's Speech, GigaSpeech, SPGISpeech, etc.
>
> The requirement for speaker ID to simulate realistic user partitions severely limits options. Among suitable datasets, LibriSpeech and CommonVoice provide the largest speaker diversity (see Table 2, Appendix).
>
> |Dataset|# speakers|minutes/speaker|domain|
> |---|---|---|---|
> |LibriSpeech|~2.3k|min: 2, max: 30|read-speech|
> |CV english|~35k|min:0.02, max: ~5k|conversational|
> |CV french|~6.9k|min:0.04, max: ~3k|conversational|
> |CV german|~7.1k|min:0.03, max: ~6k|conversational|
>
> This diversity ensures our methods are tested across the wide spectrum of FL challenges. Our evaluation spans 4 datasets (LS + 3 CV languages), whereas prior FL-ASR works typically evaluate on 1-2 datasets [19, 20, 56].
>
> ---
>
> > "The per layer clipping in the context of DP-FL does not seem specific to ASR. Are there any characteristics that are especially more useful for ASR?"
>
> You raise an excellent point about the generality of our approach. We view this generality as a strength that enhances rather than diminishes our contribution:
>
> - We demonstrate that large Transformer models benefit from per-layer clipping when trained using FL with DP unlike the observations of McMahan et al. (2018) [46] who report no significant improvement for LSTMs and LMs.
> - We provide insights by analyzing the ASR models and observing that layer-wise gradient heterogeneity is extreme in speech transformers (Figure 5: attention gradients are 10x smaller than LayerNorm).
>
> Given that our theoretical analysis is not specific to ASR, these insights should help other applications as well.
>
>
> ---
>
> > "It would be beneficial to have an ablation study that demonstrates the impact of the proposed techniques on models of different sizes."
>
> We agree with the suggestion and provide the following table summarizing a preliminary result for a 114M parameter model size that we were able to train within the rebuttal timeframe. We can observe that (i) the larger 250M parameter model is more affected by uniform clipping, and (ii) it also benefits more from the per-layer clipping.
>
> |model size|T|uniform-clipping|per-layer clipping|Rel. Improv.|
> |---|---|---:|---:|---:|
> |114M|2k|29.5|21.3|27.8\%|
> |250M|2k|31.1|20.4|34.4\%|
>
> This is in line with our theoretical analysis (Theorem 2), which emphasizes that per-layer clipping benefits models with greater depth H and layer-wise gradient heterogeneity (Ψ$_h$ terms), which is more prominent in deeper models (Chan et al. [23], Wang et al. [24]; discussed in Sec. 3). We will include this table in the camera-ready version.
>
> ---
>
> > "What are the author's thoughts on such an impact (and on σ) if the number of steps is increased."
>
>
> This is an insightful observation that touches on fundamental FL+DP tradeoffs:
>
> - Based on our theoretical analysis, it can be seen that privacy budget ε must be scaled as √T for T steps.
> - Given such a dependence of noise on the number of training steps, we make the design choice of limiting to T=2k steps, thus keeping the noise manageable. All FL deployments face similar constraints, i.e., longer training requires accepting either higher ε or lower utility.
> - We did not explicitly study extended training with higher noise levels since most FL studies aim to reduce the number of training steps (Azam et al. [16]) owing to the practical time limits for real-world federated learning deployments.
>
> ---
>
> > "Can you give an example of user level D and D' sets that are adjacent?" Each user has their own unique speech characteristics and adding/removing all their data can potentially impact the information contained inside a dataset.
>
>
> Certainly. In our user-level DP setting, if we consider a population comprising 3 users X (with 50 data samples), Y (with 30 data samples), and Z (with 45 data samples), then adjacent datasets D and D’ can be defined as:
>
> - Dataset D: Contains data from all users, i.e., X, Y, and Z, and
> - Dataset D': Contains speech data from only 2 users, i.e., X and Z.
>
> These datasets are adjacent because D' can be formed by removing all utterances from a single user Y. This DP guarantee ensures that an adversary observing the trained model cannot reliably determine whether User Y (with all 45 data samples) participated in training, protecting individual user privacy comprehensively.
>
> In practice, we train on the actual dataset (e.g., dataset D with all users), but add carefully calibrated noise during training such that the resulting model could plausibly have come from training on D' (without User Y). The unique speech characteristics you mention make this particularly challenging - say, User Y's distinctive speech patterns could otherwise be detectable in the model, which is why we need sufficient noise to mask their contribution while still maintaining model utility.
>
> ---
>
> **Summary**
> We appreciate your recognition that our work presents a "well-structured approach" with "good quality proposal and analysis." We have identified areas where we can improve the manuscript based on your inputs. We will include these changes in the final draft of the paper.

---

> > ### Comment · Reviewer_UQPF · 2025-08-04
> >
> > Thanks for the clarifications. I am fine with keeping the rating.

---

### Decision · Program_Chairs · 2025-09-17

**Decision:**

Accept (poster)

**Comment:**

Paper studies Differentially Private Federated Learning (DPFL) of automatic speech recognition (ASR) models. Compared to previous DPFL ASR works, authors are able to do DPFL on models owith 100s of millions of parameters and show that per-layer clipping and gradient normalization lead to better results. Authors provide theoretical analysis of their algorithm and a large set of promising empirical results.

Reviewer R1Qm pointed out concerns about scalability of the model. Authors answered that their focus was more on feasibility than scalability. Reviewer TNaa pointed out that these techniques are not novel in DP/FL literature and there exists much larger models for ASR. Authors responded by saying that FL literature usualy studied smaller models of size <30M. Authors also claimed that their results and analysis of per-layer operations departs from existing understanding in literature. However, meta-reviewer notes that references provided for these claims are very old and may not correctly charaterize the literature, e.g. (Ngyuen et al., 2023). Reviewer TNaa was not satisfied with these explanations. Reviwer UQPF had concerns about diversity of datasets and extrapolated results. Authors addressed these concerns with additional experiments. Overall, while the results are promising and extensive, the novelty of results is somewhat limited.

(Ngyuen et al., 2023) Nguyen, T. N., Nguyen, P. H., Nguyen, L. M., & Van Dijk, M. (2023). Batch clipping and adaptive layerwise clipping for differential private stochastic gradient descent. arXiv preprint arXiv:2307.11939.